

# One-shot holography

**Chris Akers[1]⋆, Adam Levine[1,2]†, Geoff Penington[2,3]‡ and Elizabeth Wildenhain[3]◦**

**1** Center for Theoretical Physics,
Massachusetts Institute of Technology, Cambridge, MA 02139, USA
**2** Institute for Advanced Study, Princeton, NJ 08540 USA
**3** Center for Theoretical Physics, University of California, Berkeley, CA 94720 USA

⋆ cakers@mit.edu , † arlevine@mit.edu ,
‡ geoffp@berkeley.edu , ◦ elizabeth_wildenhain@berkeley.edu

## Abstract

Following the work of [1], we define a generally covariant *max-entanglement wedge* of a boundary region *B*, which we conjecture to be the bulk region reconstructible from *B*. We similarly define a covariant *min-entanglement wedge*, which we conjecture to be the bulk region that can influence the state on *B*. We prove that the min- and max-entanglement wedges obey various properties necessary for this conjecture, such as nesting, inclusion of the causal wedge, and a reduction to the usual quantum extremal surface prescription in the appropriate special cases. These proofs rely on one-shot versions of the (restricted) quantum focusing conjecture (QFC) that we conjecture to hold. We argue that these QFCs imply a one-shot generalized second law (GSL) and quantum Bousso bound. Moreover, in a particular semiclassical limit we prove this one-shot GSL directly using algebraic techniques. Finally, in order to derive our results, we extend both the frameworks of one-shot quantum Shannon theory and state-specific reconstruction to finite-dimensional von Neumann algebras, allowing nontrivial centers.

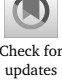

# 1 Introduction

In AdS/CFT, the entanglement wedge $\text{EW}(B)$ of a boundary region $B$ is a bulk region $b$ such that [2–16]

    1. All information within $b$ can be reconstructed from $B$,

    2. No information outside $b$ can be reconstructed from $B$.

In this sense, $\text{EW}(B)$ is holographically dual to $B$. Whether a given $b$ satisfies each condition depends on the state, and it was shown in [1] that there are many semiclassical gravity states for which no bulk region simultaneously satisfies both. For such states, therefore, no entanglement wedge exists.[1]

When $\text{EW}(B)$ does exist, however, one can find it using the following well-known "quantum extremal surface" (QES) prescription. Consider all bulk regions $b$ with conformal boundary $B$. To each $b$ assign the generalized entropy

$$S_{\text{gen}}(b) := \frac{A(\eth b)}{4G} + S(b), \tag{1}$$

where $A(\eth b)$ is the area of the edge $\eth b$ of $b$, and $S(b)$ is the von Neumann entropy of quantum fields in $b$. The region $b$ is said to be quantum extremal – and its edge $\eth b$ is called a quantum extremal surface – if $S_{\text{gen}}(b)$ is unchanged at linear order under local deformations of $\eth b$. The entanglement wedge $\text{EW}(B)$ is the quantum extremal region $b$ with minimal generalized entropy. The QES prescription further says that the boundary entanglement entropy is then given by

$$S(B) = S_{\text{gen}}(\text{EW}(B)). \tag{2}$$

---

[1]One example is the following. Consider an AdS-size black hole, and let $B$ be a spherical region that is 60% of the boundary. If the black hole is in a pure state $\rho_{\text{pure}}$, $\text{EW}(B)$ includes the black hole. If it is in a thermal state $\rho_{\text{therm}}$, then $\text{EW}(B)$ excludes the black hole. However, if the black hole is in a mixture $\frac{1}{2}\rho_{\text{pure}} + \frac{1}{2}\rho_{\text{therm}}$, then $\text{EW}(B)$ does not exist. $B$ has partial information about the black hole.

For states where no region satisfies both conditions 1 and 2, there is no entanglement wedge for the QES prescription to find, and the region $b$ found by it has no operational significance. Still, one might hope to classify the regions satisfying condition 1 and 2 separately by similar prescriptions. Exactly this was proposed in [1]. The largest region $b$ satisfying condition 1 was conjectured to be a region named the *max-entanglement wedge* (max-EW). Meanwhile, the smallest $b$ satisfying condition 2 was conjectured to be a different region named the *min-entanglement wedge* (min-EW). Both regions were defined using prescriptions analogous to the QES prescription. Indeed, it was shown that whenever the min-EW and max-EW coincided – and hence an entanglement wedge satisfying both conditions existed – the max-EW and min-EW always agreed with the region found by the traditional QES prescription. It is only when this occurs that the formula (2) for the entanglement entropy $S(B)$ is correct (even as a leading-order semiclassical approximation).[2]

The definitions of max- and min-EWs given in [1], however, were valid only for two special classes of spacetime. The first was spacetimes with a moment of time-reflection symmetry, for example static spacetimes. The second was spacetimes where all but two quantum extremal surfaces could be neglected in replica trick computations. In this work, we propose generally covariant definitions of the max- and min-EWs that are applicable in any spacetime, thus significantly extending the conjecture of [1].

To define the min- and max-EWs more precisely, we must first review some ideas from "one-shot quantum Shannon theory", which lie at the heart of our conjectures. Traditional (non-one-shot) quantum Shannon theory quantifies the information in a quantum state by studying tasks involving an infinite number of copies of the same state (often referred to as "the asymptotic i.i.d. limit"). Consider for example the communication task of quantum state merging, which will be important for us. The goal is to extract all information in a system $AB$ given access only to the subsystem $B$, along with a minimal number of additional qubits containing information about $A$.[3] When merging a large number of copies of the same quantum state, the minimal number of qubits required, per copy, is given by the conditional von Neumann entropy $S(AB) - S(B)$ [20].[4] On the other hand, the number of qubits required to merge a single copy of the state, up to errors set by some small $\varepsilon$, is given by a different, one-shot entropic quantity called the smooth conditional max-entropy $H_{\max}^{\varepsilon}(AB|B)$ [21]. (We give a formal definition of this quantity in Section 2.1.)

A rough definition of max-EW($B$) is that it is the largest region $b_1$ such that all information in $b_1$ can flow to $B$ through some Cauchy slice of $b_1$ via one-shot quantum state merging.[5] By this we mean that every subregion $b_2 \subseteq b_1$ with edge in that Cauchy slice satisfies

$$H_{\max}^{\varepsilon}(b_1|b_2) < \frac{A(\eth b_2)}{4G} - \frac{A(\eth b_1)}{4G}. \tag{3}$$

Similarly, min-EW($B$) is roughly the smallest region $b_3$ such that all information outside $b_3$ can flow through some Cauchy slice to the complementary boundary subregion $B'$ via one-shot quantum state-merging.

Notably, the distinction between these new definitions and the QES prescription comes entirely from the difference between one-shot and traditional quantum state merging. If traditional state merging through a Cauchy slice was sufficient to allow bulk reconstruction, the max- and min-EWs would always be the same and the traditional QES prescription would always be valid. However, it is instead *one-shot* quantum Shannon theory that determines

---

[2]See also [17–19] for additional discussion.

[3]One also has access to unlimited classical bits (or more generally zero-bits) containing information about $A$.

[4]Note that this conditional entropy may be negative! Bell pairs shared between $A$ and $B$ act as a resource that can be used to teleport other qubits via the free classical information. When negative, $S(AB) - S(B)$ counts how many such Bell pairs can be recovered from the state.

[5]As we shall see, for this prescription to make sense one must additionally require that $b_1$ be (max-)antinormal.

whether bulk information is accessible from a boundary subregion. This is perhaps unsurprising, because the holographic (bulk-to-boundary) map acts only on a single copy of the bulk state.

Having defined the min- and max-EW, we then corroborate their conjectured interpretations by proving that they satisfy a number of important properties. First, whenever the min- and max-EW coincide, we show that they match the traditional QES prescription for the entanglement wedge. Second, they limit to (a minor modification of) the definitions of [1] in the appropriate special cases. Finally, we show that they satisfy important consistency checks, such as nesting: max-EW$(B_1) \supseteq$ max-EW$(B_2)$ if $B_1 \supseteq B_2$.

To prove these results, we assume the validity of two new conjectures, closely related to the "quantum focusing conjecture" (QFC) of [22], which we call the min-QFC and max-QFC. Like the original QFC, these min- and max-QFCs imply many interesting results of independent interest.

The structure of the paper is as follows. In Section 2, we briefly review definitions from one-shot quantum Shannon theory and then generalize them to finite-dimensional von Neumann algebras. In Section 3, we apply those ideas to quantum gravity to define "generalized min- and max-entropies" that combine one-shot bulk entropies with area term contributions. In Section 4, we define one-shot quantum expansions and conjecture one-shot versions of the quantum focusing conjecture. In Section 5, we propose our definition of the max- and min-EWs and establish various properties for them. In Section 6, we explain how one-shot generalized entropies are concretely realized in the recently discovered Type II von Neumann algebras describing semiclassical black holes. In Section 7, we discuss the conceptual significance of our results along with open questions. Finally, in appendices, we prove various technical results about one-shot quantum Shannon theory for algebras and give a definition of state-specific reconstruction for algebras with centers, generalizing earlier work in [14].

## 2 One-shot entropies for algebras

It is our goal to discuss the one-shot quantum Shannon theory of subregions in semiclassical gravity. In this section we take the first step. In Section 2.1, we briefly review the main definitions from one-shot quantum Shannon theory in the traditional setting of a tensor product factorization of Hilbert space. (For a gentler introduction for a quantum gravity audience see [1]. For a thorough treatment see [23], and also [24–30].) Then in Section 2.2 we generalize those definitions to (finite-dimensional) von Neumann algebras.

### 2.1 Review: One-shot quantum Shannon theory

For all proofs of theorems in this subsection see [23].

**Definition 2.1** (Conditional entropies). Given a density matrix $\rho_{AB}$ on $\mathcal{H}_A \otimes \mathcal{H}_B$, the min-entropy, von Neumann entropy, and max-entropy of $AB$ conditioned on $B$ are

$$H_{\min}(AB|B)_\rho := -\min_\sigma \inf\{\lambda : \rho_A \leq e^\lambda \mathbb{1}_A \otimes \sigma_B\}, \tag{4}$$

$$S(AB|B)_\rho := -\operatorname{Tr}_{AB}[\rho_{AB} \log \rho_{AB}] + \operatorname{Tr}_B[\rho_B \log \rho_B], \tag{5}$$

$$H_{\max}(AB|B)_\rho := \sup_\sigma \log\left(\operatorname{Tr}_A\left[\sqrt{\sigma_B^{1/2} \rho_{AB} \sigma_B^{1/2}}\right]\right)^2, \tag{6}$$

where $\rho_B = \operatorname{Tr}_A[\rho_{AB}]$, $\mathbb{1}_A$ is the identity operator on $\mathcal{H}_A$, and the minimization and supremum are taken over all sub-normalized density matrices $\sigma_B$ on $\mathcal{H}_B$.

The (conditional) min-entropy and max-entropy are sometimes called the (conditional) one-shot entropies.

**Remark 2.2.** The terminology and notation used in Definition 2.1 is non-standard. More commonly, one would refer for example to the conditional von Neumann entropy of $A$ conditioned on $B$ as

$$S(A|B) = S(AB) - S(B),$$

with $S(C) = -\text{Tr}(\rho_C \log \rho_C)$. Similar notation is also standard for the conditional min- and max-entropies. However our choice of notation will be convenient later in the algebraic context where there is no analogue of the subsystem $A$ independent of $B$.

**Remark 2.3.** In the special case that $\mathcal{H}_B$ is trivial, we write $H_{\min}(A)_\rho$, $S(A)_\rho$, and $H_{\max}(A)_\rho$ and call them the (unconditional) min-entropy, von Neumann entropy, and max-entropy respectively.

**Remark 2.4.** While the conditional von Neumann entropy equals the difference of two unconditional von Neumann entropies, in general the conditional one-shot entropies do not. Instead, they are bounded by such differences via the chain rule inequality, Theorem 2.10 below.

It is often useful to allow for small errors, and for this one defines the smooth one-shot entropies. Let $\mathcal{P}_\le(AB)$ denote the set of density matrices on $\mathcal{H}_A \otimes \mathcal{H}_B$ with trace less than or equal to 1.

**Definition 2.5** (Purified distance). Let $\rho, \sigma \in \mathcal{P}_\le(AB)$. The purified distance between $\rho$ and $\sigma$ is

$$P(\rho, \sigma) := \sqrt{1 - F_*(\rho, \sigma)^2}, \tag{7}$$

where $F_*(\rho, \sigma)$ is the generalized fidelity between $\rho$ and $\sigma$, defined as

$$F_*(\rho, \sigma) := F(\rho, \sigma) + \sqrt{(1 - \text{Tr}[\rho])(1 - \text{Tr}[\sigma])}, \tag{8}$$

and $F(\rho, \sigma) := \|\sqrt{\rho}\sqrt{\sigma}\|_1$ is the (standard) fidelity, with $\|X\|_1 := \text{Tr}\sqrt{X^\dagger X}$.

**Definition 2.6** (Smooth conditional one-shot entropies). Let $\rho_{AB}$ be a normalized density matrix on $\mathcal{H}_A \otimes \mathcal{H}_B$, and let $\varepsilon > 0$. The smooth conditional min-entropy and max-entropy are

$$H_{\min}^\varepsilon(AB|B)_\rho := \sup_{\rho^\varepsilon \in \mathcal{P}_\le(AB), P(\rho^\varepsilon, \rho) \le \varepsilon} H_{\min}(AB|B)_{\rho^\varepsilon}, \tag{9}$$

$$H_{\max}^\varepsilon(AB|B)_\rho := \inf_{\rho^\varepsilon \in \mathcal{P}_\le(AB), P(\rho^\varepsilon, \rho) \le \varepsilon} H_{\max}(AB|B)_{\rho^\varepsilon}. \tag{10}$$

These have the following important properties – see [23] for proofs.

**Theorem 2.7** (Duality between min- and max-entropies). *For all $|\psi\rangle \in \mathcal{H}_A \otimes \mathcal{H}_B \otimes \mathcal{H}_C$,*

$$H_{\min}(AB|B)_\psi = -H_{\max}(AC|C)_\psi. \tag{11}$$

*Furthermore, this continues to hold under smoothing:*

$$H_{\min}^\varepsilon(AB|B)_\psi = -H_{\max}^\varepsilon(AC|C)_\psi. \tag{12}$$

**Remark 2.8.** Theorem 2.7 is the "one-shot version" of the easily-verifiable equality

$$S(AB|B)_\psi = -S(AC|C)_\psi. \tag{13}$$

**Theorem 2.9** (Quantum asymptotic equipartition principle)**.** *Let $\rho_{AB}$ be a normalized density matrix on Hilbert space $\mathcal{H}_A \otimes \mathcal{H}_B$, and let $0 < \varepsilon < 1$. It holds that*

$$\lim_{n\to\infty} \frac{1}{n} H^\varepsilon_{\min}(A^n B^n | B^n)_{\rho^{\otimes n}} = S(AB|B)_\rho = \lim_{n\to\infty} \frac{1}{n} H^\varepsilon_{\max}(A^n B^n | B^n)_{\rho^{\otimes n}}, \tag{14}$$

*where $A^n, B^n$ denote the union of each $A, B$ factor respectively from each of the $n$ copies.*

**Theorem 2.10** (Chain rule)**.** *Let $\rho_{ABC}$ be a normalized density matrix on Hilbert space $\mathcal{H}_A \otimes \mathcal{H}_B \otimes \mathcal{H}_C$. For $\varepsilon > 2\varepsilon' > 0$,*

$$H^\varepsilon_{\min}(ABC|C)_\rho \geq H^{\varepsilon'}_{\min}(ABC|BC)_\rho + H^{\varepsilon'}_{\min}(BC|C)_\rho + \mathcal{O}\left(\log\left(\frac{1}{\varepsilon - 2\varepsilon'}\right)\right), \tag{15}$$

$$S(ABC|C)_\rho = S(ABC|BC)_\rho + S(BC|C)_\rho, \tag{16}$$

$$H^\varepsilon_{\max}(ABC|C)_\rho \leq H^{\varepsilon'}_{\max}(ABC|BC)_\rho + H^{\varepsilon'}_{\max}(BC|C)_\rho + \mathcal{O}\left(\log\left(\frac{1}{\varepsilon - 2\varepsilon'}\right)\right). \tag{17}$$

**Theorem 2.11** (Strong subadditivity)**.** *Let $\rho_{ABC}$ be a normalized density matrix on Hilbert space $\mathcal{H}_A \otimes \mathcal{H}_B \otimes \mathcal{H}_C$. For $\varepsilon \geq 0$, it holds that*

$$H^\varepsilon_{\min}(ABC|BC)_\rho \leq H^\varepsilon_{\min}(AB|B)_\rho, \tag{18}$$

$$S(ABC|BC)_\rho \leq S(AB|B)_\rho, \tag{19}$$

$$H^\varepsilon_{\max}(ABC|BC)_\rho \leq H^\varepsilon_{\max}(AB|B)_\rho. \tag{20}$$

## 2.2 One-shot entropies for von Neumann algebras

We now generalize the statements of one-shot quantum Shannon theory to finite-dimensional von Neumann algebras, possibly with non-trivial center. This requires us to handle a number of additional subtleties, including an ambiguity in the trace which will be important in gravity.

Although we restrict to finite-dimensional algebras here for simplicity (and because the subtleties of von Neumann algebras in infinite-dimensions are not very important for our purposes), we expect that our framework generalizes straightforwardly to any finite von Neumann algebras (including e.g. Type II$_1$ algebras) and that large parts generalize to any semifinite algebra. We will briefly discuss how our results are related to the semifinite Type II$_\infty$ algebras that describe black holes in the semiclassical $G \to 0$ limit in Section 6.

Our presentation here will be self-contained, although closely related ideas have previously appeared in the literature. In particular, a related but different definition of conditional one-shot entropies for von Neumann algberas was considered in [31], which restricted to algebras of the form $\mathcal{M}_{AB} = \mathcal{B}(\mathcal{H}_A) \otimes \mathcal{M}_B$, for a general Hilbert space $\mathcal{H}_A$ and general von Neumann algebra $\mathcal{M}_B$, where $\mathcal{B}(\mathcal{H}_A)$ denotes the algebra of bounded operators. In contrast, here we let $AB$ be associated with a finite-dimensional algebra which does not necessarily factorize between $A$ and $B$. Indeed, we avoid talking about the analog of $AB \setminus B$ at all, because in our applications it is not necessarily associated to an algebra. In line with this, our notation starting in this subsection is to denote the joint algebra as simply $A$. Additionally, entropic certainty relations closely related to duality (Theorem 2.33) were proven for von Neumann algebras in [32] and an asymptotic equipartition principle (Theorem 2.34) was proven for the max-relative entropy in any von Neumann algebra in [33].

We use the following notation. Let $\mathcal{L}(\mathcal{H})$ denote the set of linear operators acting on a Hilbert space $\mathcal{H}$. For a von Neumann algebra $\mathcal{M} \subseteq \mathcal{L}(\mathcal{H})$, let $\mathcal{M}'$ denote its commutant, the subset of $\mathcal{L}(\mathcal{H})$ that commutes with $\mathcal{M}$. Let $Z(\mathcal{M}) = \mathcal{M} \cap \mathcal{M}'$ denote its center. $\mathcal{M}$ is called a factor if $Z(\mathcal{M})$ is trivial, meaning it contains only multiples of the identity operator.

Recall the following theorem.

**Theorem 2.12** (Structure theorem of finite-dimensional algebras (Theorem A.6 of [10])). *Let $\mathcal{M}_A$ be a von Neumann algebra acting on $\mathcal{H}$ and let $\dim \mathcal{H} < \infty$. Then there is a direct sum decomposition $\mathcal{H} = \oplus_\alpha \left( \mathcal{H}_{A_\alpha} \otimes \mathcal{H}_{A'_\alpha} \right)$ such that*

$$
\begin{aligned}
\mathcal{M}_A &= \bigoplus_\alpha \left( \mathcal{L}(\mathcal{H}_{A_\alpha}) \otimes \mathbb{1}_{A'_\alpha} \right), \\
\mathcal{M}'_A &= \bigoplus_\alpha \left( \mathbb{1}_{A_\alpha} \otimes \mathcal{L}(\mathcal{H}_{A'_\alpha}) \right).
\end{aligned}
\tag{21}
$$

**Remark 2.13.** From now on we will let *algebra* denote finite-dimensional von Neumann algebra unless otherwise stated.

Given operators $p, q \in \mathcal{L}(\mathcal{H})$, we say $q \leq p$ if $p - q$ is a positive semidefinite operator.

**Definition 2.14** (Minimal central projector). An operator $p \in \mathcal{L}(\mathcal{H})$ is a projector if $p^\dagger = p$ and $p^2 = p$. An operator $p \in Z(\mathcal{M})$ is a *minimal central projector* if it is a projector and for any projector $q \in Z(\mathcal{M})$ we have $q \leq p$ if and only if $q = 0$ or $q = p$.

**Remark 2.15.** Let $\mathcal{M}$ be an algebra on $\mathcal{H}$. With respect to the decomposition of $\mathcal{H}$ in Theorem 2.12, each minimal central projector in $\mathcal{M}$ projects onto a single $\alpha$-sector in (21), and hence we will call them $p_\alpha$:

$$
p_\alpha \mathcal{H} = \mathcal{H}_{A_\alpha} \otimes \mathcal{H}_{A'_\alpha}.
\tag{22}
$$

Note these $p_\alpha$ satisfy $p_\alpha p_\beta = p_\alpha \delta_{\alpha\beta}$ and $\sum_\alpha p_\alpha = \mathbb{1}$.

**Remark 2.16.** Any operator $C \in Z(\mathcal{M})$ can be expanded as $C = \sum_\alpha C_\alpha p_\alpha$ with $C_\alpha \in \mathbb{C}$.

We now introduce the general notion of a trace that will play an important role.

**Definition 2.17** (Trace). Let $\mathcal{M}_A$ be an algebra on a Hilbert space $\mathcal{H}$. A map $\mathrm{tr}_A : \mathcal{M}_A \to \mathbb{C}$ is said to be a trace on $\mathcal{M}_A$ if for all non-zero $m_1, m_2 \in \mathcal{M}_A$,

$$
\mathrm{tr}_A[m_1 m_2] = \mathrm{tr}_A[m_2 m_1],
\tag{23}
$$

$$
\mathrm{tr}_A[m_1 m_1^\dagger] > 0.
\tag{24}
$$

**Remark 2.18.** Given a trace $\mathrm{tr}_A$ on $\mathcal{M}_A$, then the linear functional $\mathrm{tr}'_A$ is a trace if and only if there exists a positive invertible central operator $C \in Z(\mathcal{M}_A)$ such that

$$
\mathrm{tr}'_A[\cdot] = \mathrm{tr}_A[C(\cdot)].
\tag{25}
$$

**Remark 2.19.** We will distinguish between the trace on a Hilbert space and a trace on an algebra, denoting the former by upper-case, Tr, and the latter by lower-case, tr.

**Remark 2.20.** By Remark 2.16, we can relate any algebraic trace on $\mathcal{M}_A$ to the Hilbert space trace on each sector $\mathcal{H}_{A_\alpha}$ by

$$
\mathrm{tr}_A(\cdot) = \sum_\alpha C_\alpha^A \mathrm{Tr}_{A_\alpha}[p_\alpha (\cdot) p_\alpha],
\tag{26}
$$

for some set of coefficients $C_\alpha^A > 0$ that can be computed as

$$
C_\alpha^A = \frac{1}{\dim A_\alpha} \mathrm{tr}_A[p_\alpha].
\tag{27}
$$

**Definition 2.21** (Canonical trace). We define the *canonical trace* $\text{tr}_{A,\text{can}}$ on $\mathcal{M}_A$ to be the trace with $C_\alpha^A = 1$ for all sectors $\alpha$.

Note the trace Tr is defined by a sum over a complete set of states on a Hilbert space. The canonical trace is its natural extension to algebras.

**Definition 2.22** (Complementary traces). Let $\mathcal{M}_A$ be an algebra acting on $\mathcal{H}$ and let $\mathcal{M}_{A'} := \mathcal{M}_A'$. Let $\text{tr}_A$, $\text{tr}_{A'}$ be traces for $\mathcal{M}_A$, $\mathcal{M}_{A'}$ respectively. We say these traces are complementary if

$$C_\alpha^A = C_\alpha^{A'}, \tag{28}$$

with $C_\alpha^A$, $C_\alpha^{A'}$ as defined in (27).

**Remark 2.23.** As we will see below, gravity will naturally assign complementary traces to the algebras associated to complementary subregions.

**Definition 2.24** (Density matrix). Let $\mathcal{M}$ be an algebra on Hilbert space $\mathcal{H}$ with trace tr. A positive semi-definite $\rho \in \mathcal{M}$ is a normalized density matrix if $\text{tr}(\rho) = 1$, and is subnormalized if $\text{tr}(\rho) \leq 1$. It is said to be a density matrix on $\mathcal{M}$ for $|\psi\rangle \in \mathcal{H}$ if

$$\text{tr}(\rho\, m) = \langle\psi|m|\psi\rangle, \quad \forall\, m \in \mathcal{M}. \tag{29}$$

**Remark 2.25.** Density matrices always exist and are unique. Note that the density matrix $\rho$ depends not only on the state $|\psi\rangle$ and the algebra $\mathcal{M}$ but also on the trace tr. In contrast, the *reduced state* $\psi$ of $|\psi\rangle$ on the algebra $\mathcal{M}$ is defined as the linear functional

$$\psi(m) = \langle\psi|m|\psi\rangle, \quad \forall\, m \in \mathcal{M}, \tag{30}$$

and is trace-independent.

**Remark 2.26.** The canonical density matrix $\rho_{A,\text{can}}$ for the state $|\psi\rangle$ on the algebra $\mathcal{M}_A$ with respect to the canonical trace $\text{tr}_{A,\text{can}}$ can be written as

$$\rho_{A,\text{can}} = \oplus_\alpha q_\alpha \rho_{A_\alpha}, \tag{31}$$

where $q_\alpha = \langle\psi|p_\alpha|\psi\rangle$ is the probability of the state $|\psi\rangle$ being in sector $\alpha$ and $\rho_{A_\alpha}$ is the reduced density matrix of $p_\alpha|\psi\rangle$ on $\mathcal{H}_{A_\alpha}$.

**Remark 2.27.** The density matrix $\rho_A$ associated to an arbitrary trace $\text{tr}_A$ can be written as

$$\rho_A = C^{-1}\rho_{A,\text{can}}, \tag{32}$$

where $C = \sum_\alpha p_\alpha C_\alpha^A$ is defined as in Remark 2.20.

**Definition 2.28** (Conditional entropies). Let $\mathcal{M}_B \subseteq \mathcal{M}_A$ be algebras on a Hilbert space $\mathcal{H}$, with traces $\text{tr}_B$ and $\text{tr}_A$ respectively. Given a state $|\psi\rangle \in \mathcal{H}$, the min-entropy, von Neumann entropy, and max-entropy of $A$ conditioned on $B$ are

$$H_{\min}(A|B)_\psi := -\min_\sigma \inf\{\lambda : \rho_A \leq e^\lambda \sigma_B\}, \tag{33}$$

$$S(A|B)_\psi := -\text{tr}_A(\rho_A \log \rho_A) + \text{tr}_B(\rho_B \log \rho_B), \tag{34}$$

$$H_{\max}(A|B)_\psi := \sup_\sigma \log\left(\text{tr}_A\sqrt{\sigma_B^{1/2}\rho_A\sigma_B^{1/2}}\right)^2, \tag{35}$$

where $\rho_A$, $\rho_B$ are sub-normalized density matrices on $\mathcal{M}_A$, $\mathcal{M}_B$ for $|\psi\rangle$ and the minimization and supremum are taken over all sub-normalized density matrices $\sigma_B$ on $\mathcal{M}_B$, i.e. $\text{tr}_B \sigma_B \leq 1$.

The (conditional) min-entropy and max-entropy are sometimes called the (conditional) one-shot entropies. We will often drop the $\psi$ subscript when it is clear from context.

**Remark 2.29.** In the special case that $\mathcal{M}_B$ is trivial, including only multiples of the identity, we write $H_{\min}(A)_\psi$, $S(A)_\psi$, and $H_{\max}(A)_\psi$ and call them the (unconditional) min-entropy, von Neumann entropy, and max-entropy respectively.

Given a von Neumann algebra $\mathcal{M}$ with trace tr, let $\mathcal{P}_\le(\mathcal{M})$ denote the set of subnormalized density matrices on $\mathcal{M}$.

**Definition 2.30** (Purified distance). Let $\rho, \sigma \in \mathcal{P}_\le(\mathcal{M})$. The purified distance between $\rho$ and $\sigma$ is

$$P(\rho, \sigma) := \sqrt{1 - F_*(\rho, \sigma)^2}, \tag{36}$$

where $F_*(\rho, \sigma)$ is the generalized fidelity between $\rho$ and $\sigma$, defined as

$$F_*(\rho, \sigma) := F(\rho, \sigma) + \sqrt{(1 - \operatorname{tr}[\rho])(1 - \operatorname{tr}[\sigma])}, \tag{37}$$

and $F(\rho, \sigma) := \|\sqrt{\rho}\sqrt{\sigma}\|_1$ is the (standard) fidelity, with $\|X\|_1 := \operatorname{tr}\sqrt{X^\dagger X}$.

**Definition 2.31** (Smooth conditional one-shot entropies). Let $\mathcal{M}_B \subseteq \mathcal{M}_A$ be algebras on a Hilbert space $\mathcal{H}$. Let $|\psi\rangle \in \mathcal{H}$, $\varepsilon > 0$. Furthermore, let $\rho \in \mathcal{M}_A$ be a density matrix on $\mathcal{M}_A$ for $|\psi\rangle$. The smooth conditional min-entropy and max-entropy are

$$H_{\min}^\varepsilon(A|B)_\psi := \max_{\rho^\varepsilon \in \mathcal{P}_\le(\mathcal{M}_A), P(\rho^\varepsilon, \rho) \le \varepsilon} H_{\min}(A|B)_{\rho^\varepsilon}, \tag{38}$$

$$H_{\max}^\varepsilon(A|B)_\psi := \min_{\rho^\varepsilon \in \mathcal{P}_\le(\mathcal{M}_A), P(\rho^\varepsilon, \rho) \le \varepsilon} H_{\max}(A|B)_{\rho^\varepsilon}. \tag{39}$$

**Theorem 2.32.** *Let $\mathcal{M}_B \subseteq \mathcal{M}_A$ be algebras on a Hilbert space $\mathcal{H}$, and $|\psi\rangle \in \mathcal{H}$. Then*

$$H_{\min}(A|B)_\psi \le S(A|B)_\psi \le H_{\max}(A|B)_\psi. \tag{40}$$

*Furthermore, this continues to hold for sufficiently small $\varepsilon > 0$,*

$$H_{\min}^\varepsilon(A|B)_\psi \le S(A|B)_\psi \le H_{\max}^\varepsilon(A|B)_\psi. \tag{41}$$

*Proof.* See Appendix A.4. $\qquad\square$

**Theorem 2.33** (Duality between min- and max-entropies). *Let $\mathcal{M}_B \subseteq \mathcal{M}_A$ be algebras on a Hilbert space $\mathcal{H}$ and denote their commutants by $\mathcal{M}_{A'} := \mathcal{M}_A'$ and $\mathcal{M}_{B'} := \mathcal{M}_B'$. Assuming that the traces for $\mathcal{M}_A$, $\mathcal{M}_{A'}$ and $\mathcal{M}_B$, $\mathcal{M}_{B'}$ are respectively complementary, then for any pure state $|\psi\rangle \in \mathcal{H}$ it holds that*

$$H_{\min}(A|B)_\psi = -H_{\max}(B'|A')_\psi. \tag{42}$$

*Furthermore, this continues to hold under smoothing:*

$$H_{\min}^\varepsilon(A|B)_\psi = -H_{\max}^\varepsilon(B'|A')_\psi. \tag{43}$$

*Proof.* See Appendix A.1. Using the appendix, one can check that the equality continues to hold under smoothing because of the choice to use the purified distance (36) as the metric on states. Other metrics – like the trace distance – would have led to an inequality.[6] $\qquad\square$

---

[6]These duality relations are an example of so-called entropic certainty relations which were explored in the setting of finite dimensional quantum systems in [34] and discussed in the context of QFT in [32]. We thank Thomas Faulkner for pointing out this connection to us.

**Theorem 2.34** (Quantum asymptotic equipartition principle). *Let $\mathcal{M}_B \subseteq \mathcal{M}_A$ be algebras on a Hilbert space $\mathcal{H}$, let $|\psi\rangle \in \mathcal{H}$, and let $0 < \varepsilon < 1$. It holds that*

$$\lim_{n \to \infty} \frac{1}{n} H_{\min}^{\varepsilon}(A^n|B^n)_{\psi^{\otimes n}} = S(A|B)_{\psi} = \lim_{n \to \infty} \frac{1}{n} H_{\max}^{\varepsilon}(A^n|B^n)_{\psi^{\otimes n}}. \tag{44}$$

*Proof.* See Appendix A.3.[7] □

**Theorem 2.35** (Chain rule). *Let $\mathcal{M}_A \supseteq \mathcal{M}_B \supseteq \mathcal{M}_C$ be von Neumann algebras on Hilbert space $\mathcal{H}$, and let $|\psi\rangle \in \mathcal{H}$. The chain rule states that for $\varepsilon > 2\varepsilon' > 0$, then*

$$H_{\min}^{\varepsilon}(A|C) \geq H_{\min}^{\varepsilon'}(A|B) + H_{\min}^{\varepsilon'}(B|C) + \mathcal{O}\left(\log\left(\frac{1}{\varepsilon - 2\varepsilon'}\right)\right), \tag{45}$$

$$S(A|C) = S(A|B) + S(B|C), \tag{46}$$

$$H_{\max}^{\varepsilon}(A|C) \leq H_{\max}^{\varepsilon'}(A|B) + H_{\max}^{\varepsilon'}(B|C) + \mathcal{O}\left(\log\left(\frac{1}{\varepsilon - 2\varepsilon'}\right)\right). \tag{47}$$

*Proof.* See Appendix A.5. □

**Definition 2.36** (Partial trace). Let $\mathcal{M} \supset \mathcal{N}$ be algebras with corresponding traces $\mathrm{tr}_{\mathcal{M}}$ and $\mathrm{tr}_{\mathcal{N}}$. A partial trace from $\mathcal{M}$ to $\mathcal{N}$ is a completely positive and trace-preserving linear map $\mathrm{tr}_{\mathcal{M} \to \mathcal{N}} : \mathcal{M} \to \mathcal{N}$ which obeys the so-called bi-module property[8]

$$\mathrm{tr}_{\mathcal{M} \to \mathcal{N}}[n_1 m n_2] = n_1 \, \mathrm{tr}_{\mathcal{M} \to \mathcal{N}}[m] n_2, \quad \forall \, n_1, n_2 \in \mathcal{N} \text{ and } m \in \mathcal{M}. \tag{48}$$

**Remark 2.37.** One can check that in the setting of the previous section, if we have $\mathcal{H}_{XY} = \mathcal{H}_X \otimes \mathcal{H}_Y$, with algebras $\mathcal{M}_X = \mathcal{L}(\mathcal{H}_X)$ and $\mathcal{M}_Y = \mathcal{L}(\mathcal{H}_Y)$, then the map

$$\mathrm{tr}_{XY \to Y}[\cdot] = \mathrm{Tr}_X[\cdot], \tag{49}$$

defines a partial trace from $\mathcal{L}(\mathcal{H}_{XY}) \to \mathcal{L}(\mathcal{H}_Y)$.

**Theorem 2.38.** *There exists a unique partial trace $\mathrm{tr}_{\mathcal{M} \to \mathcal{N}}$ for any algebras $\mathcal{M} \supseteq \mathcal{N}$ and pair of traces $\mathrm{tr}_{\mathcal{M}}$ and $\mathrm{tr}_{\mathcal{N}}$.*

*Proof.* Given a density matrix $\rho_{\mathcal{M}} \in \mathcal{M}$, there is a unique density matrix $\rho_{\mathcal{N}} \in \mathcal{N}$ such that for all $n \in \mathcal{N}$,

$$\mathrm{tr}_{\mathcal{M}}[\rho_{\mathcal{M}} n] = \mathrm{tr}_{\mathcal{N}}[\rho_{\mathcal{N}} n]. \tag{50}$$

Define $\mathrm{tr}_{\mathcal{M} \to \mathcal{N}} : \mathcal{M} \to \mathcal{N}$ such that for all $\rho_{\mathcal{M}}$ it holds that $\mathrm{tr}_{\mathcal{M} \to \mathcal{N}}[\rho_{\mathcal{M}}] = \rho_{\mathcal{N}}$. Then linearly extend $\mathrm{tr}_{\mathcal{M} \to \mathcal{N}}$ to all operators in $\mathcal{M}$. It follows that for all $m \in \mathcal{M}$ and $n \in \mathcal{N}$,

$$\mathrm{tr}_{\mathcal{N}}[\mathrm{tr}_{\mathcal{M} \to \mathcal{N}}[m] n] = \mathrm{tr}_{\mathcal{M}}[mn]. \tag{51}$$

By construction, $\mathrm{tr}_{\mathcal{M} \to \mathcal{N}}$ is trace-preserving and completely positive. Moreover, $\mathrm{tr}_{\mathcal{M} \to \mathcal{N}}$ obeys the bi-module property, because for all $n \in \mathcal{N}$,

$$\mathrm{tr}_{\mathcal{N}}\left(\mathrm{tr}_{\mathcal{M} \to \mathcal{N}}(n_1 m n_2) n\right) = \mathrm{tr}_{\mathcal{M}}\left(n_1 m n_2 n\right) = \mathrm{tr}_{\mathcal{N}}\left(n_1 \, \mathrm{tr}_{\mathcal{M} \to \mathcal{N}}(m) n_2 n\right), \tag{52}$$

where we used cyclicity of the trace and twice used (51).

Now we prove this $\mathrm{tr}_{\mathcal{M} \to \mathcal{N}}$ is the unique trace-preserving linear map satisfying the bi-module property. Suppose $\hat{\mathrm{tr}}_{\mathcal{M} \to \mathcal{N}}$ is another partial trace. Then for all density matrices $\rho \in \mathcal{M}$,

$$\mathrm{tr}_{\mathcal{N}}\left(\hat{\mathrm{tr}}_{\mathcal{M} \to \mathcal{N}}(\rho) n\right) = \mathrm{tr}_{\mathcal{N}}\left(\hat{\mathrm{tr}}_{\mathcal{M} \to \mathcal{N}}(\rho n)\right) = \mathrm{tr}_{\mathcal{M}}\left(\rho n\right), \tag{53}$$

where in the first equality we used the bi-module property and in the second we used the fact that $\hat{\mathrm{tr}}_{\mathcal{M} \to \mathcal{N}}$ is trace-preserving. We see that $\hat{\mathrm{tr}}_{\mathcal{M} \to \mathcal{N}}(\rho) = \mathrm{tr}_{\mathcal{M} \to \mathcal{N}}(\rho)$ for any density matrix $\rho$ and hence by linearity $\hat{\mathrm{tr}}_{\mathcal{M} \to \mathcal{N}} = \mathrm{tr}_{\mathcal{M} \to \mathcal{N}}$. □

---

[7]During the preparation of this manuscript, a proof of a (closely related) AEP for the max-relative entropy on *any* von Neumann algebra (including infinite-dimensional ones) was independently given in [33].

[8]For a definition of complete-positivity, see for example [35].

**Remark 2.39.** Note that this construction of a partial trace used the fact that all operators $m \in \mathcal{M}$ have a well-defined trace. Semifinite (infinite-dimensional) von Neumann algebras of Type $I_\infty$ and Type $II_\infty$, do not have this property.

**Theorem 2.40** (Strong subadditivity). *Let $\mathcal{M}_{A_0}$, $\mathcal{M}_{A_1}$, $\mathcal{M}_{B_0}$, and $\mathcal{M}_{B_1}$ be von Neumann algebras, each with a trace, acting on $\mathcal{H}$ with the following inclusion structure: $\mathcal{M}_{A_0} \supset \mathcal{M}_{B_0} \supset \mathcal{M}_{B_1}$ and $\mathcal{M}_{A_0} \supset \mathcal{M}_{A_1} \supset \mathcal{M}_{B_1}$. Finally, let the partial trace $\mathrm{tr}_{B_0 \to B_1} : \mathcal{M}_{B_0} \to \mathcal{M}_{B_1}$ be no less than the restriction to $\mathcal{M}_{B_0}$ of the partial trace $\mathrm{tr}_{A_0 \to A_1} : \mathcal{M}_{A_0} \to \mathcal{M}_{A_1}$, i.e. $\mathrm{tr}_{B_0 \to B_1} \geq \mathrm{tr}_{A_0 \to A_1}|_{B_0}$. Then for $\varepsilon > 0$*

$$H_{\min}^\varepsilon(A_0|B_0) \leq H_{\min}^\varepsilon(A_1|B_1), \tag{54}$$

$$S(A_0|B_0) \leq S(A_1|C_1), \tag{55}$$

$$H_{\max}^\varepsilon(A_0|B_0) \leq H_{\max}^\varepsilon(A_1|B_1). \tag{56}$$

*Proof.* See Appendix A.6. □

# 3 One-shot entropies for gravity

In this section we propose how to discuss the one-shot quantum Shannon theory of subregions in semiclassical gravity, specializing from the algebraic definitions of the previous section.

## 3.1 Definitions

Let $M$ be an (AdS-)globally hyperbolic Lorentzian spacetime with conformal boundary $M_\partial$ and let $J^\pm$ denote the causal future and past. Given any set $s \subset M$, $\partial s$ denotes the boundary of $s$ in $M$. The interior of $s$ is $s \setminus \partial s$ and is denoted $\mathrm{int}(s)$. For figures illustrating the following definitions, we refer the reader to Section 4.1 of [62].

**Definition 3.1.** The *spacelike complement* of a set $s \subset M$ is denoted $s'$, and is defined as the interior of the set of points that are spacelike related to all points in $s$,

$$s' := \mathrm{int}\left(M \setminus J^+(s) \setminus J^-(s)\right). \tag{57}$$

**Definition 3.2.** A *wedge* is a set $a \subset M$ that satisfies $a = a''$.

**Remark 3.3.** Wedges are open.

**Remark 3.4.** The intersection of two wedges can be shown to be a wedge. Similarly, the spacelike complement of a wedge is itself a wedge.

**Definition 3.5.** Given two wedges $a$ and $b$, the *wedge union* is defined as

$$a \uplus b := (a' \cap b')'. \tag{58}$$

By the above remark, $a \uplus b$ is a wedge.

**Definition 3.6.** The *edge* of wedge $a$ is defined as

$$\eth a := \partial a \cap \partial a'. \tag{59}$$

Conversely, a wedge is fully characterized by specifying its edge and one spatial side of that edge as the inside.

## 3.2 Generalized one-shot entropies

We take semiclassical gravity to mean quantum field theory (QFT) on a curved background, coupled to gravity with Newton's constant $G$ sufficiently small for perturbative approximations to be valid.

In regular QFT – without the coupling to gravity – the algebra $\mathcal{M}_b$ of operators associated to a wedge $b$ is generally of type III and density matrices do not exist. Nonetheless, one can regulate the theory, for example by introducing a lattice cutoff with spacing $\delta$. The von Neumann entropy $S(b)$ is then well defined in the regulated theory but diverges as the regulator is taken away, $\delta \to 0$, with the leading divergence proportional to the area $A(\eth b)$.

In semiclassical gravity the situation is expected to be better (see for example [22] and references therein, and [36–38] for relevant recent work). The physical entropy associated to a wedge is the generalized entropy

$$S_{\text{gen}}(b) = \frac{A(\eth b)}{4G} + S(b),\tag{60}$$

which is thought to be UV finite, the divergence in $S(b)$ cancelling against a counterterm in $A(\eth b)/4G$.[9]

In the same spirit, we conjecture that the min-entropy and max-entropy also admit UV finite "generalized" versions [1]. To introduce them, it will be helpful to UV regulate semiclassical gravity, say again by some $\delta$ such that $\delta \to 0$ removes the regulator. In this cutoff theory, the algebra $\mathcal{M}_b$ has a non-trivial center, generated by the observables measurable in both $b$ and its complement $b'$ [10]. In particular this includes geometric features of the surface $\eth b$, such as the operator $\hat{A}(\eth b)$ measuring the area of $\eth b$, which by Remark 2.16 takes the form

$$\hat{A}(\eth b) = \oplus_\alpha A_\alpha,\tag{61}$$

where $A_\alpha \in \mathbb{R}$ is the area of states in sector $\alpha$.

If the regulated algebras are finite-dimensional, we can also define canonical density matrices for the cutoff algebra $\mathcal{M}_b$. As discussed in Section 2.2, these take the form

$$\rho_{b,\text{can}} = \oplus_\alpha q_\alpha \rho_{b,\alpha},\tag{62}$$

where $q_\alpha$ is a probability distribution over $\alpha$ sectors and $\rho_{b,\alpha}$ is the normalized density matrix of the quantum fields in $b$ conditioned on the center observables being in sector $\alpha$.

Canonical density matrices are not regulator independent, however, and are not expected to have a nice limit as we take $\delta \to 0$. Instead, we focus on a trace which is expected to be UV finite.

**Definition 3.7** (Generalized trace)**.** The *generalized trace* is the canonical trace with an insertion of the exponential of the area operator,

$$\text{tr}_{b,\text{gen}}[\cdot] := \text{tr}_{b,\text{can}}\left[e^{\hat{A}(\eth b)/4G}(\cdot)\right].\tag{63}$$

We will sometimes drop the subscript $b$ when it is clear from context. Since $e^{\hat{A}(\eth b)/4G}$ is central in the algebra $\mathcal{M}_b$, $\text{tr}_{\text{gen}}$ is a trace, with coefficients $C_\alpha^b$ as defined in (27) given by

$$C_\alpha^b = e^{A_\alpha/4G}.\tag{64}$$

**Definition 3.8** (Generalized density matrices)**.** The *generalized density matrices* are

$$\rho_{b,\text{gen}} := e^{-\hat{A}(\eth b)/4G}\rho_{b,\text{can}}.\tag{65}$$

---

[9]Subleading divergences in $S(b)$ are expected to be renormalized by other geometric terms in the gravitational entropy [22,39].

The von Neumann entropy of a generalized density matrix is given by

$$S(\rho_{b,\text{gen}}) = -\langle \log \rho_{b,\text{gen}} \rangle = \langle \hat{A} \rangle /4G - \langle \log \rho_{b,\text{can}} \rangle = S_{\text{gen}}(b). \tag{66}$$

Since generalized entropy is strongly expected to be UV-finite and regulator independent, it is reasonable to expect that generalized density matrices – unlike canonical density matrices – are also regulator independent. Indeed, as we discuss in Section 6, the continuum algebra $\mathcal{M}_b$ describing a black hole in the strict $G \to 0$ limit is a Type II$_\infty$ von Neumann factor [36, 37]. As a result, the continuum algebra has a unique trace and hence unique density matrices (up to normalization); the ambiguity present in regulated descriptions where the algebras have centers vanishes. One can show that this trace indeed describes the $\delta \to 0$ limit of the generalized trace rather than e.g. the canonical trace.

With the definition of generalized traces and density matrices in hand, we can define conditional generalized one-shot entropies using the definitions given in Section 2.2.

**Definition 3.9** (Generalized conditional entropies). For any pair of wedges $a \subset b$, we define

$$H_{\text{min,gen}}(b|a)_\psi := -\min_\sigma \inf\{\lambda : \rho_{b,\text{gen}} \leq e^\lambda \sigma_{a,\text{gen}}\}, \tag{67}$$

$$S_{\text{gen}}(b|a)_\psi := S_{\text{gen}}(b)_\psi - S_{\text{gen}}(a)_\psi, \tag{68}$$

$$H_{\text{max,gen}}(b|a)_\psi := \sup_\sigma \log\left(\text{tr}_{b,\text{gen}} \sqrt{\sigma_{a,\text{gen}}^{1/2}\rho_{b,\text{gen}}\sigma_{a,\text{gen}}^{1/2}}\right)^2, \tag{69}$$

where $S_{\text{gen}}(x) = -\text{tr}_{\text{gen}}[\rho_{x,\text{gen}} \log \rho_{x,\text{gen}}]$.

**Remark 3.10.** After smoothing, these define the *smooth conditional generalized entropies*.

**Remark 3.11.** For notational convenience, we will sometimes define generalized entropies for sets $s$ that are not a wedge. In this case, $S_{\text{gen}}(s) := S_{\text{gen}}(s'')$.

Of these three quantities, the difference in generalized entropies $S_{\text{gen}}(b|a)$ is the most familiar, with a straightforward physical interpretation:

$$S_{\text{gen}}(b|a) = \frac{\langle A(\eth b) \rangle - \langle A(\eth a) \rangle}{4G} + S(b) - S(a), \tag{70}$$

where $\langle A(\eth b) \rangle, \langle A(\eth a) \rangle$ are the expectation value of area for the edges of regions $b$ and $a$ respectively.

What about the (smooth) generalized one-shot entropies? Consider the unconditional generalized min-entropy,

$$H_{\text{min,gen}}(b) = -\inf\left\{\lambda : e^{-\hat{A}(\eth b)/4G}\rho_{b,\text{can}} \leq e^\lambda\right\}. \tag{71}$$

This equals $H_{\text{min,gen}}(b) = -\log \lambda_{\text{largest}}$, where $\lambda_{\text{largest}}$ is the largest eigenvalue of the operator $e^{-\hat{A}(\partial b)/4G}\rho_b$. In other words, while the generalized von Neumann entropy is the expectation value of $\hat{A}/4G - \log\rho$, the generalized min-entropy is the minimal possible value for the operator $\hat{A}/4G - \log\rho$. The smooth generalized min-entropy is closely related: it is a lower confidence bound on $\hat{A}/4G - \log\rho$.

The unconditional generalized max-entropy

$$H_{\text{max,gen}}(b) = 2\log\left(\text{tr}_{\text{gen}}\rho_{b,\text{gen}}^{1/2}\right), \tag{72}$$

is the Rényi-1/2 entropy of the density matrix $e^{-\hat{A}(\eth b)/4G}\rho_{b,\text{can}}$ with respect to the generalized trace. Just like ordinary Rényi-1/2 entropies, it is typically dominated by the many small

eigenvalues of $e^{-\hat{A}(\eth b)/4G}\rho_{b,\text{can}}$. As a result, the smooth generalized max-entropy is an upper confidence bound on $\hat{A}/4G - \log\rho$.

As emphasized in Section 2.1, conditional one-shot entropies cannot generally be written as differences between unconditional entropies. Instead they are best understood operationally; see e.g. [27]. However there exist interesting classes of states [1] for which (regulated) bulk smooth min-, von Neumann, and smooth max-entropies all differ at $O(1/G)$ while fluctuations in areas are $O(1/\sqrt{G})$. In that case, we can treat the area terms in Definition 3.9 as c-numbers at leading order. We then obtain

$$H^{\varepsilon}_{\text{min/max,gen}}(b|a)_{\psi} \approx H^{\varepsilon}_{\text{min/max,can}}(b|a)_{\psi} + \frac{A(\eth b) - A(\eth a)}{4G}, \tag{73}$$

where $A(\eth b)$ and $A(\eth a)$ are the classical areas of the respective surfaces.

We emphasize however that this approximation only makes sense if $H^{\varepsilon}_{\text{min/max,can}}$ is explicitly regulated. While the leading divergence in $H^{\varepsilon}_{\text{min/max,can}}$ as $\delta \to 0$ is proportional to $A(\eth b) - A(\eth a)$ as for the conditional von Neumann entropy, the subleading divergences will be different.[10] As a result, $H^{\varepsilon}_{\text{min/max,can}}$ cannot be rendered UV-finite by the addition of the same area difference that works for the conditional von Neumann entropy. On the other hand we do expect Definition 3.9 to be genuinely UV-finite. We provide some evidence for this in Section 6 where we show how to define certain examples of finite conditional generalized one-shot entropies in the continuum $G \to 0$ theory.

We conclude this section by noting two important properties of generalized one-shot entropies that are inherited from the corresponding properties of general algebraic one-shot entropies from Section 2.2.

**Proposition 3.12** (Duality). Let $a \subset b$ be wedges and let $a'$, $b'$ be their complements. Then for any pure state $|\psi\rangle \in \mathcal{H}$ and $\varepsilon \geq 0$, it holds that

$$H^{\varepsilon}_{\text{min,gen}}(b|a)_{\psi} = -H^{\varepsilon}_{\text{max,gen}}(a'|b')_{\psi}. \tag{74}$$

*Proof.* Since each pair of complementary regions shares a common edge $\eth b = \eth b'$ and $\eth a = \eth a'$, we have

$$C^{b}_{\alpha} = C^{b'}_{\alpha} = e^{A_{\alpha}(\eth b)/4G}, \tag{75}$$

with a similar equality holding for $a$ and $a'$. The generalized traces on $b$ and $b'$ (and $a$ and $a'$) are therefore complementary in the sense of Definition 2.22. Consequently, the result follows from Theorem 2.33. $\square$

**Proposition 3.13** (Strong subadditivity). Let $a \supseteq b, c$ be bulk subregions with $a \cap b' \subseteq c$. Then

$$H^{\varepsilon}_{\text{min,gen}}(a|c) \leq H^{\varepsilon}_{\text{min,gen}}(b|c \cap b), \tag{76}$$

$$S_{\text{gen}}(a|c) \leq S_{\text{gen}}(b|c \cap b), \tag{77}$$

$$H^{\varepsilon}_{\text{max,gen}}(a|c) \leq H^{\varepsilon}_{\text{max,gen}}(b|c \cap b). \tag{78}$$

*Proof.* Note that we have the inclusion structure $\mathcal{M}_a \supseteq \mathcal{M}_c \supseteq \mathcal{M}_{b\cap c}$ and $\mathcal{M}_a \supseteq \mathcal{M}_b \supseteq \mathcal{M}_{b\cap c}$. According to Theorem 2.40, we then just need to verify that $\text{tr}_{a\to c,\text{gen}}|_b \leq \text{tr}_{b\to b\cap c,\text{gen}}$. Consider a general operator $O_b \in \mathcal{M}_b$. By definition,

$$\text{tr}_{a\to c,\text{gen}}[O_b] = e^{-\hat{A}(\eth c)/4G} \text{tr}_{a\to c,\text{can}}\left[e^{\hat{A}(\eth a)/4G} O_b\right], \tag{79}$$

$$\text{tr}_{b\to b\cap c,\text{gen}}[O_b] = e^{-\hat{A}(\eth(b\cap c))/4G} \text{tr}_{b\to b\cap c,\text{can}}\left[e^{\hat{A}(\eth b)/4G} O_b\right]. \tag{80}$$

---

[10]UV-divergences in QFT entanglement entropies come from UV Rindler-like modes near the edges of regions. The leading divergence is linear in the number of such modes $n$ that are below the UV-cutoff. Thanks to the asymptotic equipartition principle, this $O(n)$ divergence is the same for both one-shot and von Neumann entropies. However there will be subleading $O(\sqrt{n})$ differences between them that will still diverge as $n \to \infty$.

Because $\mathcal{M}_a \supset \mathcal{M}_b$, then $\hat{A}(\eth a)$ commutes with $\hat{A}(\eth b)$ and so we can write the exponential for the area operator for $a$ as

$$e^{\hat{A}(\eth a)/4G} = e^{\hat{A}(\eth b)/4G} e^{\hat{A}(\eth a)/4G - \hat{A}(\eth b)/4G} \, . \tag{81}$$

By the assumption that $a \cap b' \subseteq c$, we further know that $\mathcal{M}_a \cap \mathcal{M}_b' \subseteq \mathcal{M}_c$. Since $\hat{A}(\eth a), \hat{A}(\eth b) \in \mathcal{M}_a \cap \mathcal{M}_b'$, then also $\hat{A}(\eth a) - \hat{A}(\eth b) \in \mathcal{M}_c$. Using the bi-module property, we can then pull $e^{\hat{A}(\eth a)/4G - \hat{A}(\eth b)/4G}$ out of the partial trace so that

$$\mathrm{tr}_{a \to c, \mathrm{gen}}[O_b] = e^{\frac{1}{4G}\left(-\hat{A}(\eth c) + \hat{A}(\eth a) - \hat{A}(\eth b)\right)} \mathrm{tr}_{a \to c, \mathrm{can}}\left[e^{\hat{A}(\eth b)/4G} O_b\right] . \tag{82}$$

If we use the fact that in a local (regulated) quantum field theory, the restriction of $\mathrm{tr}_{a \to c, \mathrm{can}}$ to $\mathcal{M}_b$ is simply $\mathrm{tr}_{b \to (b \cap c), \mathrm{can}}$, then the necessary inequality holds if we can prove the following inequality on areas

$$-A(\eth c) + A(\eth a) - A(\eth b) \le -A(\eth(b \cap c)), \tag{83}$$

but this is just the statement of strong sub-additivity for areas, a true fact about geometric area. $\qquad\square$

## 4 One-shot quantum expansion and focusing conjectures

The goal of this section is to define new, one-shot versions of ideas that have been important in the study of quantum gravity: min- and max-quantum expansions and min- and max-quantum focusing conjectures (QFC). While also of intrinsic interest themselves, these will play vital roles in Section 5, helping us define and prove theorems about covariant min- and max-entanglement wedges.

### 4.1 Min- and max-quantum expansions

Given a wedge $a$, there are two outwards-directed null hypersurfaces orthogonal to $\eth a$, one future-directed (past-directed) which we will call $N_+$ ($N_-$), forming part of the boundary of the causal future and past of $a$ respectively. Let $N$ denote either one. Through each point of $\eth a$ passes one generator of $N$. Let $\lambda$ be an affine parameter along this generator, such that $\lambda = 0$ on $\eth a$ and $\lambda$ increases away from $\eth a$. This defines a coordinate system $(\lambda, y)$ on $N$. A continuous function $V(y) \ge 0$ defines a slice of $N$, consisting of the point on each generator $y$ for which $\lambda = V$. Any such $V$ defines a new wedge $a(V)$ with $\eth a(V) = V$ and the inside chosen in the direction of decreasing $\lambda$.

A local deformation of wedge $a$ can be defined as follows. Consider $\eth a$ and a second slice of $N$ that differs from $\eth a$ only in a neighborhood of generators with infinitesimal area $\mathcal{A}$ around a generator $y_0$:

$$V_{\mathcal{A}, \delta, y_0}(y) := \delta f_{\mathcal{A}, y_0}(y). \tag{84}$$

Here $\delta \ge 0$ and we define $f_{\mathcal{A}, y_0} = 1$ in a neighborhood of area $\mathcal{A}$ around point $y_0$ and $f_{\mathcal{A}, y_0} = 0$ everywhere else (smoothed out to be appropriately continuous). See Figure 1.

**Definition 4.1** (von Neumann expansion)**.** Let $a$ be a wedge, let $y_0 \in \eth a$, and let $V^+$ ($V^-$) be associated to a future-directed (past-directed) outwards null hypersurface orthogonal to $\eth a$. The future (past) *von Neumann expansion* $\Theta_+[a, y_0]$ ($\Theta_-[a, y_0]$) is the derivative of the

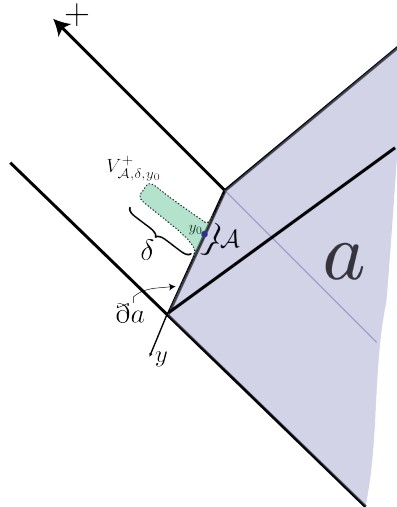

Figure 1: This figure depicts the deformation of a wedge. The undeformed wedge is $a$ with edge $\eth a$ drawn with the solid line. We deform the region $a$ by deforming $\eth a$ in the null direction by the bump $V_{\mathcal{A},\delta,y_0}^+$ at transverse coordinate $y_0$ with width $\mathcal{A}$ and height $\delta$. This takes $\eth a$ to the dashed line. The new, deformed wedge $a(V_{\mathcal{A},\delta,y_0}^+)$ then has edge given by $V_{\mathcal{A},\delta,y_0}^+$. Expansions are then defined via limits as $\mathcal{A},\delta \to 0$.

generalized entropy with respect to local deformation (84) along the future (past) null congruence:[11]

$$\Theta_{\pm}[a;y_0] := \lim_{\mathcal{A}\to 0} \lim_{\delta\to 0} \frac{4G}{\mathcal{A}\delta} S_{\text{gen}}\left(a\left(V_{\mathcal{A},\delta,y_0}^{\pm}\right)\Big|a\right). \tag{85}$$

**Remark 4.2.** An equivalent but perhaps more familiar definition is

$$\Theta_{\pm}[a;y_0] = \frac{4G}{\sqrt{h(y_0)}} \frac{\delta}{\delta V(y_0)} S_{\text{gen}}(a), \tag{86}$$

where $h$ is the induced area element on $\eth a$. We use (85) because it nicely generalizes to the one-shot expansions.

**Remark 4.3.** The von Neumann expansion can be decomposed as

$$\Theta_{\pm}[a;y_0] = \theta[a;y_0] + 4G \lim_{\mathcal{A}\to 0} \lim_{\delta\to 0} \frac{1}{\mathcal{A}\delta} S\left(a\left(V_{\mathcal{A},\delta,y_0}^{\pm}\right)\Big|a\right), \tag{87}$$

where $\theta$ is the classical expansion and $S(a(V)|a)$ is the conditional von Neumann entropy of $a(V)$ conditioned on $a$.

This von Neumann expansion is used in a number of conjectures, such as the generalized second law (GSL) and QFC, which we will review momentarily. We first construct the following one-shot versions of the quantum expansions.

**Definition 4.4** (One-shot expansions). Let $a$ be a wedge, let $y_0 \in \eth a$, and let $V^+$ ($V^-$) be associated to a future-directed (past-directed) outwards null hypersurface orthogonal to $\eth a$.

---

[11]$\Theta_{\pm}$ is often called the *quantum* expansion, to emphasize the use of generalized entropy instead of just the area. We use this new name to distinguish the use of generalized von Neumann entropy from the generalized one-shot entropies.

Let $\varepsilon > 0$. The future (past) *max-expansion* $\Theta_{+,\mathrm{max}}^{\varepsilon}[a, y_0]$ ($\Theta_{-,\mathrm{max}}^{\varepsilon}[a, y_0]$) is the smooth conditional generalized max-entropy associated to local deformation (84) along the future (past) null congruence:

$$\Theta_{\pm,\mathrm{max}}^{\varepsilon}[a; y_0] := \lim_{\mathcal{A}\to 0} \lim_{\delta\to 0} \frac{4G}{\mathcal{A}\delta} H_{\mathrm{max,gen}}^{\varepsilon}\left(a\left(V_{\mathcal{A},\delta,y_0}^{\pm}\right)|a\right). \tag{88}$$

The future (past) *min-expansion* $\Theta_{+,\mathrm{min}}^{\varepsilon}[a, y_0]$ ($\Theta_{-,\mathrm{min}}^{\varepsilon}[a, y_0]$) is the smooth conditional generalized min-entropy associated to local deformation (84) along the future (past) null congruence:

$$\Theta_{\pm,\mathrm{min}}^{\varepsilon}[a; y_0] := \lim_{\mathcal{A}\to 0} \lim_{\delta\to 0} \frac{4G}{\mathcal{A}\delta} H_{\mathrm{min,gen}}^{\varepsilon}\left(a\left(V_{\mathcal{A},\delta,y_0}^{\pm}\right)|a\right). \tag{89}$$

**Remark 4.5.** We shall assume that these limits are well defined and depend continuously on the wedges $a(V)$ for semiclassical states.

**Remark 4.6.** Unlike the von Neumann expansion, the one-shot expansions cannot in general be decomposed as in Remark 4.3, with one term pertaining to the area and a separate term to the one-shot entropy. Furthermore, the one-shot conditional generalized entropies, e.g. $H_{\mathrm{max,gen}}^{\varepsilon}(a(V)|a)$, cannot be written as a difference by Remark 2.4, and therefore under the limits they do not describe a standard derivative.

These min- and max-expansions inherit useful properties from the generalized min- and max-entropies. In the following we assume the global state is pure for simplicity, such that for example $S_{\mathrm{gen}}(a) = S_{\mathrm{gen}}(a')$. This can always be achieved by purifying the system with a reference $R$ and including $R \subset a'$ when $R \not\subset a$.

**Lemma 4.7** (Complementary expansions). *It holds that*

$$\Theta_{\pm,\mathrm{min}}^{\varepsilon}[a; y_0] = -\Theta_{\mp,\mathrm{max}}^{\varepsilon}[a'; y_0]. \tag{90}$$

*Proof.* Let $b := a(V_{\mathcal{A},\delta,y_0})$ denote a wedge defined by local deformation of $a$, for some finite $\mathcal{A}, \delta$. By Theorem 2.33, it holds that $H_{\mathrm{min,gen}}^{\varepsilon}(b|a) = -H_{\mathrm{max,gen}}^{\varepsilon}(a'|b')$. By Remark 4.5, this continues to hold in the limits $\mathcal{A}, \delta \to 0$. $\square$

**Lemma 4.8** (Ordering of expansions). *For sufficiently small $\varepsilon > 0$,*

$$\Theta_{\pm,\mathrm{min}}^{\varepsilon}[a; y_0] \le \Theta_{\pm}[a; y_0] \le \Theta_{\pm,\mathrm{max}}^{\varepsilon}[a; y_0]. \tag{91}$$

*Proof.* Let $b := a(V_{\mathcal{A},\delta,y_0})$ denote a wedge defined by local deformation of $a$, for some finite $\mathcal{A}, \delta$. By Theorem 2.32, for sufficiently small $\varepsilon$ it holds that $H_{\mathrm{min,gen}}^{\varepsilon}(b|a) \le S(b|a) \le H_{\mathrm{max,gen}}^{\varepsilon}(b|a)$. By Remark 4.5 this continues to hold in the limits $\mathcal{A}, \delta \to 0$. $\square$

**Lemma 4.9** (Strong subadditivity of expansions). *Let $a \subseteq b$ be wedges in $M$. Let $y_0 \in \eth a, \eth b$, and let there be a non-zero open ball $O \subset M$ containing $y_0$ such that $a \cap O = b \cap O$. Then*

$$\Theta_{\pm,\mathrm{min}}^{\varepsilon}[b; y_0] \le \Theta_{\pm,\mathrm{min}}^{\varepsilon}[a; y_0], \tag{92}$$

$$\Theta_{\pm}[b; y_0] \le \Theta_{\pm}[a; y_0], \tag{93}$$

$$\Theta_{\pm,\mathrm{max}}^{\varepsilon}[b; y_0] \le \Theta_{\pm,\mathrm{max}}^{\varepsilon}[a; y_0]. \tag{94}$$

*Proof.* By assumption, there exists a small enough $\mathcal{A}, \delta$ such that we can take $V_{\mathcal{A},\delta,y_0}$ from (84) to describe a deformation of both $b$ and $a$. Then, for any finite $\mathcal{A}, \delta$ smaller than that, we have $b(V_{\mathcal{A},\delta,y_0}) \supset b \supset a$ and $b(V_{\mathcal{A},\delta,y_0}) \supset a(V_{\mathcal{A},\delta,y_0}) \supset a$. Furthermore, by Proposition 3.13 the generalized conditional entropies satisfy strong subadditivity, Theorem 2.40. Therefore

$$H_{\mathrm{min,gen}}^{\varepsilon}(b(V_{\mathcal{A},\delta,y_0})|b) \le H_{\mathrm{min,gen}}^{\varepsilon}(a(V_{\mathcal{A},\delta,y_0})|a), \tag{95}$$

and similarly for $S_{\mathrm{gen}}$ and $H_{\mathrm{max,gen}}^{\varepsilon}$. This continues to hold in the limits $\mathcal{A}, \delta \to 0$ by Remark 4.5. $\square$

## 4.2 One-shot quantum focusing conjectures

**Definition 4.10** (Quantum focusing conjecture [22, 40])**.** Let $a$ be a wedge, and let $V_1$ and $V_2 \geq V_1$ each define a slice of the same outwards-directed null hypersurface orthogonal to $\eth a$. Let $\Theta$ be the von Neumann expansion associated to this null hypersurface. For all $y \in \eth a$ such that $V_2(y) > 0$ (i.e. $y \in \text{supp}\, V_2$), let $\Theta[a; y] \leq 0$. Then

$$S_{\text{gen}}(a(V_2)|a(V_1)) \leq 0 \,. \tag{96}$$

**Remark 4.11.** The above QFC is weaker than the original version defined in [22], and was first defined in [40] where it was called the *restricted* QFC.[12] We use it for three reasons: (1) While there are no proofs of the original QFC, there are settings where this (restricted) QFC can be derived [40]. (2) While weaker, it seems to be sufficient to obtain the desirable implications of the original QFC. (3) It generalizes nicely to a one-shot version.

**Conjecture 4.12** (Max-quantum focusing)**.** Let $a$ be a wedge, and let $V_1$ and $V_2 \geq V_1$ each define a slice of the same outwards-directed null hypersurface orthogonal to $\eth a$. Let $\varepsilon > 0$, and let $\Theta_{\text{max}}^{\varepsilon}$ be the max-expansion associated to this null hypersurface. For all $y \in \eth a$ such that $V_2(y) > 0$, let $\Theta_{\text{max}}^{\varepsilon}[a; y] \leq 0$. Then

$$H_{\text{max,gen}}^{\varepsilon}(a(V_2)|a(V_1)) \leq 0 \,. \tag{97}$$

**Conjecture 4.13** (Min-quantum focusing)**.** This conjecture takes the same form as Conjecture 4.12 but with min replacing max everywhere.

**Remark 4.14.** The min- and max-quantum focusing conjectures are not equivalent because the requirement $\Theta_{\text{max}}^{\varepsilon}[a; y] \leq 0$ at the *beginning* of a null congruence is dual to a condition on $\Theta_{\text{min}}^{\varepsilon}[a; y]$ at the *end* of a congruence.

One could instead conjecture the following stronger statement, analogous to the QFC of [22], that one could call the "unrestricted one-shot QFC":

$$\Theta_{\text{max}}^{\varepsilon}[a(V); p] \leq \Theta_{\text{max}}^{\varepsilon}[a; p] \,. \tag{98}$$

This is equivalent by Lemma 4.7 to the same statement with max replaced by min. It is easy to verify that (98) alone would therefore imply both Conjectures 4.12 and 4.13 (up to $\mathcal{O}(\log \varepsilon)$ corrections) using the chain rule. However since Conjectures 4.12 and 4.13 are sufficient for all our results, we will never assume (98).

**Proposition 4.15** ($\Theta_{\text{max/min}}^{\varepsilon}$ remains non-positive)**.** Let $a$ be a wedge, let $V$ define a slice of an outwards-directed null hypersurface orthogonal to $\eth a$, let $\varepsilon > 0$, and let $\Theta_{\text{max/min}}^{\varepsilon}$ be the max-expansion associated to this null hypersurface. Denote by $X_{\text{max/min}}$ the set of $y \in \eth a$ such that $\Theta_{\text{max/min}}^{\varepsilon}[a; y] \leq 0$, and denote by $Y_{\text{max/min}} \subseteq X_{\text{max/min}}$ the set of $y \in \eth a$ such that $V(y) > 0$. Then assuming Conjectures 4.12 and 4.13, it holds for all $y \in X_{\text{max/min}}$ that

$$\Theta_{\text{max/min}}^{\varepsilon}[a(V); y] \leq 0 \,. \tag{99}$$

*Proof.* Consider a local deformation (84) of $a(V)$ at a point $y_0 \in Y_{\text{max}}$,

$$\widetilde{V}_{\mathcal{A}, \delta, y_0}(y) := V_{\mathcal{A}, \delta, y_0}(y) + V(y) \,. \tag{100}$$

Because $V(y)$ is continuous, there are small enough $\mathcal{A}, \delta$ such that $\widetilde{V}_{\mathcal{A}, \delta, y_0}(y) - V(y) > 0$ only for $y \in Y$. Therefore, for sufficiently small $\mathcal{A}, \delta$, Conjecture 4.12 implies that

$$H_{\text{max,gen}}^{\varepsilon}\left(a\left(\widetilde{V}_{\mathcal{A}, \delta, y_0}\right) | a(V)\right) \leq 0 \,. \tag{101}$$

---

[12]Technically our QFC is different than the restricted QFC of [40] in the following sense. One could obtain our QFC from that restricted QFC by integrating it and using the assumption that generators which exit the null hypersurface do not increase $S_{\text{gen}}$.

By Remark 4.5 this continues to hold in the limits $\mathcal{A}, \delta \to 0$. The proof for the min-entropy works analogously. $\square$

**Proposition 4.16.** *The min-QFC implies the (restricted) QFC.*

*Proof.* Our strategy is to apply the min-QFC to many independent copies of the spacetime, then use the quantum asymptotic equipartition principle to relate the min-entropy of this replicated setup to the von Neumann entropy of the original setup.

Say we are given a spacetime $M$, a wedge $a \subset M$, and an outwards-directed null hypersurface $N$ orthogonal to $\eth a$. Let $\Theta$ and $\Theta^\varepsilon_{\min}$ be the von Neumann and min-expansion associated to $N$.

Consider $n$ copies of $M$, which we will denote $M_n$. Let $a_n$ denote the union of each copy of $a$ in $M_n$, which is itself a wedge in $M_n$. Finally, let $V_1$ and $V_2 \geq V_1$ be slices of $N$, and let $a_n(V_i)$ for $i \in \{1, 2\}$ denote the union of $a(V_i)$ over each copy in $M_n$.

Suppose that $\Theta[a; y] \leq 0$ for all $y \in \eth a$ such that $V_2(y) > 0$. Denote by $y_i$ the transverse position along $\eth a_n$ in the $i$th copy of the spacetime. By the fact that the generalized entropy of a tensor product of two states is the sum of the generalized entropy for each state, we find that

$$\Theta[a_n; y_i] = \Theta[a; y], \tag{102}$$

and so $\Theta[a_n; y_i] \leq 0$ for all $1 \leq i \leq n$. By Lemma 4.8, we then have that $\Theta^\varepsilon_{\min}[a_n; y_i] \leq 0$ for small enough $\varepsilon$. By the min-QFC applied to the replicated spacetime, we then have that

$$H^\varepsilon_{\min,\mathrm{gen}}(a_n(V_2) | a_n(V_1)) \leq 0, \tag{103}$$

for slices $V_2 \geq V_1$. By the quantum asymptotic equipartition principle, Theorem 2.34, as applied to the generalized conditional entropies, we see that

$$H^\varepsilon_{\min,\mathrm{gen}}(a_n(V_2) | a_n(V_1)) = n S_{\mathrm{gen}}(a(V_2) | a(V_1)) + \mathcal{O}(\sqrt{n}) \leq 0, \tag{104}$$

as we take $n \to \infty$. Therefore $S_{\mathrm{gen}}(a(V_2) | a(V_1)) \leq 0$ as we wanted to show. $\square$

**Remark 4.17.** (One-shot covariant entropy bound) The one-shot QFCs imply a one-shot covariant entropy bound (see [61] for the original). That is, for a wedge $a$, slice $V$, and $\varepsilon > 0$, if $V(y) > 0$ only for $y$ such that $\Theta^\varepsilon_{\max/\min,\mathrm{gen}}[a; y] \leq 0$, then

$$H^\varepsilon_{\max/\min,\mathrm{gen}}(a(V) | a) \leq 0. \tag{105}$$

**Proposition 4.18** (One-shot generalized second law)**.** *The one-shot QFCs imply a min- and max-GSL. Let $a_1, a_2$ be wedges such that $\eth a_1, \eth a_2$ are slices of a future (past) causal horizon, with $\eth a_2$ everywhere to the future (past) of $\eth a_1$, and $a_2 \subseteq a_1$. Let $\varepsilon > 0$. Then assuming the one-shot QFCs,*

$$H^\varepsilon_{\max/\min,\mathrm{gen}}(a_1 | a_2) \leq 0. \tag{106}$$

*Proof sketch.* Without loss of generality we restrict to future causal horizons. Let $\Sigma_\partial \subset M_\partial$ be a spacelike Cauchy slice for (a subregion of) the asymptotic boundary $M_\partial$. The boundary (in the bulk) of the past of $\Sigma_\partial$, $\partial J^-(\Sigma_\partial)$, forms a future causal horizon in the bulk. Now consider a wedge $\tilde{a}$ with edge $\eth \tilde{a} \subseteq \partial J^-(\Sigma_\partial)$, such that $\Theta^\varepsilon_{-,\max/\min}$ is the expansion of the causal horizon. For $\eth \tilde{a}$ sufficiently close to asymptotic infinity, $\Theta^\varepsilon_{-,\max/\min}$ will approach its classical value which is negative everywhere. The desired result for the causal horizon $\eth \tilde{a} \subseteq \partial J^-(\Sigma_\partial)$ then follows directly from the max-/min-QFC. To extend this result to all causal horizons in asymptotically-AdS spacetimes, we note that all such causal horizons can be approached uniformly at any finite affine parameter by $J^-(\Sigma_\partial^n)$ for a sequence of spacelike boundary Cauchy slices $\Sigma_\partial^n$, indexed by $n$. The result therefore follows from the special case above by assuming continuity of $H^\varepsilon_{\max/\min,\mathrm{gen}}(a_1 | a_2)$. $\square$

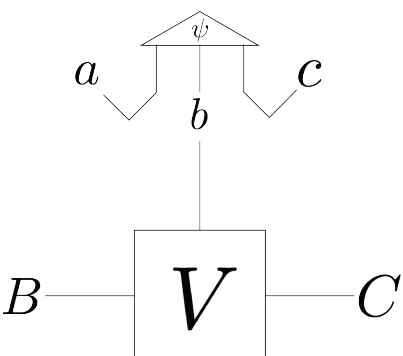

Figure 2: An illustration of $V$ as a tensor "network" composed of a single, random tensor from $b$ to outputs $B$ and $C$. We then feed the state $|\psi\rangle \in \mathcal{H}_a \otimes \mathcal{H}_b \otimes \mathcal{H}_c$ into this random tensor on $b$.

## 5 Covariant min- and max-entanglement wedges

We now turn to the central goal of this paper: proposing a fully covariant generalization of the min- and max-entanglement wedges (EW) of [1] that can be applied in arbitrary time-dependent spacetimes. We first review known results about one-shot quantum Shannon theory and information flow in tensor networks and gravity in Section 5.1. In Section 5.2, we then explain the intuition behind our proposal for the generalization of those results to arbitrary time-dependent spacetimes and give formal definitions of the min- and max-EWs. Finally, in Section 5.3, we show that the min- and max-EWs satisfy many desirable properties that support their conjectured operational interpretations.

### 5.1 State merging and gravity

Let $V : \mathcal{H}_b \to \mathcal{H}_B \otimes \mathcal{H}_C$ be a Haar random isometry[13] with output Hilbert space dimensions $d_B$ and $d_C$, as in Figure 2. Let $|\psi\rangle \in \mathcal{H}_a \otimes \mathcal{H}_b \otimes \mathcal{H}_c$ be an arbitrary state with reduced density matrix $\psi_c$ on $\mathcal{H}_c$. A standard fact from one-shot quantum Shannon theory [41] says that we have

$$\mathrm{tr}_{aB}[V |\psi\rangle\langle\psi| V^\dagger] \approx \frac{1}{d_C} \mathbb{1}_C \otimes \psi_c \,, \tag{107}$$

with high probability whenever

$$H_{\max}^\varepsilon(ab|a) + \log d_C - \log d_B \ll 0 \,. \tag{108}$$

Conversely, (107) never holds when

$$H_{\max}^\varepsilon(ab|a) + \log d_C - \log d_B \gg 0 \,. \tag{109}$$

A consequence is that one can do "state-specific reconstruction" [14] of operators in $\mathcal{H}_b$ from $\mathcal{H}_a \otimes \mathcal{H}_B$ for the state $|\psi\rangle$ if and only if (108) holds. By state-specific reconstruction, we mean that for any unitary $U_b$ there exists a unitary $U_{aB}$ on $\mathcal{H}_a \otimes \mathcal{H}_B$ such that

$$U_{aB}V |\psi\rangle \approx V U_b |\psi\rangle \,. \tag{110}$$

That such a $U_{aB}$ exists follows from (107) because $|\psi\rangle$ and $U_b |\psi\rangle$ have the same reduced density matrix on $\mathcal{H}_C \otimes \mathcal{H}_c$, and all purifications are related by a unitary on the purifying

---

[13]$V : \mathcal{H}_1 \to \mathcal{H}_2$ is a Haar random isometry if it can be written as $V = UV_0$, with $V_0 : \mathcal{H}_1 \to \mathcal{H}_2$ a fixed isometry and $U$ a Haar random unitary on $\mathcal{H}_2$.

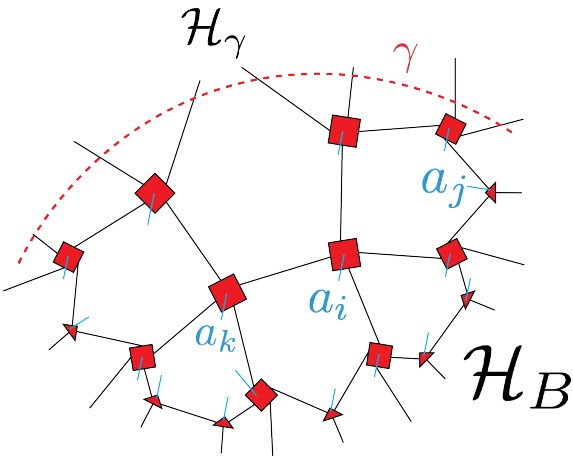

Figure 3: An illustration of a random tensor network described in the text. Each square or triangle represents a single random tensor with a dangling bulk leg (in blue), denoted by $a_i$, with local Hilbert space $\mathcal{H}_{a_i}$. The network maps the tensor product of $\mathcal{H}_{a_i}$ over all $i$ into the boundary Hilbert space $\mathcal{H}_B \otimes \mathcal{H}_\gamma$. In the analogy to AdS/CFT, we can think of $\mathcal{H}_B$ as being associated to some CFT subregion and $\mathcal{H}_\gamma$ as associated to degrees of freedom localized to the entangling surface of the bulk legs $\mathcal{H}_a = \otimes_i \mathcal{H}_{a_i}$.

system. From a quantum information perspective, the existence of $U_{aB}$ can be thought of as a Heisenberg-picture version of quantum state merging; giving access to $\mathcal{H}_B$ to an observer that controls $\mathcal{H}_a$ allows them to manipulate all information in $\mathcal{H}_b$.

The same inequalities applied to the complement, using the duality between min- and max-entropies, say that when

$$H^\varepsilon_{\min}(ab|a) = -H^\varepsilon_{\max}(bc|c) \gg \log d_B - \log d_C \,, \tag{111}$$

then

$$\text{tr}_{Cc}[V|\psi\rangle\langle\psi|V^\dagger] \approx \frac{1}{d_B}\mathbb{1}_B \otimes \psi_a \,, \tag{112}$$

and $\mathcal{H}_B$ alone carries no useful information about $b$. In the intermediate regime with

$$H^\varepsilon_{\min}(ab|a) \ll \log d_B - \log d_C \ll H^\varepsilon_{\max}(ab|a), \tag{113}$$

the Hilbert space $\mathcal{H}_B$ carries some but not all information in $\mathcal{H}_b$.

It was shown in [1] using Euclidean replica trick computations that a similar result holds in gravity, with $\log d_B$ and $\log d_C$ replaced by the areas of extremal surfaces. Specifically, when only two extremal surfaces, bounding wedges $b_1$ and $b_2 \supset b_1$ respectively, are relevant in replica trick computations, one finds that state-specific reconstruction of $b_2 \backslash b_1$ is possible if and only if

$$H^\varepsilon_{\max,\text{gen}}(b_2|b_1) \ll 0 \,, \tag{114}$$

while no information is accessible from $b_2 \backslash b_1$ if and only if

$$H^\varepsilon_{\min,\text{gen}}(b_2|b_1) \gg 0 \,. \tag{115}$$

In contrast, a naive application of the QES prescription would lead to (von Neumann) generalized entropies appearing in both (114) and (115).

In general, there is no reason that only two extremal surfaces can contribute in replica trick computations. So one would like a more general prescription. Suppose we have a random

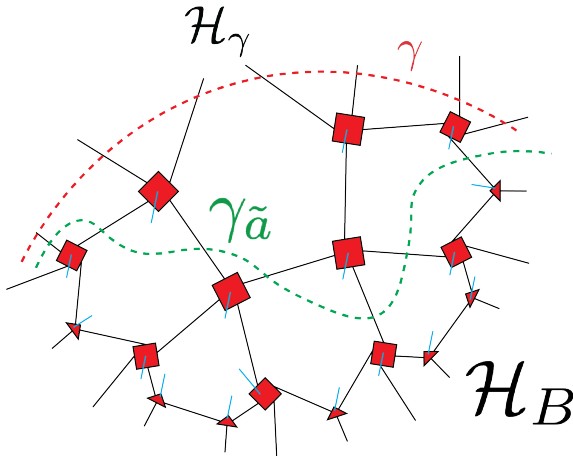

Figure 4: An illustration of a random tensor network as described in the text. This time we denote a candidate surface $\gamma_{\tilde{\mathbf{a}}}$ which bounds all the bulk sites $\tilde{\mathbf{a}}$ between $\gamma_{\tilde{\mathbf{a}}}$ and $B$. The dimension $\dim \gamma_{\tilde{\mathbf{a}}}$ is then the product of dimensions of the black legs cut by the dashed green line.

tensor network $V$ with bulk legs $a_1 \ldots a_n$ and boundary legs divided into $\mathcal{H}_B$ and $\mathcal{H}_\gamma$ as shown in Figure 3. Let $|\psi\rangle \in \bigotimes_i \mathcal{H}_{a_i} \otimes \mathcal{H}_r$ be an arbitrary state. It was shown in [42] (in somewhat different language) that with high probability

$$\mathrm{tr}_B[V\,|\psi\rangle\langle\psi|\,V^\dagger] \approx \frac{1}{d_\gamma}\mathbb{1}_\gamma \otimes \psi_r\,, \tag{116}$$

whenever

$$H_{\max}(a_1\ldots a_n|\tilde{\mathbf{a}}) - \log d_{\gamma_{\tilde{\mathbf{a}}}} + \log d_\gamma \ll 0\,, \tag{117}$$

for *all* subsets $\tilde{\mathbf{a}} \subset \{a_1\ldots a_n\}$. Here $d_\gamma$ is the dimension of $\mathcal{H}_\gamma$ and $d_{\gamma_{\tilde{\mathbf{a}}}}$ is the dimension of the cut $\gamma_{\tilde{\mathbf{a}}}$ bounding $\tilde{\mathbf{a}}$ and $B$, as shown in Figure 4. The authors of [42] conjectured that this continues to be true if the max-entropies in (117) are replaced by smooth max-entropies, so that

$$\forall\, \tilde{\mathbf{a}} \subseteq \{a_1...a_n\}, \quad H^\varepsilon_{\max}(a_1\ldots a_n|\tilde{\mathbf{a}}) - \log d_{\gamma_{\tilde{\mathbf{a}}}} + \log d_\gamma \ll 0\,. \tag{118}$$

This conjecture was recently proved in [43]. Conversely, the results of [41] show that (116) is never true if

$$\exists\, \tilde{\mathbf{a}} \subseteq \{a_1...a_n\}, \quad H^\varepsilon_{\max}(a_1\ldots a_n|\tilde{\mathbf{a}}) - \log d_{\gamma_{\tilde{\mathbf{a}}}} + \log d_\gamma \gg 0\,. \tag{119}$$

So (118) is optimal. It follows from (116) that any unitary $U_a$ on $\bigotimes_i \mathcal{H}_{a_i}$ that preserves (118) can be state-specifically reconstructed on $\mathcal{H}_B$.

For most tensor networks, (118) will not be satisfied if $a_1...a_n$ is the entire set of bulk sites. However, you can use the chain rule to show that there always exists a unique largest subset $\mathbf{a}_{\max} \subseteq \{a_1...a_n\}$ of bulk legs[14] such that (118) holds. This is the "max-EW" of the tensor network; it is the largest region $\mathbf{a}_{\max}$ such that state-specific reconstruction of everything in $\mathbf{a}_{\max}$ is possible [1]. (See Appendix B or [14] for a precise definition of what state-specific reconstruction means in this context.) Similarly there is a smallest region $\mathbf{a}_{\min}$ such that the part of the tensor network *outside* $\mathbf{a}_{\min}$ satisfies (118) for the complement $B'$ and so no information from outside $\mathbf{a}_{\min}$ can ever reach $B$. This is the "min-EW" of the tensor network; it is the bulk

---

[14]By "largest" we mean a subset that contains all other subsets satisfying the same property.

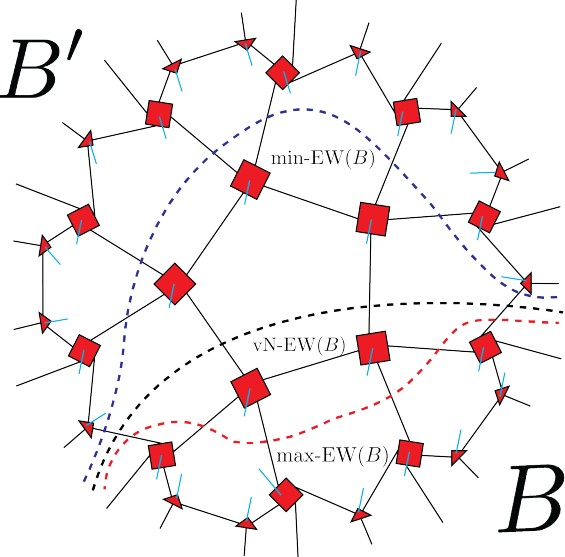

Figure 5: A tensor network with the regions max-EW[$B$] and min-EW[$B$] labeled. As discussed in the main text, the max-EW is conjectured to be the largest bulk region that can be state-specifically reconstructed from $B$. The min-EW is the bulk region whose state possibly affects the state of $B$. The vN-EW, which we discuss in the next subsection, is bounded by the minimal generalized entropy surface. The vN-EW lies between the min- and max-EWs.

complement of the max-EW for the complementary boundary region $B'$. The max-EW and min-EW are illustrated in Figure 5.

In [1], analogous results were conjectured to hold for time-reflection symmetric states in gravity.[15] The max-EW was defined as the largest wedge $b_1$ with edge in the time-reflection symmetric time slice such that

$$H_{\text{max,gen}}^{\varepsilon}(b_1|b_2) \ll 0, \tag{120}$$

for any smaller wedge $b_2 \subset b_1$ with edge in that slice. It was conjectured to be the largest wedge for which state-specific reconstruction is possible. Similarly, the min-EW was defined as the smallest time-reflection symmetric wedge $b_1$ such that any larger time-reflection symmetric wedge $b_2 \supset b_1$ has

$$H_{\text{min,gen}}^{\varepsilon}(b_2|b_1) \gg 0. \tag{121}$$

By duality, the min-EW of $B$ is the complement of the max-EW of $B'$. It follows from the conjectured properties of the max-EW that no information outside the min-EW is present in $B$.

It is worth noting that the discussion in [1] treated the algebra associated to a bulk region $b$ as tensor product factor, ignoring the existence of central operators such as $A(ðb)$. In fact, until now no precise definition of state-specific reconstruction for algebras with centers has appeared in the literature. We rectify this deficiency in Appendix B.

## 5.2 Definitions

The primary goal of the present paper is to extend the definitions of the max- and min-EW from [1] to general time-dependent spacetimes while preserving the conjectured operational interpretations described above.

---

[15]The paper [42] was not actually cited in [1] because of an embarrassing failure of one of the authors' knowledge of his own PhD advisor's prior work on the subject.

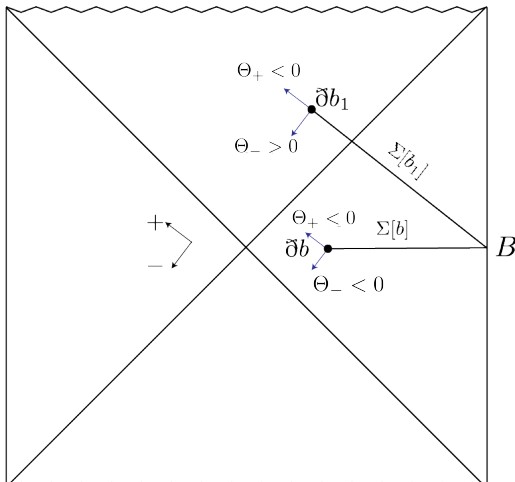

Figure 6: An illustration why an anti-normality condition is needed. Consider a BTZ black hole and let $\eth b_1$ be a trapped surface in the black hole interior. It is easy to find a Cauchy slice $\Sigma[b_1]$ for $b_1$ such that all sub-wedges $b_2 \subset b_1$ with $\eth b_2 \subset \Sigma[b_1]$ have $A(\eth b_2) > A(\eth b_1)$. However, one can send signals to $b_1$ from the left boundary, and hence it cannot be reconstructible from $B$. On the other hand, imposing an anti-normality condition ensures that $\eth b$ lies in the right black hole exterior.

Before giving a formal definition of our proposal, it is helpful to discuss the intuition behind it. (We focus on the max-EW case since the min-EW is directly related by duality.) The most naive generalization of (120) to arbitrary spacetimes would be to simply remove the requirement that $b_1$ and $b_2$ be time-reflection symmetric. In other words, we would require

$$H^\varepsilon_{\max,\text{gen}}(b_1|b_2) \ll 0, \tag{122}$$

for *any* wedge $b_2 \subseteq b_1$. But this is too strong! In the strict classical limit, we have

$$4G H^\varepsilon_{\max,\text{gen}}(b_1|b_2) \to A(\eth b_1) - A(\eth b_2). \tag{123}$$

If the area $A(\eth b_1) > 0$, then this will always be positive for some $b_2$ because we can choose the edge of $b_2$ to be piecewise lightlike.

A slightly more sophisticated guess would be to require (122) only for all wedges $b_2$ whose edge lies within one particular Cauchy slice $\Sigma[b_1]$ for $b_1$. This condition is at least achievable since one can choose $\Sigma[b_1]$ to exclude wedges with a piecewise-lightlike edge. However, it turns out to have the opposite problem of being too easily satisfied. Let us again consider the strict classical limit. As shown in Figure 6, one can easily find a wedge $b_1$ and Cauchy slice $\Sigma[b_1]$ such that $A(\eth b_2) > A(\eth b_1)$ for all wedges $b_2$ with edge $\eth b_2 \in \Sigma[b_1]$ even though $b_1$ is not reconstructible by its conformal boundary.

The fact that the proposal above is too weak suggests we need an additional condition on the wedge $b_1$. An answer that seems to work is to require $b_1$ to be max-antinormal, defined below to mean that both outgoing max-expansions are everywhere negative. This rules out, for example, the problematic wedge in Figure 6.

The previous discussion will straightforwardly lead to our proposed definition of the max-EW. However, since one-shot entropies may not be very familiar to the reader, it will be illuminating to first reformulate the standard QES prescription in terms of conditional von Neumann entropies in a similar manner, before turning to a formal definition of the max-EW.

**Definition 5.1** (vN-normal & vN-antinormal)**.** A wedge $b$ is called *vN-normal* (respectively *vN-antinormal*) if $\Theta_\pm[b; p] \geq 0$ (respectively $\Theta_\pm[b; p] \leq 0$) for all $p \in \eth b$.

**Definition 5.2** (vN-accessible)**.** Given a wedge $B \subset M_\partial$, a wedge $b_1 \subset M$ is said to be *vN-accessible for B* if $b_1 \cap M_\partial = B$, it is vN-antinormal, and it has a Cauchy slice $\Sigma[b_1]$ such that for all wedges $b_2 \subset b_1$ with edge $\eth b_2 \in \Sigma[b_1]$ and $B \subset b_2$,

$$S_{\text{gen}}(b_1|b_2) = S_{\text{gen}}(b_1) - S_{\text{gen}}(b_2) < 0. \tag{124}$$

**Definition 5.3** (vN-entanglement wedge)**.** Given a wedge $B \subseteq M_\partial$ and a state $|\psi\rangle$, let $F(B)$ be the set of wedges in $M$ that are vN-accessible for $B$. The von Neumann-entanglement wedge is the wedge union over all wedges in $F(B)$:

$$\text{vN-EW}[B] = \mathbb{U}_{b \in F(B)} b. \tag{125}$$

**Remark 5.4.** We will eventually show in Theorem 5.28 that the vN-EW is itself vN-accessible, and therefore is the unique largest vN-accessible wedge. We will also show in Theorem 5.23 that the vN-EW is bounded by the minimal generalized entropy quantum extremal surface, in accordance with the usual QES prescription.

The definition of max-EW is almost identical to the vN-EW, except with conditional generalized entropies replaced by $\varepsilon$-smooth conditional max-generalized entropies.

**Convention 5.5.** In all the definitions below we have $0 < \varepsilon \ll 1$ and $-\log \varepsilon \ll K \ll O(1/G)$ unless otherwise stated.

Intuitively, the parameter $\varepsilon$ will capture the accuracy with which reconstruction is possible. Note that $\varepsilon$ may be perturbatively small in $G$, but cannot be exponentially small without rendering the bounds on $K$ inconsistent. This is related to the fact that entanglement wedge reconstruction always has nonperturbative corrections from subleading saddle point contributions [12].

The parameter $K$ will describe how close the max-EW is allowed to be to a phase transition that would make it smaller. It has long been understood (see e.g. [1, 12, 44]) that the entanglement wedge is not sharply defined unless the difference between the generalized entropy of the QES region and that of any nonminimal QES region is much larger than $O(1)$. The parameter $K$ characterizes how sharply defined it is.

**Definition 5.6** (max-normal & max-antinormal)**.** A wedge $b$ is called $\varepsilon$ *max-normal* if

$$\Theta^\varepsilon_{\pm,\text{max}}[b; p] \geq 0, \tag{126}$$

and $\varepsilon$ *max-antinormal* if

$$\Theta^\varepsilon_{\pm,\text{max}}[b; p] \leq 0, \tag{127}$$

for all $p \in \eth b$.

**Definition 5.7** (max-accessible)**.** Given a wedge $B \subseteq M_\partial$ and a state $|\psi\rangle$, a wedge $b_1$ is said to be $(\varepsilon, K)$ *max-accessible for B* if (1) $b_1 \cap M_\partial = B$, (2) it is $\varepsilon$ max-antinormal, and (3) there exists a Cauchy slice $\Sigma[b_1]$ such that for all macroscopically distinct wedges $b_2 \subset b_1$ with edge $\eth b_2 \subset \Sigma[b_1]$ and $B \subset b_2$,

$$H^\varepsilon_{\text{max,gen}}(b_1|b_2) \leq -K. \tag{128}$$

The phrase "macroscopically distinct" here needs some clarification. Clearly, if $b_2 = b_1$, then $H^\varepsilon_{\text{max,gen}}(b_1|b_2) = 0$ and (128) is not satisfied for $K > 0$. But if $H^\varepsilon_{\text{max,gen}}(b_1|b_2)$ is a continuous function of $b_2$ then presumably you can also always violate (128) by making $b_2$ be sufficiently close to $b_1$. However, since $K \ll O(1/G)$ doing so will generally require $b_2$ to be perturbatively close to $b_1$ in the limit $G \to 0$. In order to avoid issues with Planckian perturbations, by macroscopically distinct, we mean that the difference between $b_2$ and $b_1$ is at least comparable in size to the smallest scale allowed in the bulk effective field theory.

**Definition 5.8** (max-entanglement wedge). Given a boundary region $B$ and a state $|\psi\rangle$, let $G_{(\varepsilon,K)}(B)$ be the set of wedges in $M$ that are $(\varepsilon,K)$ max-accessible for $B$. The $(\varepsilon,K)$ max-entanglement wedge of a boundary region $B$ is the wedge union over all wedges in $G_{(\varepsilon,K)}(B)$:

$$\text{max-EW}_{(\varepsilon,K)}[B] = \mathbb{U}_{b\in G_{(\varepsilon,K)}(B)}\, b\,. \tag{129}$$

**Remark 5.9.** As we show in Theorem 5.28, the max-EW is itself $(\varepsilon',K)$ max-accessible with $\varepsilon' = O(\varepsilon)$. In this sense it is therefore the unique largest max-accessible wedge.

**Remark 5.10.** The $(\varepsilon,K)$ max-EW monotonically increases in size when increasing $\varepsilon$ at fixed $K$ and monotonically decreases in size when increasing $K$ at fixed $\varepsilon$.

**Conjecture 5.11.** Consider a wedge $B \subseteq M_\partial$ and a state $|\psi\rangle$. The $(\varepsilon,K)$ max-EW of $B$ with $K \gg -\log\varepsilon$ for $|\psi\rangle$ can be state-specifically reconstructed from $B$ with error at most $\varepsilon^{O(1)}$. Conversely, for $K \ll \log\varepsilon$ no region $b$ outside the $(\varepsilon,K)$ max-EW of $B$ can be state-specifically reconstructed from $B$ with error smaller than $\varepsilon^{O(1)}$.

**Remark 5.12.** We define state-specific reconstruction formally for algebras in appendix B.

The min-EW is the complement of the max-EW of the complement.[16]

**Definition 5.13** (min-entanglement wedge). Given a state $|\psi\rangle$, the $(\varepsilon,K)$ min-entanglement wedge of a boundary subregion $B$ is the spacelike complement of max-EW$_{(\varepsilon,K)}[B']$,

$$\text{min-EW}_{(\varepsilon,K)}[B] = \left(\text{max-EW}_{(\varepsilon,K)}[B']\right)'\,. \tag{130}$$

**Remark 5.14.** By duality (Theorem 2.33) the min-EW could also be defined directly as the intersection of all min-normal wedges $b$ where there exists a Cauchy slice $\Sigma[b']$ for wedge $b'$ such that $H^\varepsilon_{\text{min,gen}}(a|b) > K$ for all macroscopically distinct $a \supset b$ with $\eth a \in \Sigma[b']$ and $a \cap M_\partial = B$.

**Remark 5.15.** An immediate consequence of Conjecture 5.11 is that no information from outside the min-EW can affect the state of $B$ by more than an $\varepsilon^{O(1)}$-amount.

## 5.3 Properties

We now prove properties about the min-EW, max-EW, and vN-EW. These properties are consistency conditions which corroborate Conjecture 5.11. We will assume throughout that the max-QFC and (von Neumann) QFC both hold.[17]

Let us motivate these consistency conditions. The first is that in certain cases, the max-EW and min-EW should coincide, and in such cases should equal the QES region. Indeed for special "compressible" states, the QES region is believed to satisfy the conditions in Conjecture 5.11 for both the max-EW and min-EW [1,9,10,14].

The second consistency condition is that the max-EW should be contained inside the min-EW. This follows from a well-known principle in quantum information theory called the information-disturbance trade-off, which says that a system $B$ fully encodes some quantum information if and only if the complementary subsystem $B'$ knows nothing about it (see e.g. [45]).[18] If Conjecture 5.11 is right, then the max-EW of $B$ and $B'$ cannot overlap.

---

[16]We continue to assume the global state is pure for simplicity, such that $S_{\text{gen}}(a) = S_{\text{gen}}(a')$. Again, this can always be achieved by purifying the system with a reference $R$ and including $R \subset a'$ when $R \not\subset a$.

[17]We could alternatively assume the max-QFC and min-QFC since the latter implies the von Neumann QFC by Proposition 4.16, or we could assume the unrestricted one-shot QFC from Remark 4.14, which implies both the max- and min-QFCs.

[18]The famous quantum no-cloning and no-erasure theorems can be thought of as examples of this principle.

The third consistency condition is that the max-EW contains subregions of the bulk that we know $B$ can reconstruct. For example, the max-EW[$B$] should include the causal wedge of $B$, which we know is reconstructible via the HKLL protocol [46,47]. Finally, the max-EW should also nest, which means it includes the max-EW of smaller regions: if $B \supseteq A$, then max-EW[$B$] $\supseteq$ max-EW[$A$].

Throughout this section we take $M_\partial$ to be the conformal boundary of $M$, and we will assume the following generic condition on $M$:

**Definition 5.16.** The *generic condition* is an assumption that all inequalities involving generalized conditional entropies apply strictly at some scale $\kappa$. For example, the max-QFC states that $H_{\text{max,gen}}(a(V_2)|a(V_1)) \leq 0$ for $V_2 \geq V_1 \geq 0$ slices of some outward null congruence emanating from a wedge $a$ with non-positive initial max-expansion. The generic condition assumes the stronger condition that instead

$$H_{\text{max,gen}}(a(V_2)|a(V_1)) \ll -\kappa \,. \tag{131}$$

It is often assumed that the scale of $\kappa$ is leading order ($\kappa = O(\ell^{d-2}/G)$ with $\ell$ a characteristic scale in the state). However, in our case it will be acceptable for $\kappa$ to be much smaller than this, so long as $\kappa \gg K$.

**Definition 5.17** (Causal wedge [48]). Given a wedge $B \subseteq M_\partial$, the *causal wedge* of $B$ is $C[B] := J^+[B] \cap J^-[B]$.

**Lemma 5.18.** *Given a wedge $B \subseteq M_\partial$ with complement $B'$ in $M_\partial$, assuming the QFC then its causal wedge $C[B]$ is spacelike to $B'$.*

*Proof.* The QFC implies the GSL which implies $C[B] \cap J^\pm[B'] = \varnothing$ [49]. Note this also follows from the Gao-Wald theorem, which requires only the weaker condition that the achronal average null energy condition holds [50]. □

**Lemma 5.19.** *Assuming the QFC, the causal wedge $C[B]$ of a boundary wedge $B$ is vN-accessible. Assuming the max-QFC and the generic condition, it is $(\varepsilon, K)$ max-accessible for any $\varepsilon > 0$ and $K \ll \kappa$. Moreover, in both cases, given any Cauchy slice $\Sigma[B]$ for $B$, we can always choose the Cauchy slice $\Sigma[C[B]]$ to have $\Sigma[B]$ as its conformal boundary.*

*Proof.* The boundary of the causal wedge $C[B]$ is the union of portions of past and future causal horizons, denoted $\mathscr{I}^-(C[B])$ and $\mathscr{I}^+(C[B])$ respectively. By the GSL and the max-GSL, $C[B]$ is therefore vN-antinormal and max-antinormal. (Note that $C[B] \cap M_\partial = B$ by lemma 5.18.)

To finish the proof, we now want to construct a Cauchy slice $\Sigma[C[B]]$ with conformal boundary $\Sigma[B]$ satisfying the appriopriate conditions. Define $\beta^+ := (\partial J^+(\Sigma[B]) \cap C[B])$ to be the co-dimension one region which is given by the portion of the future light sheet from $\Sigma[B]$ that lies inside $C[B]$. Define $\beta^- := (\mathscr{I}^+(C[B]) \cap \Sigma[B]')$ to be the portion of the future horizon of $C[B]$ that is space-like separated from $\Sigma[B]$. We define the Cauchy slice $\Sigma[C[B]]$ as their union

$$\Sigma[C[B]] = \beta \equiv \beta^+ \cup \beta^- \,. \tag{132}$$

Let us first consider the vN-accessible case. We want to show that $S_{\text{gen}}(\beta) \leq S_{\text{gen}}(\alpha)$ for any $\alpha \subset \beta$. By the QFC, we have $S_{\text{gen}}(\alpha) \geq S_{\text{gen}}(\alpha \cup \beta^+)$ and $S_{\text{gen}}(\alpha \cup \beta^+) \geq S_{\text{gen}}(\beta)$, which completes the proof.[19]

---

[19]Note that $\eth a$ cannot intersect the same generator of $\beta^+$ or $\beta^-$ more than once. If it did, there would exist a lightlike geodesic between two points on $\eth a$. If any such geodesic is not contained in $a$, then it will be contained in $a''$. On the other hand if all such geodesics are contained in $a$ then they cannot be contained in $a''$. Both contradict the requirement that $a$ be a wedge.

For the max-accessible case, by the max-QFC and generic condition, we have $H^{\varepsilon/4}_{\text{max,gen}}(\alpha \cup \beta^+|\alpha) \ll -\kappa$ and $H^{\varepsilon/4}_{\text{max,gen}}(\beta|\alpha \cup \beta^+) \ll -\kappa$. But, by the chain rule,

$$H^{\varepsilon}_{\text{max,gen}}(\beta|\alpha) \leq H^{\varepsilon/4}_{\text{max,gen}}(\alpha \cup \beta^+|\alpha) + H^{\varepsilon/4}_{\text{max,gen}}(\beta|\alpha \cup \beta^+) + O(\log \varepsilon) \ll -K, \tag{133}$$

which is what we needed to show. $\qquad\square$

**Corollary 5.20.** *The causal wedge is contained in the max-entanglement wedge and vN-entanglement wedge,*

$$C[B] \subseteq \text{max-EW}[B], \text{vN-EW}[B]. \tag{134}$$

**Lemma 5.21.** *Let $b_1$ and $b_2$ be vN-accessible wedges with complementary conformal boundaries $B$ and $B'$, and let $\Sigma[B]$ (resp. $\Sigma[B']$) be the conformal boundary of the Cauchy slice $\Sigma[b_1]$ (resp. $\Sigma[b_2]$). Assuming the QFC, if $b_1$ (resp. $b_2$) is spacelike separated from $\Sigma[B']$ (resp. $\Sigma[B]$), then $b_1$ will also be spacelike separated from the entirety of $b_2$.*

*Proof.* The edge $\eth b_1$ can be decomposed as a disjoint union $\eth b_1 = \eth b_{1,0} \sqcup \eth b_{1,+} \sqcup \eth b_{1,-}$ where $\eth b_{1,0}$ is spacelike separated from $b_2$, $\eth b_{1,+}$ lies in the future of $\Sigma[b_2]$, and $\eth b_{1,-}$ lies in the past of $\Sigma[b_2]$. We define the deformed wedge $\tilde{b}_1 \supseteq b_1$ by shooting outwards, past lightrays from $\eth b_{1,+}$ and outwards, future lightrays from $\eth b_{1,-}$ until they hit $\Sigma[b_2]$. (These lightrays intersect $\Sigma[b_2]$ before reaching the asymptotic boundary because $b_1$ is assumed spacelike from $\Sigma[B']$.) By the QFC, $S_{\text{gen}}(\tilde{b}_1) \leq S_{\text{gen}}(b_1)$.

Let $b_2'$ be the spacelike complement of $b_2$. We can similarly decompose $\eth b_2' = \eth b_{2,0}' \sqcup \eth b_{2,+}' \sqcup \eth b_{2,-}'$ where $\eth b_{2,0}'$ is spacelike separated from $b_1$, $\eth b_{2,+}'$ is in the future of $\Sigma[b_1]$ and $\eth b_{2,-}'$ is in the past of $\Sigma[b_1]$. We define $\widetilde{b_2'}$ by shooting inwards, past lightrays from $\eth b_{2,+}'$ and inwards, future lightrays from $\eth b_{2,-}'$ until they hit $\Sigma[b_1]$. (These lightrays intersect $\Sigma[b_1]$ before reaching the asymptotic boundary because $b_2$ is assumed spacelike from $\Sigma[B]$.) By the QFC, $S_{\text{gen}}(\widetilde{b_2'}) \leq S_{\text{gen}}(b_2') = S_{\text{gen}}(b_2)$. (Recall again our convention that the global state is always purified using reference systems as necessary.)

Finally by strong sub-additivity we have

$$S_{\text{gen}}(\tilde{b}_1 \cap \widetilde{b_2'}) + S_{\text{gen}}(\tilde{b}_1 \cup \widetilde{b_2'}) \leq S_{\text{gen}}(\tilde{b}_1) + S_{\text{gen}}(\widetilde{b_2'}). \tag{135}$$

Combining inequalities, we have

$$S_{\text{gen}}(\tilde{b}_1 \cap \widetilde{b_2'}) + S_{\text{gen}}((\tilde{b}_1 \cup \widetilde{b_2'})') \leq S_{\text{gen}}(b_1) + S_{\text{gen}}(b_2). \tag{136}$$

But $\tilde{b}_1 \cap \widetilde{b_2'} \subseteq b_1$ and $(\tilde{b}_1 \cup \widetilde{b_2'})' \subseteq b_2$ with equalities if and only if $b_1$ is spacelike separated from $b_2$. Therefore because we assumed that $b_1$ and $b_2$ are vN-accessible, we get the reverse inequality

$$S_{\text{gen}}(\tilde{b}_1 \cap \widetilde{b_2'}) + S_{\text{gen}}((\tilde{b}_1 \cup \widetilde{b_2'})') \geq S_{\text{gen}}(b_1) + S_{\text{gen}}(b_2). \tag{137}$$

This completes the proof. $\qquad\square$

**Corollary 5.22** (Complementary causal wedge exclusion)**.** *Given a boundary wedge $B$ with complement $B'$, and assuming the QFC, the causal wedge of $B'$ lies in the complement of the vN-entanglement wedge of $B$:*

$$C[B'] \subseteq \text{vN-EW}[B]'. \tag{138}$$

*Proof.* It suffices to show that an arbitrary vN-accessible wedge $b$ is spacelike separated from $C[B']$. Let $\Sigma$ be a Cauchy slice for $M$ such that $\Sigma[b] \subseteq \Sigma$, and let $\Sigma[B']$ be the intersection of its conformal boundary with $B'$. From lemma 5.19, we know that $C[B']$ is vN-accessible, and that we can choose $\Sigma[C[B']]$ to have conformal boundary $\Sigma[B']$. Moreover, from lemma 5.18 it follows that $B$ is spacelike to $C[B']$. We can therefore apply lemma 5.21. $\qquad\square$

**Theorem 5.23** (vN-Entanglement wedge complementarity)**.** *Assuming the QFC, the complement of the vN-entanglement wedge of $B$ is equal to the vN-entanglement wedge of the complement,*

$$\text{vN-EW}[B]' = \text{vN-EW}[B'].\tag{139}$$

*Moreover the vN-EW is vN-accessible and its edge $\eth\text{vN-EW}[B]$ is the minimal generalized entropy quantum extremal surface.*

*Proof.* By corollary 5.22, we see that *any* wedges $b_1$ vN-accessible to $B$ and $b_2$ vN-accessible to $B'$ satisfy all the conditions of lemma 5.21 and so must be everywhere space-like separated. It follows that $\text{vN-EW}[B]$ and $\text{vN-EW}[B']$ must be spacelike separated.

To show that they are in fact complementary, it only remains to find a single complementary pair of wedges $b$ and $b'$ that are both vN-accessible. (This also shows that the vN-EW is vN-accessible.) To do so, we consider the quantum maximin wedge $b$ [4, 51]. This is defined by first choosing a Cauchy slice $\Sigma$ for $M$ that contains $\eth B$ and finding the minimal-$S_{\text{gen}}$ wedge $b$ with $\eth b \in \Sigma$. One then maximizes that minimal-$S_{\text{gen}}$ wedge over all possible Cauchy slices $\Sigma$. Both $b$ and $b'$ are therefore vN-accessible, with $\Sigma[b] = \Sigma \cap b$ and $\Sigma[b'] = \Sigma \cap b'$. It can be shown that $b$ (and hence also $b'$) is extremal.

To show that $b$ has minimal-$S_{\text{gen}}$ among all extremal wedges, and hence is the same as the region found by the QES prescription, one simply shoots lightrays from any other extremal wedge $b_3$ to obtain a wedge $\tilde{b}_3$ with edge $\eth\tilde{b}_3 \subseteq \Sigma$. By the QFC, $S_{\text{gen}}(\tilde{b}_3) \leq S_{\text{gen}}(b_3)$. But by definition $S_{\text{gen}}(\tilde{b}_3) \geq S_{\text{gen}}(b_1)$. See [4, 51] for details. $\qquad\square$

**Theorem 5.24** (max-EW $\subseteq$ vN-EW $\subseteq$ min-EW)**.** *Let $B \subset M_\partial$ be a wedge. For sufficiently small $\varepsilon$, we have*

$$\text{max-EW}[B] \subseteq \text{vN-EW}[B] \subseteq \text{min-EW}[B].\tag{140}$$

*This is shown in Figure 7.*

*Proof.* It will suffice to prove that the max-EW is always contained in the vN-EW. Applying this and Theorem 5.23 to the complementary region $B' \subseteq M_\partial$ immediately implies that the vN-EW is contained in the min-EW.

For sufficiently small $\varepsilon$, every max-accessible wedge is also vN-accessible because $H_{\text{max,gen}}^{\varepsilon}(b|b') \geq S_{\text{gen}}(b|b')$ and $\Theta_{\text{max}}^{\varepsilon} \geq \Theta$ by Lemma 4.8. Therefore the wedge union defining the vN-EW is at least as large as that defining the max-EW. $\qquad\square$

**Remark 5.25.** When the max-EW and min-EW are equal (up to perturbatively small corrections), we say that an entanglement wedge $\text{EW}(B) = \text{max-EW}(B) = \text{min-EW}(B)$ exists. When it exists, the entanglement wedge is also equal to the vN-EW, by Theorem 5.24, and hence is bounded by the minimal QES by Theorem 5.23.

**Corollary 5.26** (max- and min-EW conformal boundaries)**.** *The conformal boundary of the max-EW and min-EW for any boundary wedge $B$ is itself equal to $B$,*

$$\text{max-EW}[B] \cap M_\partial = \text{min-EW}[B] \cap M_\partial = B.\tag{141}$$

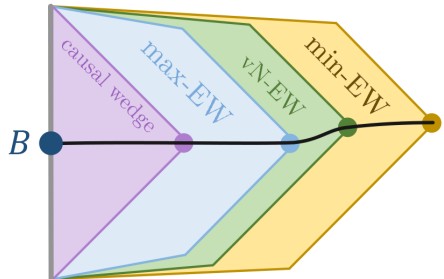

Figure 7: The containment of each wedge discussed in this section, assuming the validity of the QFC and max-QFC.

*Proof.* By definition, the conformal boundary of max-EW$[B]$ includes $B$ whenever $G_{(\varepsilon,K)}(B)$ is nonempty, which is always true because of Corollary 5.20. The converse statement, that max-EW$[B] \cap M_\partial \subseteq B$, follows from combining Theorem 5.24 and Theorem 5.23. The min-EW proof follows from applying the same arguments to $B'$. □

**Lemma 5.27** (Unions of accessible wedges are accessible)**.** *Let $b_1$ and $b_2$ be $(\varepsilon,K)$ max-accessible (resp. vN-accessible) wedges with conformal boundary $B$. Then their wedge union $b = b_1 \uplus b_2$ is $(5\varepsilon,K)$ max-accessible (resp. vN-accessible). Moreover the conformal boundary of $\Sigma[b]$ can be chosen to agree with conformal boundary of $\Sigma[b_1]$.*

*Proof.* It is helpful to classify the edge $\eth b_1$ based on its relationship to $b_2$, and vice versa. Let

1. $\eth b_{1,I}$ be the part of $\eth b_1$ inside $b_2$,

2. $\eth b_{1,O}$ be the part inside $b_2'$,

3. $\eth b_{1,F}$ be the part in the future of $\eth b_2$,

4. $\eth b_{1,P}$ be the part in the past of $\eth b_2$,

and analogously for $b_1 \leftrightarrow b_2$. The edge $\eth b$ can also be decomposed into four pieces as follows:

$$\eth b = \eth b_{1,O} \sqcup \eth b_{2,O} \sqcup F[b_1, b_2] \sqcup F[b_2, b_1], \tag{142}$$

where we have defined $F[b_1, b_2] := \eth b \cap \partial J^-[\eth b_{1,F}] \cap \partial J^+[\eth b_{2,P}]$ and $F[b_2, b_1] := \eth b \cap \partial J^-[\eth b_{2,F}] \cap \partial J^+[\eth b_{1,P}]$. Note that, thanks to corollary 5.26, the conformal boundary of $b$ is itself $B$. Therefore by corollary 5.22 and Theorem 5.24, the entire future outwards null congruence from $\eth b_{1,P}$ hits the edge $\eth b$ before it reaches the asymptotic boundary, and likewise for the past congruence from $\eth b_{1,F}$.

We first show that $b$ is max-antinormal (resp. vN-antinormal). Consider some $p \in \eth b_{1,O}$. By assumption, $\Theta^\varepsilon_{\max}[b_1; p] \le 0$ (resp. $\Theta[b_1; p] \le 0$), in both the future and past directions. Since $b_1 \subseteq b$, we also have $\Theta^\varepsilon_{\max}[b; p] \le 0$ (resp. $\Theta[b; p] \le 0$) by strong subadditivity, lemma 4.9. An analogous argument applies for $p \in \eth b_{2,O}$.

Now consider the other two pieces of $\eth b$. By symmetry, it is sufficient to consider only $F[b_1, b_2]$. For $p \in F[b_1, b_2]$, let $q \in \partial b_{1,F}$ be lightlike separated from $p$. Then the max-QFC implies

$$\Theta^\varepsilon_{-,\max}[\widetilde{b_1}, p] \le \Theta^\varepsilon_{-,\max}[b_1, q] \le 0, \tag{143}$$

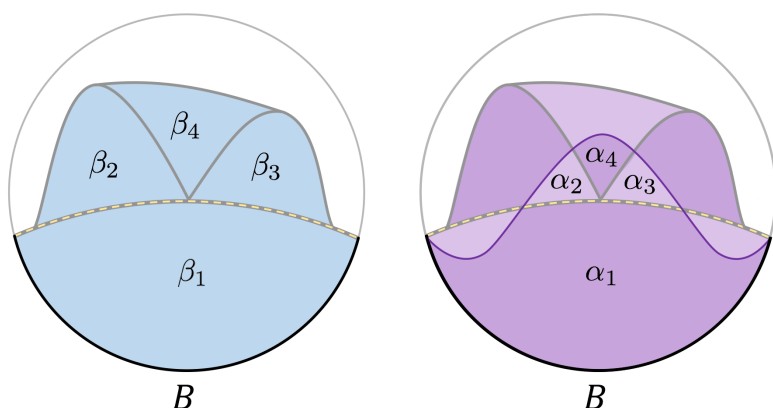

Figure 8: Left: A bulk Cauchy slice with an example $\beta = \bigsqcup_i \beta_i = \Sigma[b]$ divided into its four constituents. Right: an arbitrary region $\alpha = \bigsqcup_i \alpha_i \subseteq \beta$ is drawn, showing each constituent $\alpha_i \subseteq \beta_i$. Different shadings are meant to clarify the boundaries between each region.

where $\widetilde{b_1}$ is formed from $b_1$ by shooting an outwards, past-directed null congruence from a neighbourhood of $p$ to a neighbourhood of $q$ on $\eth b$. Finally strong subadditivity implies $\Theta^\varepsilon_{-,\max}[b;p] \leq \Theta^\varepsilon_{-,\max}[\widetilde{b_1};p]$. Analogous arguments bound $\Theta^\varepsilon_{+,\max}[b;p]$ using the max-antinormality of $b_2$ and bound $\Theta_\pm[b;p]$ in the vN-accessible case.

It remains to construct a Cauchy slice $\Sigma[b]$ and prove that it satisfies the desired properties. We define

$$\Sigma[b] = \Sigma[b_1] \sqcup (\partial J^+[b_1] \cap J^-[\Sigma[b_2]]) \sqcup (\partial J^-[b_1] \cap J^+[\Sigma[b_2]]) \sqcup (\Sigma[b_2] \cap b_1'). \quad (144)$$

This is notationally somewhat messy so let us explain each portion and introduce some simpler notation. The first piece of the Cauchy slice $\beta = \Sigma[b]$ consists of the full Cauchy slice $\beta_1 = \Sigma[b_1]$ for $b_1$. We then attach future ($\beta_2 = \partial J^+[b_1] \cap J^-[\Sigma[b_2]]$) or past ($\beta_3 = \partial J^-[b_1] \cap J^+[\Sigma[b_2]]$) outwards null congruences from the parts of the edge $\eth b_1$ that lie in the interior of $b$ (i.e. $\eth b_{1,P}$, $\eth b_{1,F}$). These null congruences are included until either they hit the edge of $b$, or they reach the Cauchy slice $\Sigma[b_2]$. Finally we need to attach $\beta_4 = \Sigma[b_2] \cap b_1'$, namely the part of the Cauchy slice for $b_2$ that lies outside $b_1$. (Note that the conformal boundary of $\Sigma[b]$ is the same as that of $\Sigma[b_1]$ by construction.)

The full construction is illustrated in Figure 8.

We first prove that $\beta$ is a suitable Cauchy slice for $b$ in the von Neumann case, because it is somewhat simpler and so will serve as a warm up for the max-entropy problem. (For notational simplicity, below we will sometimes refer to $S_{\text{gen}}$ of a Cauchy slice of a wedge when we mean the $S_{\text{gen}}$ of the wedge.) If $a \subseteq b$ has edge $\eth a \in \Sigma[b]$ then $a$ has a Cauchy slice $\alpha = a \cap \Sigma[b]$. Let $\alpha_i = \alpha \cap \beta_i$. Since $b_1$ is vN-accessible, we have $S_{\text{gen}}(\beta_1) \leq S_{\text{gen}}(\alpha_1)$. Strong subadditivity therefore implies $S_{\text{gen}}(\beta_1 \sqcup \alpha_2 \sqcup \alpha_3 \sqcup \alpha_4) \leq S_{\text{gen}}(\alpha)$. The antinormality of $b_1$ ensures via the QFC and strong subadditivity that $S_{\text{gen}}(\beta_1 \sqcup \beta_2 \sqcup \beta_3 \sqcup \alpha_4) \leq S_{\text{gen}}(\beta_1 \sqcup \alpha_2 \sqcup \alpha_3 \sqcup \alpha_4)$. Finally the vN-accessibility of $b_2$ and strong subadditivity means that $S_{\text{gen}}(\beta) \leq S_{\text{gen}}(\beta_1 \sqcup \beta_2 \sqcup \beta_2 \sqcup \alpha_4)$. In summary, we have $S_{\text{gen}}(b) = S_{\text{gen}}(\beta) \leq S_{\text{gen}}(\alpha) = S_{\text{gen}}(b')$, which is what we needed to show.

Now let us consider the max-entropy case. By strong subadditivity and the max-accessibility of $b_1$, we have

$$H_1 := H^\varepsilon_{\max,\text{gen}}(\beta_1 \sqcup \alpha_2 \sqcup \alpha_3 \sqcup \alpha_4 | \alpha) \leq -K, \quad (145)$$

whenever the inclusion $\alpha \subseteq \beta_1$ is strict.

By the max-QFC, the generic condition, and strong subadditivity, we have

$$H_2 := H^\varepsilon_{\text{max,gen}}(\beta_1 \sqcup \beta_2 \sqcup \alpha_3 \sqcup \alpha_4 | \beta_1 \sqcup \alpha_2 \sqcup \alpha_3 \sqcup \alpha_4) \lesssim -\kappa \ll -K, \tag{146}$$

whenever the inclusion $\alpha_2 \subseteq \beta_2$ is strict. Similarly,

$$H_3 := H^\varepsilon_{\text{max,gen}}(\beta_1 \sqcup \beta_2 \sqcup \beta_3 \sqcup \alpha_4 | \beta_1 \sqcup \beta_2 \sqcup \alpha_3 \sqcup \alpha_4) \lesssim -\kappa \ll -K, \tag{147}$$

whenever $\alpha_3 \subseteq \beta_3$ is strict. Finally, strong subadditivity and the max-accessibility of $b_2$ ensure

$$H_4 := H^\varepsilon_{\text{max,gen}}(\beta | \beta_1 \sqcup \beta_2 \sqcup \beta_3 \sqcup \alpha_4) \le -K, \tag{148}$$

whenever $\alpha_4 \subseteq \beta_4$ is strict.

Since $\alpha \subseteq \beta$ is required to be a strict inclusion, at least one of the four inclusions $\alpha_i \subseteq \beta_i$ must be strict. If only one inclusion is strict, then the corresponding inequality immediately gives $H^\varepsilon_{\text{max,gen}}(\beta | \alpha) < -K$. If more than one inclusion is strict then we can use the chain rule to write

$$H^{5\varepsilon}_{\text{max,gen}}(\beta | \alpha) \le H_1 + H_2 + H_3 + H_4 + O(\log \varepsilon) \le -4K + O(\log \varepsilon) \le -K. \tag{149}$$

In the last step we used the assumption $K \gg -\log \varepsilon$. This completes the proof. $\square$

**Theorem 5.28** (The max-EW is max-accessible). *For any boundary region $B$, the $(\varepsilon, K)$ max-entanglement wedge $b$ is $(\varepsilon', K)$ max-accessible with $\varepsilon' = O(\varepsilon)$. Moreover, the conformal boundary of the Cauchy slice $\Sigma[b]$ can be chosen to be any desired Cauchy slice for $B$.*

*Proof.* The proof follows immediately from Lemma 5.27. To select a particular conformal boundary for the Cauchy slice $\Sigma[b]$, we again apply Lemma 5.27 with $b_1$ equal to the causal wedge $C[B]$ and $C[B]$ chosen to have the desired conformal boundary, as allowed by Lemma 5.19. $\square$

**Theorem 5.29** (Vanishing expansions). *If the $(\varepsilon, K)$ max-entanglement wedge $b$ of a boundary region $B$ is itself $(\varepsilon/3, K)$ max-accessible, then we must have*

$$\Theta^\varepsilon_{+,\text{max}}[b; p], \ \Theta^\varepsilon_{-,\text{max}}[b; p] = 0.$$

*for all $p \in \eth b$.*

*Proof.* To derive a contradiction, we can assume without loss of generality that there exists $p \in \eth b$ such that $\Theta^\varepsilon_{+,\text{max}}[b; p] < 0$. We will construct an $(\varepsilon, K)$ max-accessible region not contained in the max-EW. By Remark 4.5, we must also have $\Theta^\varepsilon_{+,\text{max}}[\tilde{b}; \tilde{p}] < 0$ for any sufficiently small deformation $\tilde{b}$ of $b$ mapping $p$ to $\tilde{p}$. Suppose we define $\tilde{b}$ by deforming outwards along a null congruence in the past direction. Then we also have $\Theta^\varepsilon_{-,\text{max}}[\tilde{b}; \tilde{p}] \le 0$ by the max-QFC, and hence $\tilde{b}$ is $\varepsilon/3$ max-antinormal (and hence also $\varepsilon$ max-antinormal).

Now, take $\beta = \Sigma[\tilde{b}]$ to be the union of $\beta_1 = \Sigma[b]$ with the null congruence $\beta_2$ from $b$ to $\tilde{b}$. Let $a \subseteq \tilde{b}$ with $\eth a \in \Sigma[\tilde{b}]$ have Cauchy slice $\alpha = a \cap \beta$ and let $\alpha_i = \alpha \cap \beta_i$. By the max-QFC and the generic condition, we have

$$H^{\varepsilon/3}_{\text{max,gen}}(\beta | \beta_1 \sqcup \alpha_2) \lesssim -\kappa \ll -K, \tag{150}$$

if $\alpha_2 \subseteq \beta_2$ is strict. Meanwhile by the $(\varepsilon/3, K)$ max-accessibility of $b$ and strong subadditivity we have

$$H^{\varepsilon/3}_{\text{max,gen}}(\beta_1 \sqcup \alpha_2 | \alpha) \le -K, \tag{151}$$

if $\alpha_1 \subseteq \beta_1$ is strict. The desired inequality $H^\varepsilon_{\text{max,gen}}(\beta | \alpha) \le -K$ follows via the chain rule. $\square$

**Remark 5.30.** The assumption in Theorem 5.29 is slightly stronger than that derived in Theorem 5.28, which only showed that the max-EW is $(\varepsilon', K)$ max-accessible for some $\varepsilon' = O(\varepsilon)$. In most situations of physical interest, one expects the max-EW to be (approximately) constant over a wide range of values for $\varepsilon$. In such a situation, the assumption of Theorem 5.29 is always (approximately) satisfied.

**Theorem 5.31** (Nesting). *For any two boundary wedges $B_2 \subseteq B_1$, the $(\varepsilon, K)$ max-EW, vN-EW, and $(\varepsilon, K)$ min-EW of $B_2$ are entirely contained respectively in the $(5\varepsilon, K)$ max-EW, vN-EW, and $(5\varepsilon, K)$ min-EW of $B_1$.*

*Proof.* Since we have already proven the equivalence of the vN-EW and the region found by the QES prescription, the von Neumann case is a standard result, but we include it here for completeness. The proof of the min-EW case follows by applying the max-EW result to the complementary regions $B_1' \subseteq B_2'$.

Let $b_2$ be an $(\varepsilon, K)$ max-accessible (resp. vN-accessible) wedge with conformal boundary $B_2$. Let $b_1$ be an $(\varepsilon, K)$ max-accessible wedge with conformal boundary $B_1$. We can then take the union of these two wedges in exactly the same way as described in Lemma 5.27. Call this union $b$. The only difference in the current setting will be that $b$ will contain some portion of the conformal boundary which is not in the domain of dependence of the conformal boundary of $b_2$. This does not affect any of the relevant inequalities (e.g. the chain rule, strong sub-additivity) assuming reflecting boundary conditions at the asymptotic boundary. By Lemma 5.27, we end up with a $(5\varepsilon, K)$ max-accessible (resp. vN-accessible) wedge, $b$, whose conformal boundary is $B_1$ and which contains $b_2$. This produces the desired statement. $\qquad\square$

**Theorem 5.32** (Time-reflection symmetric wedges). *Let $M$ be time-reflection symmetric with invariant Cauchy slice $\Sigma$ and let $B$ be a boundary region with $\eth B \in \Sigma$. Let $b$ be the $(\varepsilon, K)$ max-entanglement wedge for $B$. Then $\eth b \in \Sigma$.*

*Proof.* By time-reflection symmetry of $M$, for every $(\varepsilon, K)$ max-accessible wedge, $b_1$, there exists a time-reflected version, $\hat{b}_1$, which is also $(\varepsilon, K)$ max-accessible. The wedge union over all max-accessible wedges will then manifestly produce a time-reflection symmetric wedge. By the definition of $b$, we see that $b$ itself must be time-reflection symmetric and so $\eth b \in \Sigma$. $\qquad\square$

Note that this statement is significantly weaker than what one might have hoped for. A reasonable sounding statement is that when $M$ has a moment of time-reflection symmetry the max-EW for a region $B$ with $\eth B \in \Sigma$ should be max-accessible with $\Sigma = \Sigma[b_1]$ in Definition 5.7. While this statement is true for the vN-EW, it appears likely that the corresponding statement fails for the max- and min-EW in general. We suspect that this may be related to upcoming work [52], which suggests that a tensor network representation of a bulk state cannot necessarily be associated to the time-symmetric slice, even when such a slice exists.

# 6 The continuum limit and type II von Neumann algebras

Until now, we have focused our attention on regulated bulk theories featuring finite-dimensional algebras $\mathcal{M}_b$, while conjecturing that generalized one-shot entropies should be UV-finite and regulator-independent. However, it has recently been shown that in certain settings one can make interesting progress in understanding generalized entropy without regulation by studying the algebraic structure of quantum gravity in the weak coupling limit [36, 37, 53, 54]. We now briefly discuss how generalized one-shot entropies can be understood in such a framework; we refer readers to the aforementioned papers for more details.

Following [36, 37, 54], we will consider the $G \to 0$ limit of small perturbations around a black hole background, and take the bulk wedge $b$ of interest to be the right black hole exterior. In this limit, the quantum gravity Hilbert space can be understood without introducing any regulator as the Hilbert space of continuum quantum field theory (QFT) on the black hole background, together with an additional degree of freedom describing the timeshift between the two boundaries. QFT operators in the right exterior are described by a Type III von Neumann algebra $\mathcal{A}_{r,0}$, which means that density matrices from a regulated field theory have no continuum limit.

Meanwhile, the operator $\hat{A}(\eth b)/4G$ generates boosts at the horizon, which change the timeshift while keeping fields in each exterior fixed relative to their respective boundaries. Such an operator renders the quantum fields singular at the horizon and hence also has no continuum limit. Indeed, Raychaudhuri's equation together with Einstein's equations show that

$$\frac{\hat{A}(\eth b) - A_0}{4G} + \hat{h}_r = \hat{H}_R - E_0 \,, \tag{152}$$

where $\hat{H}_R$ is the right ADM mass, $A_0$ and $E_0$ are respectively the reference horizon area and mass of the black hole background, and $\hat{h}_r$ is a one-sided boost operator on the quantum fields in the right exterior. In fixed-background QFT, the operator $\hat{h}_r$ is UV-divergent. In gravity, however, this divergence is absorbed into a renormalization of $G$ in $[\hat{A}(\eth b) - A_0]/4G$. On the right hand side, the ADM mass $H_R$ is UV-finite, but diverges for a fixed radius black hole as $G \to 0$. This divergence is cancelled by subtracting $E_0$. The result is that the renormalized ADM mass $\hat{h}_R = \hat{H}_R - E_0$ is a finite operator in the $G \to 0$ continuum quantum gravity theory that is not present in a quantum field theory on the black hole background.

The addition of this extra quantum gravity operator $\hat{h}_R$ to the QFT algebra $\mathcal{A}_{r,0}$ leads to the full quantum gravity algebra $\mathcal{A}_r$ for the black hole right exterior. This algebra turns out to be a Type II von Neumann factor, implying that the center of $\mathcal{A}_r$ consists only of multiples of the identity; all central operators such as $[\hat{A}(\eth b) - A_0]/4G$ in the regulated theory are UV-divergent and hence do not exist in the continuum theory.[20] It also means that one can define a trace $\mathrm{tr}_{\mathrm{II}}$ – and hence also density matrices – for $\mathcal{A}_r$, which are unique up to an overall factor related to the choice of reference energy $E_0$. One can show [37] that the density matrix of this Type II algebra is proportional to the continuum limit of

$$\rho_{b,\mathrm{gen}} = e^{(A_0 - \hat{A}(\eth b))/4G} \rho_{b,\mathrm{can}} \,, \tag{153}$$

while the trace is

$$\mathrm{tr}_{\mathrm{II}}[\cdot] = \lim_{G \to 0} e^{-A_0/4G} \mathrm{tr}_{\mathrm{gen}}[\cdot] = \lim_{G \to 0} \lim_{\delta \to 0} \mathrm{tr}_{\mathrm{can}}\left[ e^{(\hat{A}(\eth b) - A_0)/4G} (\cdot) \right] \,. \tag{154}$$

In other words, the only choice of trace (and density matrices) in the regulated theory, where the algebra has a center, with a sensible semiclassical, continuum limit $G, \delta \to 0$ (up to a state-independent factor $e^{-A_0/4G}$) is the generalized trace (and generalized density matrices) that we defined in Section 3.

**The one-shot GSL for Type II$_\infty$ algebras**

In Section 4, we argued for the existence of a one-shot GSL. One setting in which the ordinary GSL can be rigorously defined as an inequality between entropies was described in Section 4

---

[20]One can make the area operator UV finite by smearing it over some small region of spacetime. However doing so makes it no longer central.

of [37]. We now introduce a similarly rigorous continuum definition of a one-shot GSL, along with a direct proof that does not rely on the one-shot QFC.

In the construction of [37], one first introduces a new timescale $T$ that diverges in the $G \to 0$ limit.[21] We consider black holes that have arbitrary boundary excitations at times $t = O(1)$, and additional arbitrary boundary excitations at times $t = T + O(1)$, but with the black hole allowed to equilibrate during the intervening period.

There is then a Type $\text{II}_\infty$ von Neumann algebra $\mathcal{A}_R$ generated by the (renormalized) right boundary Hamiltonian along with both early- and late-time right boundary (noncentral) single-trace operators. The entropy of this algebra is equal to the generalized entropy of the black hole bifurcation surface. The algebra $\mathcal{A}_R$ contains a Type $\text{II}_\infty$ von Neumann subalgebra $\widetilde{\mathcal{A}}_R$ generated by only the boundary Hamiltonian and late-time single-trace operators. The entropy of this subalgebra is equal to the generalized entropy of the black hole horizon during the equilibration period between the two sets of excitations.

If we choose the constant factor $e^{A_0/4G}$ from (154) to be the same for both $\widetilde{\mathcal{A}}_R$ and $\mathcal{A}_R$, the inclusion $\widetilde{\mathcal{A}}_R \subseteq \mathcal{A}_R$ is trace-preserving, meaning that the trace (on $\widetilde{\mathcal{A}}_R$) of an operator in $\widetilde{\mathcal{A}}_R$ is equal to its trace as an element of the larger algebra $\mathcal{A}_R$. It is a standard fact about von Neumann algebras [55] that entropy is monotonically decreasing under trace-preserving inclusions. This fact is sufficient to derive a "discretized" version of the generalized second law: namely that the entropy of any state on $\mathcal{A}_R$ (i.e. the generalized entropy of the bifurcation surface) is less than or equal to the entropy on $\widetilde{\mathcal{A}}_R$ (the generalized entropy of the temporarily equilibrated black hole horizon).

This derivation extends to a one-shot GSL as follows. Let $\tilde{b}$ be the outer wedge of a cut of the temporarily equilibrated horizon, and let $b$ be the entire black hole exterior. Finally let $\rho_{\tilde{b}}$ and $\rho_b$ be the density matrices of a state $|\Psi\rangle$ on $\widetilde{\mathcal{A}}_R$ and $\mathcal{A}_R$ respectively.

Because of the relationship (154) between the unique trace on the Type II algebras and the generalized trace, the conditional generalized min-entropy limits to the conditional min-entropy on the Type II algebra as

$$\lim_{G \to 0} H^\varepsilon_{\min,\text{gen}}(b|\tilde{b}) = H^\varepsilon_{\min,\text{II}}(b|\tilde{b}) = -\inf_{\rho_b^\varepsilon \in \mathcal{B}^\varepsilon(\rho_b)} \inf_{\sigma_{\tilde{b}}} \inf \left\{ \lambda : \rho_b^\varepsilon \leq e^\lambda \sigma_{\tilde{b}} \right\}, \qquad (155)$$

where $\sigma_{\tilde{b}}$ is a normalized density matrix on $\widetilde{\mathcal{A}}_R$. Note that (155) is independent of the choice of normalization for the traces on $\widetilde{\mathcal{A}}_R$ and $\mathcal{A}_R$ so long as their relative normalization is chosen so that the inclusion is trace-preserving.

It is easy to check $H^\varepsilon_{\min,\text{gen}}(b|\tilde{b}) \leq 0$: suppose there existed a normalized density matrix $\sigma_{\tilde{b}}$ such that $e^\lambda \sigma_{\tilde{b}} - \rho_b^\varepsilon \geq 0$. Because the inclusion $\widetilde{\mathcal{A}}_R \subseteq \mathcal{A}_R$ is trace-preserving, we have

$$\text{tr}_{\mathcal{A}_R}(e^\lambda \sigma_{\tilde{b}} - \rho_b^\varepsilon) = e^\lambda \text{tr}_{\widetilde{\mathcal{A}}_R}(\sigma_{\tilde{b}}) - 1 = e^\lambda - 1 \leq 0. \qquad (156)$$

Thus, $\lambda \leq 0$ for all candidate $\lambda$ in the allowed set. The optimal $\lambda$ will saturate this inequality, $\lambda = 0$, if and only if $\rho_b^\varepsilon \in \widetilde{\mathcal{A}}_R$.

Similarly, the limit of the conditional generalized max-entropy $H^\varepsilon_{\max,\text{gen}}(b|\tilde{b})$ is simply

$$\lim_{G \to 0} H^\varepsilon_{\max,\text{gen}}(b|\tilde{b}) = 2 \inf_{\rho_b^\varepsilon \in \mathcal{B}^\varepsilon(\rho_b)} \sup_{\sigma_{\tilde{b}}} \log \text{tr} \left| \left( \rho_b^\varepsilon \right)^{1/2} \sigma_{\tilde{b}}^{1/2} \right|, \qquad (157)$$

where $|X| := \sqrt{X^\dagger X}$. Von Neumann algebras always admit polar decompositions, so there

---

[21]More precisely, we require $\beta \ll T \ll t_{\text{scr}}$ where $t_{\text{scr}}$ is the scrambling time of the black hole.

exists a partial isometry $v \in \mathcal{A}_R$ such that $v\left(\rho_b^\varepsilon\right)^{1/2} \sigma_{\tilde{b}}^{1/2} = |\left(\rho_b^\varepsilon\right)^{1/2} \sigma_{\tilde{b}}^{1/2}|$. Hence

$$\lim_{G \to 0} H_{\max,\text{gen}}^\varepsilon(b|\tilde{b}) = 2 \inf_{\rho_b^\varepsilon \in \mathcal{B}^\varepsilon(\rho_b)} \sup_{\sigma_{\tilde{b}}} \log \text{tr}\left[v\left(\rho_b^\varepsilon\right)^{1/2} \sigma_{\tilde{b}}^{1/2}\right] \tag{158}$$

$$\leq 2 \inf_{\rho_b^\varepsilon \in \mathcal{B}^\varepsilon(\rho_b)} \sup_{\sigma_{\tilde{b}}} \log\left(\text{tr}[v^\dagger v \rho_b^\varepsilon]\,\text{tr}[\sigma_{\tilde{b}}]\right) \leq 0. \tag{159}$$

In the second step we used the Cauchy-Schwarz inequality.

# 7 Discussion

One of the biggest lessons we have learned in the last decade of quantum gravity research is that you can get an awfully long way by taking theorems in classical general relativity and turning them into correct statements about semiclassical gravity simply by replacing areas with generalized entropies [7,22,49]. On the other hand, the lesson of one-shot quantum Shannon theory is that von Neumann entropies should almost never feature in operational statements – such as entanglement wedge reconstruction – that involve only a single copy of a state. If they appear to do so, it is probably because you're only considering special classes of nice states where those von Neumann entropies are equal to the one-shot entropies that actually matter. Our goal in this paper was to synthesize both of these lessons into a consistent framework of holographic one-shot information theory.

We defined two regions, the max-EW and min-EW, associated to any boundary subregion $B$, that we conjectured to have operational interpretations valid for any semiclassical state. The max-EW is the largest region that can be state-specifically reconstructed with access just to $B$. The min-EW is the smallest region whose complement cannot influence the state on $B$. We also provided multiple pieces of evidence corroborating these conjectures, demonstrating self-consistency and reduction to known correct statements in certain cases. To do so, we conjectured new quantum focusing conjectures for max- and min-entropies and extended the frameworks of both one-shot quantum Shannon theory and state-specific reconstruction to finite-dimensional von Neumann algebras.

## Entanglement wedge reconstruction as quantum state merging

A guiding principle of this work and the work of [1] is that bulk reconstruction can be viewed through the operational lens of (one-shot) quantum state-merging. In [1] this was argued in special cases. Here we have improved that argument, explaining how in any spacetime the QES prescription can be reformulated in terms of (traditional) quantum state merging through a Cauchy slice. In turn, this reformulation helped us propose one-shot versions of the QES prescription by replacing state merging with one-shot state merging, leading to our max-EW and min-EW.

While tensor network models [56–60] oversimplify quantum gravity in many ways (as we shall discuss below), the success of (multiparty) state merging in describing bulk reconstruction suggests something is deeply correct about them. The holographic map seems to push information "outwards" toward the boundary by acting in a spatially local way on some time slice, similar to how tensors act locally in a tensor network.

On a different note, one main advantage of phrasing entanglement wedge reconstruction operationally is to detach entanglement wedges from the restrictive context of AdS/CFT. In particular, the framework we have put forth leads to a nice picture for the flow of quantum information in general, dynamical spacetimes. It is thus natural to expect that our prescription can help to understand entanglement wedge reconstruction for general regions in spacetime, as was explored in [62,63].

### The emergence of time

A major open problem in holography is to give an information-theoretic interpretation of the emergence of dynamical (and generally covariant) bulk time; that is, how bulk time fits into the story of bulk reconstruction. Tensor networks have helped us understand the emergence of an extra bulk spatial dimension, but so far have not provided a satisfactory understanding of general covariance.

As a generally covariant information-theoretic property of holographic spacetimes, the QES prescription seemingly should provide hints towards the right answer to this question, in the same way that tensor network models were inspired by the earlier Ryu-Takayanagi formula [2] which describes the classical limit of the QES prescription for time-reflection symmetric states.

However, so far no clear hint has appeared. In particular, the number of equivalent ways that the QES prescription can be formulated make it hard to know what the correct insight is supposed to be. Is the key point the local invariance of $S_{\text{gen}}$ under small perturbations of the quantum extremal surface? Or is the natural operational explanation in terms of the "maximin" prescription, the global maximization of minimum-$S_{\text{gen}}$ surfaces over all Cauchy slices [4,51]? Or perhaps even the maximization of $S_{\text{gen}}$ within a timelike hypersurface [64]?

Because one-shot entropies only satisfy the chain-rule as an (approximate) inequality and not as an equality, there are far fewer equivalent definitions of the max- and min-EWs. In fact, we are not aware of any nontrivial ways of reformulating our covariant definitions of those wedges, or of any alternative proposals that could satisfy the required properties. We therefore expect that our proposal (Conjecture 5.11) will significantly narrow the search for an information-theoretic meaning for dynamical bulk time.

The first lesson of our proposal is that the state-merging process described in [1] can happen through any Cauchy slice of max-EW($B$); only one slice needs to satisfy the required properties for information to successfully flow to the boundary. This seems relatively intuitive even if we don't have a specific microscopic explanation for it. But we also learned that the edge of max-EW($B$) needs to satisfy a anti-normality property to act as an origin for information flow. So both a global condition on a Cauchy slice of max-EW($B$) and a local condition on the edge of max-EW($B$) seem important. We don't have good intuition for why the latter condition is necessary from an information-theoretic point of view, but its existence seems key to understanding the emergence of time.

### One-shot energy conditions

A great deal of progress has been made by taking information-theoretic constraints from quantum gravity and taking a $G \rightarrow 0$ limit to recover purely field-theoretic statements. A prime example of this is discussed in [22], where the authors took the $G \rightarrow 0$ limit of the quantum focussing conjecture (QFC) and obtained the so-called quantum null energy condition (QNEC). This condition was later derived using purely field-theoretic techniques [65, 66], thus corroborating aspects of the quantum focusing conjecture itself.

In principle, the same game could be played here with the one-shot QFCs proposed in Conjectures 4.12 and 4.13. One could imagine taking the $G \rightarrow 0$ limits of the one-shot QFCs in the hopes of recovering interesting field theoretic inequalities. It is not obvious, however, exactly how to phrase these limits in terms of continuum field theoretic quantities, and naive attempts to do so suffer from various technical issues. We therefore leave the task of defining one-shot versions of the QNEC to future work.

The proof of the QNEC due to Ceyhan & Faulkner [65] was inspired by the so-called *Ant Conjecture* of Wall [67]. We expect that a one-shot version of Wall's conjecture will concern the nature of fluctuations in null energy, whereas the ant conjecture as presented in [67] is about the mean null energy flowing past a point. Understanding this better may prove helpful

in determining the correct statement of one-shot versions of the QNEC. Again, we defer a detailed analysis of these issues to future work.

# Acknowledgments

We thank Elba Alonso-Monsalve, Raphael Bousso, Netta Engelhardt, Thomas Faulkner, Daniel Harlow, Patrick Hayden, Arvin Shahbazi-Moghaddam, Renato Renner, Ronak Soni, Jon Sorce, Michael Walter, Jinzhao Wang, Edward Witten and Freek Witteveen for discussions.

**Funding information** CA is supported by the Simons Foundation as an "It from Qubit" fellow, the Air Force Office of Scientific Research under the award number FA9550-19-1-0360, the US Department of Energy under grant DE-SC0012567, the John Templeton Foundation and the Gordon and Betty Moore Foundation via the Black Hole Initiative, and the National Science Foundation under grant no. PHY-2011905. AL was supported by the Massachusetts Institute of Technology, the Packard Foundation and the National Science Foundation through grant no. PHY-1911298. GP was supported by the University of California, Berkeley; by the Department of Energy through DE-SC0019380 and DE-FOA-0002563; by AFOSR award FA9550-22-1-0098; and by an IBM Einstein Fellowship at the Institute for Advanced Study.

# A Properties of the min- and max-entropies

In this appendix, we collect the proofs of properties of the conditional min- and max-entropies used in the main text. While these proofs mostly follow those in [23], we generalize them where necessary to (finite) non-factor algebras, based on the definitions given in Section 2.2.

## A.1 Duality between min- and max-entropies

Here we prove the first part of Theorem 2.33. Our discussion closely follows that in [68], generalized to the algebraic setting. The theorem states that given a pure state $|\psi\rangle$ on a finite Hilbert space $\mathcal{H}$ and given the nested subalgebras $\mathcal{M}_B \subset \mathcal{M}_A \subset \mathcal{L}(\mathcal{H})$, with complementary traces on $\mathcal{M}_A$, $\mathcal{M}_{A'}$ and separately on $\mathcal{M}_B$, $\mathcal{M}_{B'}$, then

$$H_{\min}(A|B)_\psi = -H_{\max}(B'|A')_\psi \,. \tag{A.1}$$

To prove it, we first rewrite the min- and max-entropies in terms of the so-called sandwiched Renyi divergences, defined as follows.

**Definition A.1.** Let $\rho_A, \sigma_A$ be density matrices on algebra $\mathcal{M}_A$ with trace $\mathrm{tr}_A$. The *sandwiched quantum Renyi divergences* are

$$S_\alpha(\rho_A||\sigma_A) := \begin{cases} \frac{1}{\alpha-1} \log \mathrm{tr}_A \left[ \sigma_A^{\frac{1-\alpha}{2\alpha}} \rho_A \sigma_A^{\frac{1-\alpha}{2\alpha}} \right]^\alpha \,, & \text{if } \mathrm{supp}\, \rho_A \subseteq \mathrm{supp}\, \sigma_A \,, \\ \infty \,, & \text{else.} \end{cases} \tag{A.2}$$

Using these sandwiched Renyi divergences we can define Renyi conditional entropies for every $\alpha$.

**Definition A.2.** Let $\mathcal{M}_A \supseteq \mathcal{M}_B$ be algebras on $\mathcal{H}$ with traces $\mathrm{tr}_A$ and $\mathrm{tr}_B$, and let $|\psi\rangle \in \mathcal{H}$ be a pure state. Let $\rho_A$ be a density matrix on $\mathcal{M}_A$ for $|\psi\rangle$. The *conditional $\alpha$-entropy* is

$$H_\alpha(A|B)_\psi := \sup_{\sigma_B} -S_\alpha(\rho_A||\sigma_B) \,, \tag{A.3}$$

where the supremum is over density matrices $\sigma_B$ on $\mathcal{M}_B$ that are sub-normalized with respect to $\mathrm{tr}_B$, and we regard $\sigma_B$ as an operator in $\mathcal{M}_A$ via the natural inclusion $\mathcal{M}_B \subseteq \mathcal{M}_A$. The trace used in the definition of the sandwiched Renyi entropy is $\mathrm{tr}_A$.

**Definition A.3.** Let $\mathcal{M}$ be an algebra on $\mathcal{H}$ with trace tr, let $m \in \mathcal{M}$ be positive semi-definite, and let $p \in (0, \infty)$. The *Schatten p-norm* of $m$ is

$$\|m\|_p := (\mathrm{tr}[m^p])^{1/p} \, . \tag{A.4}$$

Note that when $p < 1$, $\|m\|_p$ is not technically a norm. The Schatten $p$-norms satisfy a useful relationship:

**Lemma A.4** (Lemma 12 of [68]). *Let $\mathcal{M}$ be an algebra on $\mathcal{H}$ with trace* tr. *Let $p, q \in \mathbb{R} \setminus \{0, 1\}$ satisfy $\frac{1}{p} + \frac{1}{q} = 1$. Then for any positive semi-definite $m \in \mathcal{M}$,*

$$\|m\|_p = \sup_{\substack{z \geq 0 \\ \mathrm{tr}\, z \leq 1}} \mathrm{tr}\left[ m z^{\frac{1}{q}} \right] \text{ if } p > 1 \, , \quad \text{and} \quad \|m\|_p = \inf_{\substack{z \geq 0 \\ \mathrm{tr}\, z \leq 1 \\ \mathrm{supp}\, z \supseteq \mathrm{supp}\, m}} \mathrm{tr}\left[ m z^{\frac{1}{q}} \right] \text{ if } p < 1 \, . \tag{A.5}$$

*Proof sketch.* For $p > 1$, this statement follows directly from the duality statement on $p$-norms:

$$\|m\|_p = \sup_{\|x\|_q \leq 1} |\mathrm{tr}[mx]| \, . \tag{A.6}$$

This duality statement follows in turn directly from Holder's inequality on the Schatten $p$-norms. Holder's inequality holds if the trace used to define the $p$-norm on the algebra is faithful, normal, and semi-finite, which ours is by assumption. For a proof of Holder's inequality that only uses these assumptions see [69].[22]

For $p < 1$, $\| \cdot \|_p$ is not a norm and so we cannot use Holder's inequality. Instead, we prove the statement following [68]. One can solve the optimization problem

$$\inf_{\substack{z \geq 0 \\ \mathrm{tr}\, z \leq 1 \\ \mathrm{supp}\, z \supseteq \mathrm{supp}\, m}} \mathrm{tr}\left[ m z^{\frac{1}{q}} \right] \, , \tag{A.7}$$

via Lagrange multipliers. Note that without loss of generality we can take $z$ to commute with $m$ by basic theorems in matrix analysis. Furthermore, we can take $z$ to be trace one, $\mathrm{tr}\, z = 1$. Otherwise, we could re-scale $z$ by its trace and get a lower value for $\mathrm{tr}[m z^{1/q}]$ since $q < 0$. Therefore, we can write a Lagrangian like

$$L = \mathrm{tr}\left[ m z^{1/q} \right] - \mu(\mathrm{tr}[z] - 1) \, , \tag{A.8}$$

with $\mu$ the Lagrange multiplier. Solving the equations for each component of $z$, we find the optimum $(z_*, \mu_*)$ satisfy

$$m z_*^{1/q - 1} = q \mu_* \mathbb{1} \, . \tag{A.9}$$

Remembering $\mathrm{tr}\, z_* = 1$, the trace of this equation tells us $q \mu_* = \mathrm{tr}(m^p)^{1/p}$. Moreover, (A.9) gives an optimum value of $L_* = q \mu_*$, which we recall equals $\mathrm{tr}[m z_*^{\frac{1}{q}}]$. Therefore $\mathrm{tr}(m^p)^{1/p} = q \mu_* = \mathrm{tr}(m z_*^{1/q})$, completing the argument. $\square$

Following [68], we now prove the following statement, which is stronger than (A.1).

---

[22]We thank Jon Sorce for pointing us to this reference.

**Theorem A.5** (Adapted from Theorem 10 of [68]). *Let $\mathcal{M}_B \subseteq \mathcal{M}_A \subseteq \mathcal{L}(\mathcal{H})$ be algebras on a Hilbert space $\mathcal{H}$ and denote their complements by $\mathcal{M}_{A'} := \mathcal{M}'_A$ and $\mathcal{M}_{B'} := \mathcal{M}'_B$. Let $|\psi\rangle$ be a pure state in $\mathcal{H}$ and let $\alpha, \beta \in (\frac{1}{2}, 1) \cup (1, \infty)$ be related by $\frac{1}{\alpha} + \frac{1}{\beta} = 2$. Then for traces on A and B which are complementary, as in definition 2.22, to those on $A'$ and $B'$ respectively, it holds*

$$H_\alpha(A|B)_\psi = -H_\beta(B'|A')_\psi \,. \tag{A.10}$$

*Proof.* Assuming $\operatorname{supp}\sigma_A \supseteq \operatorname{supp}\rho_A$, it holds that

$$S_\alpha(\rho_A||\sigma_A) = \frac{\alpha}{\alpha-1} \log \left\| \sigma_A^{\frac{1-\alpha}{2\alpha}} \rho_A \sigma_A^{\frac{1-\alpha}{2\alpha}} \right\|_\alpha = \frac{\alpha}{\alpha-1} \log \left\| \rho_A^{1/2} \sigma_A^{\frac{1-\alpha}{\alpha}} \rho_A^{1/2} \right\|_\alpha \,, \tag{A.11}$$

where the second equality uses the cyclicity of the trace. Applying lemma A.4, we have

$$S_\alpha(\rho_A||\sigma_A) = \begin{cases} \frac{\alpha}{\alpha-1} \log \sup_{\tau_A} \operatorname{tr}_A\left[ \rho_A^{1/2} \sigma_A^{\frac{1-\alpha}{\alpha}} \rho_A^{1/2} \tau_A^{\frac{\alpha-1}{\alpha}} \right], & \text{if } \alpha > 1, \\ \frac{\alpha}{\alpha-1} \log \inf_{\tau_A} \operatorname{tr}_A\left[ \rho_A^{1/2} \sigma_A^{\frac{1-\alpha}{\alpha}} \rho_A^{1/2} \tau_A^{\frac{\alpha-1}{\alpha}} \right], & \text{if } \alpha < 1, \end{cases} \tag{A.12}$$

where we define $\operatorname{tr}_A\left[ \rho_A^{1/2} \sigma_A^{\frac{1-\alpha}{\alpha}} \rho_A^{1/2} \tau_A^{\frac{\alpha-1}{\alpha}} \right] = +\infty$ if $\alpha < 1$ and $\operatorname{supp}\rho_A \nsubseteq \operatorname{supp}\tau_A$. It follows that

$$H_\alpha(A|B) = \begin{cases} \frac{\alpha}{1-\alpha} \log \inf_{\sigma_B} \sup_{\tau_A} \operatorname{tr}_A\left[ \rho_A^{1/2} \sigma_B^{\frac{1-\alpha}{\alpha}} \rho_A^{1/2} \tau_A^{\frac{\alpha-1}{\alpha}} \right], & \text{if } \alpha > 1, \\ \frac{\alpha}{1-\alpha} \log \sup_{\sigma_B} \inf_{\tau_A} \operatorname{tr}_A\left[ \rho_A^{1/2} \sigma_B^{\frac{1-\alpha}{\alpha}} \rho_A^{1/2} \tau_A^{\frac{\alpha-1}{\alpha}} \right], & \text{if } \alpha < 1. \end{cases} \tag{A.13}$$

Theorem 2.12 allows us to decompose $\mathcal{H} = \oplus_\alpha(\mathcal{H}_{A_\alpha} \otimes \mathcal{H}_{A'_\alpha})$, and write the purification of $\rho_A$ in terms of its Schmidt decomposition as

$$|\psi\rangle = \sum_\alpha \sum_i r_i^\alpha |\alpha; i\rangle_{A_\alpha} |\alpha; i\rangle_{A'_\alpha} \,. \tag{A.14}$$

From this we get a simple representation of $\rho_A$ of the form (32) that is diagonal within each $\alpha$-block. Using this representation, it is straightforward to find that

$$\operatorname{tr}_A\left[ \rho_A^{1/2} \sigma_B^{\frac{1-\alpha}{\alpha}} \rho_A^{1/2} \tau_A^{\frac{\alpha-1}{\alpha}} \right] = \langle\psi| \sigma_B^{\frac{1-\alpha}{\alpha}} \tau_{A'}^{\frac{\alpha-1}{\alpha}} |\psi\rangle \,, \tag{A.15}$$

where $\tau_{A'}$ is defined as the transpose of $\tau_A$ with respect to the Schmidt basis in equation (A.14) and so obeys $\rho_A^{1/2} \tau_A^n \rho_A^{-1/2} |\psi\rangle = \tau_{A'}^n |\psi\rangle$, as one can easily check. More explicitly, we can define the un-normalized pure state derived from $|\psi\rangle$

$$|\mathbb{1}\rangle = \sum_\alpha \sum_i |\alpha; i\rangle_{A_\alpha} |\alpha; i\rangle_{A'_\alpha} \,. \tag{A.16}$$

Then the operator $\tau_{A'}$ obeys the equation

$$\tau_A |\mathbb{1}\rangle = \tau_{A'} |\mathbb{1}\rangle \,. \tag{A.17}$$

By remarks 2.18 and 2.20, we can write $\operatorname{tr}_A$ and $\operatorname{tr}_{A'}$ in terms of the canonical trace and central operators

$$C^A = \sum_\alpha \frac{p_\alpha}{\dim A_\alpha} \operatorname{tr}_A[p_\alpha], \qquad C^{A'} = \sum_\alpha \frac{p_\alpha}{\dim A'_\alpha} \operatorname{tr}_{A'}[p_\alpha]. \tag{A.18}$$

In the optimization over $\tau_A$ in (A.13), it suffices to optimize only over $\tau_A$ which have the same support as $|\psi\rangle$, and similarly for $\sigma_B$. Therefore

$$\operatorname{tr}_A[\tau_A] = \langle\mathbb{1}|C^A \tau_A|\mathbb{1}\rangle = \langle\mathbb{1}|C^{A'}(C^{A'})^{-1} C^A \tau_{A'}|\mathbb{1}\rangle = \operatorname{tr}_{A'}\left[(C^{A'})^{-1} C^A \tau_{A'}\right] = \operatorname{tr}_{A'}[\tau_{A'}], \tag{A.19}$$

where in the last equality we used $C^A = C^{A'}$ because the traces are complementary.

This then allows us to write

$$H_\alpha(A|B)_\psi = \begin{cases} \frac{\alpha}{1-\alpha} \log \inf_{\sigma_B} \sup_{\tau_{A'}} \langle\psi| \sigma_B^{\frac{1-\alpha}{\alpha}} \tau_{A'}^{\frac{\alpha-1}{\alpha}} |\psi\rangle\,, & \text{if } \alpha > 1\,, \\ \frac{\alpha}{1-\alpha} \log \sup_{\sigma_B} \inf_{\tau_{A'}} \langle\psi| \sigma_B^{\frac{1-\alpha}{\alpha}} \tau_{A'}^{\frac{\alpha-1}{\alpha}} |\psi\rangle\,, & \text{if } \alpha < 1\,, \end{cases} \tag{A.20}$$

where the sup and inf are over $\tau_{A'}$ such that $\operatorname{tr}_{A'}[\tau_{A'}] \leq 1$. For $\alpha < 1$, this is concave in $\sigma_B$ and convex in $\tau_{A'}$. When $\alpha > 1$, the reverse is true: it is convex (concave) in $\sigma_B$ ($\tau_{A'}$). For such a function which is concave-convex in its two arguments, von Neumann's minimax theorem allows us to swap the inf and the sup.

Now, we could proceed by using similar manipulations to replace $\sigma_B$ with $\sigma_{B'}$. However, we are already done. Take (A.20) and plug in $A \to B'$, $B \to A'$, and use that for $\alpha, \beta$ with $\frac{1}{\alpha} + \frac{1}{\beta} = 2$ it holds that $\frac{\alpha}{\alpha-1} = -\frac{\beta}{\beta-1}$. Up to a sign this gives an identical expression on the right hand side, proving

$$H_\alpha(A|B)_\psi = -H_\beta(B'|A')_\psi\,. \tag{A.21}$$

$\square$

Finally, we relate these conditional $\alpha$-entropies to the min- and max-entropies used in the main text.

**Proposition A.6** (theorem 5 of [68]). From definition 2.28, it follows that

$$H_\infty(A|B) = H_{\min}(A|B)\,, \tag{A.22}$$
$$H_{1/2}(A|B) = H_{\max}(A|B)\,. \tag{A.23}$$

*Proof.* Equation (A.23) for $H_{\max}(A|B)$ follows directly from equations (A.2), (A.3) and (35). To prove (A.22), we first note that by equation (A.13) and the manipulations in (A.11), we have that

$$H_\infty(A|B) = -\log \inf_{\sigma_B} \sup_{\tau_A} \operatorname{tr}_A\left( \rho_A^{1/2} \sigma_B^{-1} \rho_A^{1/2} \tau_A \right) = -\log \inf_{\sigma_B} \sup_{\tau_A} \operatorname{tr}_A\left( \sigma_B^{-1/2} \rho_A \sigma_B^{-1/2} \tau_A \right)\,. \tag{A.24}$$

The supremum over $\tau_A$ is achieved by the $\tau_A$ projecting onto the largest eigenvalue of $\sigma_B^{-1/2} \rho_A \sigma_B^{-1/2}$. This log of the maximum eigenvalue can alternatively be written as

$$-\inf_{\sigma_B} \inf\{\lambda : \sigma_B^{-1/2} \rho_A \sigma_B^{-1/2} \leq e^\lambda\} = H_\infty(A|B)\,. \tag{A.25}$$

$\square$

**Remark A.7.** With equations (A.22) and (A.23), we can take the $\alpha \to \infty$ limit of Theorem A.5 to get

$$H_{\min}(A|B) = -H_{\max}(B'|A')\,. \tag{A.26}$$

## A.2 Smoothed duality

We now extend the results of the previous subsection to the *smoothed* one-shot entropies, proving

$$H_{\min}^\varepsilon(A|B)_\psi = -H_{\max}^\varepsilon(B'|A')_\psi\,. \tag{A.27}$$

**Definition A.8.** The $\varepsilon$-ball around $\rho$ is

$$\mathcal{B}^{\varepsilon}(\rho_A) := \{\rho_A^{\varepsilon} : P(|\rho\rangle, |\rho^{\varepsilon}\rangle)_A < \varepsilon\}, \tag{A.28}$$

with $P(|\rho\rangle, |\rho^{\varepsilon}\rangle)_A := P(\rho_A, \rho_A^{\varepsilon})$ and $P(\rho_A, \rho_A^{\varepsilon})$ from Definition 2.30.

**Remark A.9.** As before, the smoothed min- and max-entropies are related to limits of the smoothed $\alpha$-entropies as

$$H_{\min}^{\varepsilon}(A|B)_{\psi} = \max_{\rho^{\varepsilon} \in \mathcal{B}^{\varepsilon}(\rho)} H_{\infty}(A|B)_{\rho^{\varepsilon}}, \tag{A.29}$$

$$H_{\max}^{\varepsilon}(A|B)_{\psi} = \min_{\rho^{\varepsilon} \in \mathcal{B}^{\varepsilon}(\rho)} H_{1/2}(A|B)_{\rho^{\varepsilon}}. \tag{A.30}$$

**Remark A.10.** One can show that $P(|\Psi\rangle, |\Omega\rangle)_A$ defined above gives a good metric on the space of states and in particular obeys the triangle inequality. Furthermore, this metric is monotonic under inclusion so that if we have two nested algebras $\mathcal{M}_A \supset \mathcal{M}_B$, then

$$P(|\Psi\rangle, |\Omega\rangle)_A \geq P(|\Psi\rangle, |\Omega\rangle)_B. \tag{A.31}$$

We will need these properties below.

In what follows, we will use isometries to map algebras into larger algebras. Of course, when these algebras are non-factors, there is an ambiguity in each choice of trace. It will be important to define a special class of isometries which preserve the trace.

**Definition A.11** (Isometry between algebras)**.** Let the algebra $\mathcal{M}$ act both on $\mathcal{H}_1$ and on $\mathcal{H}_2$ and let the commutant algebras of $\mathcal{M}$ on those Hilbert space be $\mathcal{M}_1'$ and $\mathcal{M}_2'$ respectively. We say that an isometry $W : \mathcal{H}_1 \to \mathcal{H}_2$ maps $\mathcal{M}_1'$ into $\mathcal{M}_2'$ if

$$[W, \mathcal{M}] = 0. \tag{A.32}$$

**Lemma A.12.** *An isometry $W : \mathcal{H}_1 \to \mathcal{H}_2$ mapping $\mathcal{M}_1'$ into $\mathcal{M}_2'$ satisfies the following properties:*

1.  $W^{\dagger}W = \mathbb{1}$,
2.  $\Pi := WW^{\dagger} \in \mathcal{M}_2'$,
3.  $\Pi\mathcal{M}_2'\Pi = W\mathcal{M}_1'W^{\dagger}$.

*Proof.* Property 1 is the definition of an isometry. Property 2 follows directly from (A.32) and its conjugate since $\mathcal{M}^{\dagger} = \mathcal{M}$. To see Property 3, note that $[W\mathcal{M}_1'W^{\dagger}, \mathcal{M}] = 0$ and $\Pi W\mathcal{M}_1'W^{\dagger}\Pi = W\mathcal{M}_1'W^{\dagger}$. Hence $W\mathcal{M}_1'W^{\dagger} \subseteq \Pi\mathcal{M}_2'\Pi$. Similarly, $[W^{\dagger}\mathcal{M}_2'W, \mathcal{M}] = 0$ and hence $\Pi\mathcal{M}_2'\Pi \subseteq W\mathcal{M}_1'W^{\dagger}$.

$\square$

**Definition A.13** (Trace-preserving isometry)**.** We say that an isometry $W : \mathcal{H} \to \widetilde{\mathcal{H}}$ mapping $\mathcal{M}_A$ into $\mathcal{M}_{\widetilde{A}}$ is trace-preserving with respect to $\mathcal{M}_A$ if for all $m \in \mathcal{M}_A$,

$$\mathrm{tr}_{\widetilde{A}}[WmW^{\dagger}] = \mathrm{tr}_A[m]. \tag{A.33}$$

**Remark A.14.** A sufficient condition for an isometry to be trace-preserving is if for every minimal central projector $p_{\alpha}^A \in Z(\mathcal{M}_A)$ there exists a minimal central projector $p_{\alpha}^{\widetilde{A}}$ such that

$$V^{\dagger}p_{\alpha}^{\widetilde{A}}V = p_{\alpha}^A, \tag{A.34}$$

$$\mathrm{tr}_A(p_{\alpha}) = \mathrm{tr}_{\widetilde{A}}\left(\Pi p_{\alpha}^{\widetilde{A}}\Pi\right), \tag{A.35}$$

where $\Pi = VV^\dagger$. Note these conditions imply

$$V(\mathcal{H}_{A_\alpha} \otimes \mathcal{H}_{A'_\alpha}) = V p^A_\alpha \mathcal{H} = \Pi p^{\widetilde{A}}_\alpha \Pi \widetilde{\mathcal{H}} \subseteq p^{\widetilde{A}}_\alpha \widetilde{\mathcal{H}} =: \widetilde{\mathcal{H}}_{\widetilde{A}_\alpha} \otimes \mathcal{H}_{A'_\alpha}, \tag{A.36}$$

with the action of $V$ commuting with operators on $\mathcal{H}_{A'_\alpha}$. In the last equality we used the fact that $\mathcal{M}'_A \cong \mathcal{M}'_{\widetilde{A}}$ to identify $\mathcal{H}_{A'_\alpha}$ with $\mathcal{H}_{\widetilde{A}'_\alpha}$. In other words, within each $\alpha$-sector, $V$ embeds $\mathcal{H}_{A_\alpha}$ isometrically into $\widetilde{\mathcal{H}}_{\widetilde{A}_\alpha}$. In what follows, we will often need to construct embeddings with these properties.

**Remark A.15.** An important example of a trace-preserving isometry is the map $V_0 : \mathcal{H} \to \mathcal{H} \otimes \mathcal{H}_R$ defined by

$$V_0 |\psi\rangle = |\psi\rangle |0\rangle, \tag{A.37}$$

for some fixed state $|0\rangle \in \mathcal{H}_R$. This maps any algebra $\mathcal{M}_A$ acting on $\mathcal{H}$ into $\mathcal{M}_{\widetilde{A}} := \mathcal{M}_A \otimes \mathcal{L}(\mathcal{H}_R)$ and is trace-preserving for $\mathrm{tr}_{\widetilde{A}} = \mathrm{tr}_A \otimes \mathrm{Tr}_R$.

**Lemma A.16** (Adapted from Proposition 5.3 of [23]). *Let $V : \mathcal{H} \to \widetilde{\mathcal{H}}$ be a trace-preserving isometry mapping the algebra $\mathcal{M}_A$ into the algebra $\mathcal{M}_{\widetilde{A}}$. Let $\mathcal{M}_B \subseteq \mathcal{M}_A$ and $\mathcal{M}_{\widetilde{B}} \subseteq \mathcal{M}_{\widetilde{A}}$ be subalgebras such that $V^\dagger \mathcal{M}_{\widetilde{B}} V \subseteq \mathcal{M}_B$ with $\mathrm{tr}_B[V^\dagger O_{\widetilde{B}} V] = \mathrm{tr}_{\widetilde{B}}[O_{\widetilde{B}}]$ for all $O_{\widetilde{B}} \in \mathcal{M}_{\widetilde{B}}$. Finally let $\mathcal{T} : \mathcal{M}_B \to \mathcal{M}_{\widetilde{B}}$ be a trace-preserving completely positive superoperator such that $V O_B = \mathcal{T}(O_B) V$ for all $O_B \in \mathcal{M}_B$. The smoothed conditional min- and max-entropies are invariant under $V$:*

$$H^\varepsilon_{\min}(\widetilde{A}|\widetilde{B})_{\widetilde{\rho}} = H^\varepsilon_{\min}(A|B)_\rho, \tag{A.38}$$

$$H^\varepsilon_{\max}(\widetilde{A}|\widetilde{B})_{\widetilde{\rho}} = H^\varepsilon_{\max}(A|B)_\rho, \tag{A.39}$$

*where $\rho_A \in \mathcal{M}_A$ and $\widetilde{\rho}_{\widetilde{A}} := V \rho_A V^\dagger \in \mathcal{M}_{\widetilde{A}}$ are density matrices.*

*Proof.* We first prove this for $\varepsilon = 0$. By definition, for $\lambda = H_{\min}(A|B)_\rho$ there exists a $\sigma_B$ such that

$$\rho_A \le e^{-\lambda} \sigma_B, \tag{A.40}$$

and hence

$$V \rho_A V^\dagger \le e^{-\lambda} V \sigma_B V^\dagger = e^{-\lambda} \mathcal{T}(\sigma_B) \Pi = e^{-\lambda} \mathcal{T}(\sigma_B)^{1/2} \Pi \mathcal{T}(\sigma_B)^{1/2} \le e^{-\lambda} \mathcal{T}(\sigma_B). \tag{A.41}$$

By assumption, $V \rho_A V^\dagger$ and $\mathcal{T}(\sigma_B)$ are normalized density matrices on $\mathcal{M}_{\widetilde{A}}$ and $\mathcal{M}_{\widetilde{B}}$ respectively. Hence

$$H_{\min}(A|B)_\rho \le H_{\min}(\widetilde{A}|\widetilde{B})_{\widetilde{\rho}}. \tag{A.42}$$

Conversely, let $\tilde{\lambda} = H_{\min}(\widetilde{A}|\widetilde{B})_{\widetilde{\rho}}$. There exists a sub-normalized $\sigma_{\widetilde{B}}$ such that $\widetilde{\rho}_{\widetilde{A}} \le e^{-\tilde{\lambda}} \sigma_{\widetilde{B}}$. Conjugating by $V^\dagger$,

$$V^\dagger \widetilde{\rho}_{\widetilde{A}} V = \rho_A \le e^{-\tilde{\lambda}} V^\dagger \sigma_{\widetilde{B}} V. \tag{A.43}$$

By assumption, $\sigma_B = V^\dagger \sigma_{\widetilde{B}} V$ is a sub-normalized density matrix on $\mathcal{M}_B$. Hence $H_{\min}(A|B)_\rho \ge H_{\min}(\widetilde{A}|\widetilde{B})_{\widetilde{\rho}}$. The proof for the max-entropy works analogously.

To prove the statement for $\varepsilon > 0$, we will need the fact that to optimize the min- or max-entropies in the target algebra $\mathcal{M}_{\widetilde{A}}$, it is enough to consider density matrices in $\Pi \mathcal{M}_{\widetilde{A}} \Pi = V \mathcal{M}_A V^\dagger$, within $\mathcal{B}^\varepsilon(V \rho_A V^\dagger)$. To see this, first note that

$$\max_{\Pi \widetilde{\omega} \Pi \in \mathcal{B}^\varepsilon(V \rho V^\dagger)} H_{\min}(\widetilde{A}|\widetilde{B})_{\Pi \widetilde{\omega} \Pi} \le H^\varepsilon_{\min}(\widetilde{A}|\widetilde{B})_{V \rho V^\dagger}, \tag{A.44}$$

because the restriction to $\Pi \mathcal{M}_{\widetilde{A}} \Pi$ can only decrease the maximum. Conversely, note that for the density matrix $\widetilde{\rho}_* \in \mathcal{M}_{\widetilde{A}}$ such that $H_{\min}^\varepsilon(\widetilde{A}|\widetilde{B})_{V\rho V^\dagger} = H_{\min}(\widetilde{A}|\widetilde{B})_{\widetilde{\rho}_*}$, it *decreases* the min-entropy if $\widetilde{\rho}_*$ has support in a subspace orthogonal to $\Pi$. Indeed, $H_{\min}(\widetilde{A}|\widetilde{B})_{\widetilde{\rho}_*} \leq H_{\min}(\widetilde{A}|\widetilde{B})_{\Pi\widetilde{\rho}_*\Pi}$, which follows from $\Pi\widetilde{\rho}_*\Pi \leq \widetilde{\rho}_*$. Furthermore, by monotonicity of the purified distance under projections we know that $\Pi\widetilde{\rho}_*\Pi \in \mathcal{B}^\varepsilon(\Pi V\rho_A V^\dagger \Pi)$, and moreover we know that $\Pi V = V$. Therefore

$$\max_{\Pi\widetilde{\omega}\Pi \in \mathcal{B}^\varepsilon(V\rho V^\dagger)} H_{\min}(\widetilde{A}|\widetilde{B})_{\Pi\widetilde{\omega}\Pi} \geq H_{\min}^\varepsilon(\widetilde{A}|\widetilde{B})_{V\rho V^\dagger}. \tag{A.45}$$

In particular, $H_{\min}^\varepsilon(\widetilde{A}|\widetilde{B})_{V\rho V^\dagger} = H_{\min}(\widetilde{A}|\widetilde{B})_{\Pi\widetilde{\rho}_*\Pi}$. The analogous statement also holds for the max-entropy.

Now let $\rho_* \in \mathcal{M}_A$ be such that $H_{\min}^\varepsilon(A|B)_\rho = H_{\min}(A|B)_{\rho_*}$ and $\Pi\widetilde{\rho}_*\Pi$ be defined as above. Then since the isometry is trace preserving, we have that both $V\rho_* V^\dagger \in \mathcal{B}^\varepsilon(V\rho_A V^\dagger)$ and $V^\dagger\widetilde{\rho}_* V \in \mathcal{B}^\varepsilon(\rho_A)$. Therefore,

$$H_{\min}^\varepsilon(A|B)_\rho = H_{\min}(A|B)_{\rho_*} = H_{\min}(\widetilde{A}|\widetilde{B})_{V\rho_* V^\dagger} \leq H_{\min}^\varepsilon(\widetilde{A}|\widetilde{B})_{V\rho V^\dagger} \tag{A.46}$$

$$= H_{\min}(\widetilde{A}|\widetilde{B})_{\Pi\widetilde{\rho}_*\Pi} = H_{\min}(A|B)_{V^\dagger\widetilde{\rho}_* V} \leq H_{\min}^\varepsilon(A|B)_\rho. \tag{A.47}$$

The proof for the max-entropy works analogously. $\qquad\square$

**Remark A.17.** Lemma A.16 is very general. We will be primarily interested in two special cases, both related to the isometry $V_0$ from Remark A.15. In both cases, we have $\mathcal{M}_{\widetilde{A}} := \mathcal{M}_A \otimes \mathcal{L}(\mathcal{H}_R)$. In the first case we have $\mathcal{M}_{\widetilde{B}} := \mathcal{M}_B$ and $\mathcal{T}$ is the identity channel, $\mathcal{T}(\rho_B) = \rho_B \otimes \mathbb{1}_R$. In the second case we have $\mathcal{M}_{\widetilde{B}} := \mathcal{M}_B \otimes \mathcal{L}(\mathcal{H}_R)$ and $\mathcal{T}(\rho_B) = \rho_B \otimes |0\rangle\langle 0|$.

**Lemma A.18** (Uhlmann's theorem). *Let $\rho$ and $\sigma$ be positive operators. For any purification $|\phi\rangle$ of $\rho$,*

$$F(\rho, \sigma) = \max_{|\psi\rangle} |\langle\phi|\psi\rangle|, \tag{A.48}$$

*where the maximum is taken over all purifications $|\psi\rangle$ of $\sigma$.*

For a proof of Uhlmann's theorem that applies to general algebras, see [70].

**Theorem A.19** (Adapted from theorem 5.4 of [23]). *Let $|\psi\rangle \in \mathcal{H}$ be a pure state. Assuming that the traces on the algebras $\mathcal{M}_A, \mathcal{M}_B \subset \mathcal{L}(\mathcal{H})$ are complementary to those on $\mathcal{M}_A', \mathcal{M}_B'$ respectively, then the smoothed conditional min- and max-entropies obey the duality statement*

$$H_{\min}^\varepsilon(A|B)_\psi = -H_{\max}^\varepsilon(B'|A')_\psi. \tag{A.49}$$

*Proof.* We would like to write

$$H_{\min}^\varepsilon(A|B)_\psi = H_{\min}(A|B)_{\rho^\varepsilon} = -H_{\max}(B'|A')_{\rho^\varepsilon} \leq -H_{\max}^\varepsilon(B'|A')_\rho, \tag{A.50}$$

and then obtain the opposite inequality from analogous manipulations starting from $H_{\max}^\varepsilon(B'|A')$. However, the second equality is too quick.

We have not proven that given a density matrix $\rho_A^\varepsilon$, there is a density matrix $\rho_{B'}^\varepsilon \in \mathcal{B}^\varepsilon(\rho_{B'})$ that purifies $\mathrm{tr}_{A\to B}[\rho_A^\varepsilon]$.

Let $V_0$ be defined as in Remark A.15 with $\mathcal{M}_{\widetilde{B}} := \mathcal{M}_B \otimes \mathcal{L}(\mathcal{H}_R)$ as in case 2 of Remark A.17. We wish to prove that given a $\rho_A^\varepsilon \in \mathcal{B}^\varepsilon(\rho_A)$, there exists a $\rho_{\widetilde{B}'}^\varepsilon \in \mathcal{B}^\varepsilon(V_0\rho_{B'}V_0^\dagger)$ that purifies $\mathrm{tr}_{A\to B}[\rho_A^\varepsilon]$. By Uhlmann's theorem, for any $\rho_A^\varepsilon$, there is a purification $|\widetilde{\psi}\rangle \in \widetilde{\mathcal{H}} = \mathcal{H} \otimes \mathcal{H}_R$ for sufficiently large $\mathcal{H}_R$ such that $P(\rho_A, \rho_A^\varepsilon) = |\langle\widetilde{\psi}|V|\psi\rangle|$. Now let $\rho_{\widetilde{B}'}^\varepsilon$ be the density matrix of $|\widetilde{\psi}\rangle$ on $\mathcal{M}_{\widetilde{B}'}$. Tracing out $\mathcal{M}_B$ can only decrease the purified distance, and hence $P(\rho_A, \rho_A^\varepsilon) \geq P(V\rho_{B'}V^\dagger, \rho_{\widetilde{B}'}^\varepsilon)$. Therefore indeed $\rho_{\widetilde{B}'}^\varepsilon \in \mathcal{B}^\varepsilon(V_0\rho_{B'}V_0^\dagger)$.

Now we can run the argument. Let $\rho_A^\varepsilon$ optimize the min-entropy on $A$. By Lemma A.16 and Theorem A.5 we have that

$$H_{\min}^\varepsilon(A|B)_\psi = H_{\min}(A|B)_{\rho^\varepsilon} = -H_{\max}(\widetilde{B}'|\widetilde{A}')_{\rho^\varepsilon} \leq -H_{\max}^\varepsilon(\widetilde{B}'|\widetilde{A}')_{V\rho V^\dagger} = -H_{\max}^\varepsilon(B'|A')_\rho. \quad (A.51)$$

Conversely, we have

$$H_{\max}^\varepsilon(B'|A')_\psi = H_{\max}(B'|A')_{\rho^\varepsilon} = -H_{\min}(\widetilde{B}|\widetilde{A})_{\rho^\varepsilon} \geq -H_{\min}^\varepsilon(\widetilde{B}|\widetilde{A})_{V\rho V^\dagger} = -H_{\min}^\varepsilon(B|A)_\rho. \quad (A.52)$$

$\square$

## A.3 Quantum asymptotic equipartition principle

In this subsection, we present the proof of the quantum asymptotic equipartition principle (QAEP), following [23, 71]. During the preparation of this manuscript, a proof of an asymptotic equipartition principle for max-relative entropies in any von Neumann algebras (including infinite-dimensional ones) was independently given in [33]. Let $\mathcal{M}_A \supseteq \mathcal{M}_B$ be algebras on Hilbert space $\mathcal{H}$ with trace tr and $\mathrm{tr}_B$ respectively. Let $\Psi_A \in \mathcal{M}_A$ be a normalized density matrix and $\Psi_B = \mathrm{tr}_{A\to B}\Psi_A$. The operators $\sigma, \rho \in \mathcal{M}_A$ are positive and we assume $\mathrm{tr}(\rho) \leq 1$. Finally, we let $0 < \varepsilon < 1$.

**Theorem A.20** (QAEP). *It holds that*

$$\lim_{n\to\infty} \frac{1}{n} H_{\min}^\varepsilon(A^n|B^n)_{\Psi^{\otimes n}} = S(A|B)_\Psi = \lim_{n\to\infty} \frac{1}{n} H_{\max}^\varepsilon(A^n|B^n)_{\Psi^{\otimes n}}. \quad (A.53)$$

We begin with some preliminary definitions and lemmas.

**Definition A.21.** The smoothed version of the $\alpha$-Renyi entropies defined in A.2 are

$$S_\alpha^\varepsilon(\rho||\sigma) = \begin{cases} \inf_{\widetilde{\rho}\in\mathcal{B}^\varepsilon(\rho)} S_\alpha(\widetilde{\rho}||\sigma), & \text{if } \alpha > 1, \\ \sup_{\widetilde{\rho}\in\mathcal{B}^\varepsilon(\rho)} S_\alpha(\widetilde{\rho}||\sigma), & \text{if } \alpha < 1, \end{cases} \quad (A.54)$$

where again $\mathcal{B}^\varepsilon(\rho)$ is as in definition A.8.

**Remark A.22.** By using equation (A.12) and an argument similar to that in Proposition A.6, we obtain

$$S_\infty(\rho||\sigma) = \inf\left\{\lambda : \rho \leq e^\lambda \sigma\right\}. \quad (A.55)$$

**Proposition A.23.** It holds that

$$S(A|B)_\Psi = -S(\Psi_A||\Psi_B), \quad (A.56)$$
$$H_{\min}^\varepsilon(A|B)_\Psi \geq -S_\infty^\varepsilon(\Psi_A||\Psi_B). \quad (A.57)$$

*Proof.* To derive the first equality, note that

$$S(\Psi_A||\Psi_B)_\Psi := \mathrm{tr}(\Psi_A \log \Psi_A) - \mathrm{tr}(\Psi_A \log \Psi_B) = S(B) - S(A),$$

the last equality following from the definition of the density matrix on $\mathcal{M}_A$ and $\mathcal{M}_B$.

To derive the second inequality, from definition A.1 we have

$$H_{\min}(A|B)_\Psi := -\min_{\widetilde{\Psi}_B} \inf\{\lambda : \Psi_A \leq e^\lambda \widetilde{\Psi}_B\} = \max_{\widetilde{\Psi}_B} \sup\{\lambda : \Psi_A \leq e^{-\lambda} \widetilde{\Psi}_B\}$$
$$\geq \sup\{\lambda : \Psi_A \leq e^{-\lambda}\Psi_B\} = -S_\infty(\Psi_A||\Psi_B).$$

The inequality continues to hold under smoothing. $\square$

**Lemma A.24** (lemma 6.1 of [23]). *Let $\lambda \leq S_\infty(\rho\|\sigma)$. Then*

$$S_\infty^\varepsilon(\rho\|\sigma) \leq \lambda, \quad where \quad \varepsilon = \sqrt{2\operatorname{tr}(\Delta) - \operatorname{tr}(\Delta)^2}, \quad and \quad \Delta = \{\rho - e^\lambda \sigma\}_+, \tag{A.58}$$

*where for Hermitian operator $X$, $\{X\}_+$ is the positive operator defined by setting all negative eigenvalues to zero.*

*Proof.* The strategy is to choose $\tilde{\rho}$, bound $S_\infty(\tilde{\rho}\|\sigma)$ and then show that $\tilde{\rho} \in \mathcal{B}^\varepsilon(\rho)$. This then bounds the smoothed-entropy. Define $\Lambda := e^\lambda \sigma$ and also

$$\tilde{\rho} := G\rho G^\dagger, \quad where \quad G := \Lambda^{\frac{1}{2}}(\Lambda + \Delta)^{-\frac{1}{2}}, \tag{A.59}$$

using the generalized inverse. From the definition of $\Delta$ we have $\rho \leq \Lambda + \Delta$ and therefore $\tilde{\rho} \leq \Lambda$ and so $S_\infty(\tilde{\rho}\|\sigma) \leq \lambda$.

Now let $|\psi\rangle$ be a purification of $\rho$. Then $G|\psi\rangle$ is a purification of $\tilde{\rho}$. Using Uhlmann's theorem for the generalized fidelity,

$$F(\tilde{\rho}, \rho) \geq |\langle\psi|G|\psi\rangle| + \sqrt{(1 - \operatorname{tr}[\rho])(1 - \operatorname{tr}[\tilde{\rho}])} \tag{A.60}$$

$$\geq \operatorname{Re}(\operatorname{tr}[\rho G]) + 1 - \operatorname{tr}[\rho] \tag{A.61}$$

$$\geq 1 - \operatorname{tr}[(\mathbb{1} - \bar{G})\rho], \tag{A.62}$$

where we have introduced $\bar{G} := (G + G^\dagger)/2$ and in going from the first to second line used that $\operatorname{tr}[\tilde{\rho}] \leq \operatorname{tr}[\rho]$. It also holds that

$$G^\dagger G = (\Lambda + \Delta)^{-\frac{1}{2}}\Lambda(\Lambda + \Delta)^{-\frac{1}{2}} \leq \mathbb{1}, \tag{A.63}$$

where the final inequality follows from multiplying $\Lambda \leq \Lambda + \Delta$ with $(\Lambda + \Delta)^{-\frac{1}{2}}$ from the left and right. It follows that $\bar{G} \leq \mathbb{1}$ by the triangle inequality. Moreover,

$$\operatorname{tr}[(\mathbb{1} - \bar{G})\rho] \leq \operatorname{tr}[\Lambda + \Delta] - \operatorname{tr}[\bar{G}(\Lambda + \Delta)]$$
$$= \operatorname{tr}[\Lambda + \Delta] - \operatorname{tr}\left[(\Lambda + \Delta)^{\frac{1}{2}}\Lambda^{\frac{1}{2}}\right] \tag{A.64}$$
$$\leq \operatorname{tr}[\Delta],$$

where we have used $\rho \leq \Lambda + \Delta$ and $\sqrt{\Lambda + \Delta} \geq \sqrt{\Lambda}$, the latter following from the monotonicity of the square root. Finally we can combine all of this to bound the purified distance

$$P(\tilde{\rho}, \rho) := \sqrt{1 - F^2(\tilde{\rho}, \rho)} \leq \sqrt{1 - (1 - \operatorname{tr}[\Delta])^2} = \sqrt{2\operatorname{tr}[\Delta] - \operatorname{tr}[\Delta]^2}. \tag{A.65}$$

This confirms $\tilde{\rho} \in \mathcal{B}^\varepsilon(\rho)$ and so, by use of definition A.21, concludes the proof. $\qquad\square$

**Definition A.25.** When $\operatorname{supp}(\rho) \subseteq \operatorname{supp}(\sigma)$, the *$\alpha$-Petz relative entropy* is

$$D_\alpha(\rho\|\sigma) := \frac{1}{\alpha - 1}\log\operatorname{tr}\left[\rho^\alpha \sigma^{1-\alpha}\right], \tag{A.66}$$

where for $\alpha > 1$ we use the generalized inverse of $\sigma$. When $\operatorname{supp}(\rho) \nsubseteq \operatorname{supp}(\sigma)$, it is defined to equal $\infty$.

**Lemma A.26** (proposition 6.2 of [23]). *Let $\alpha \in (1, 2]$. Then*

$$S_\infty^\varepsilon(\rho\|\sigma) \leq D_\alpha(\rho\|\sigma) + \frac{g(\varepsilon)}{\alpha - 1}, \quad where \quad g(\varepsilon) = \log\frac{1}{1 - \sqrt{1 - \varepsilon^2}}. \tag{A.67}$$

*Proof.* Suppose $\mathrm{supp}(\rho) \nsubseteq \mathrm{supp}(\sigma)$. Then $D_\alpha(\rho||\sigma)$ diverges to $+\infty$ and the inequality holds trivially.

Now suppose $\mathrm{supp}(\rho) \subseteq \mathrm{supp}(\sigma)$. Then for the sake of this proof we can assume $\sigma$ is invertible. (More precisely, we can define an isometry $\mathcal{H}' \to \mathcal{H}$ that maps $\sigma' \mapsto \sigma$ and $\rho' \mapsto \rho$ such that $\sigma'$ has full support, and the entropies are invariant under such an isometry.)

From lemma A.24 we have $S_\infty^\varepsilon(\rho||\sigma) \leq \lambda$ for some $\lambda$. Introduce the operator $X := \rho - e^\lambda \sigma$ with eigenbasis $\{|e_i\rangle\}$ for $i \in S$. The set $S^+ \subseteq S$ is the indices $i$ corresponding to positive eigenvalues of $X$. Therefore $P^+ := \sum_{i\in S^+} |e_i\rangle\langle e_i|$ is the projector on the positive eigenspace of $X$ and $P^+ X P^+ = \Delta$ as defined in lemma A.24. Let $r_i := \langle e_i|\rho|e_i\rangle \geq 0$ and $s_i := \langle e_i|\sigma|e_i\rangle > 0$. Now, the trace on the algebra is related to the canonical trace via the action of a central operator. In particular, we can write

$$\mathrm{tr}[\,\cdot\,] = \mathrm{tr}_{\mathrm{can}}[C\,\cdot\,]. \tag{A.68}$$

We define $C_i := \langle e_i|C|e_i\rangle$. Note that $C_i \geq 0$ by assumption.

Using this, we note that

$$\forall i \in S^+ : r_i - e^\lambda s_i \geq 0, \quad \text{thus} \quad \frac{r_i}{s_i}e^{-\lambda} \geq 1. \tag{A.69}$$

Then for any $\alpha \in (1,2]$ we bound $\mathrm{tr}(\Delta) = 1 - \sqrt{1-\varepsilon^2}$ with

$$
\begin{aligned}
1 - \sqrt{1-\varepsilon^2} = \mathrm{tr}(\Delta) &= \sum_{i\in S^+} C_i\left(r_i - e^\lambda s_i\right) \leq \sum_{i\in S^+} C_i r_i \\
&\leq \sum_{i\in S^+} C_i r_i \left(\frac{r_i}{s_i}e^{-\lambda}\right)^{\alpha-1} \leq e^{\lambda(1-\alpha)}\sum_{i\in S} C_i r_i^\alpha s_i^{1-\alpha}.
\end{aligned} \tag{A.70}
$$

We then take the logarithm and divide by $\alpha - 1 > 0$ to get

$$\lambda \leq \frac{1}{\alpha-1}\log\sum_{i\in S} C_i r_i^\alpha s_i^{1-\alpha} + \frac{1}{\alpha-1}\log\frac{1}{1-\sqrt{1-\varepsilon^2}}. \tag{A.71}$$

Finally, define the completely-positive trace-preserving map $\mathcal{N} : \omega \mapsto \sum_{i\in S} |e_i\rangle\langle e_i|\omega|e_i\rangle\langle e_i|$, and use the monotonicity of the Petz relative entropies [70] to obtain

$$D_\alpha(\rho||\sigma) \geq D_\alpha(\mathcal{N}(\rho)||\mathcal{N}(\sigma)) = \frac{1}{\alpha-1}\log\sum_{i\in S} C_i r_i^\alpha s_i^{1-\alpha}. \tag{A.72}$$

Combining this with (A.71) and the lowerbound on $\lambda$ from lemma A.24 concludes the proof. $\square$

The following quantity will help us describe how fast the $\alpha$-entropies converge to the von Neumann entropy.

**Definition A.27.** We define the *$\alpha$-entropy convergence parameter*

$$\Upsilon(\rho||\sigma) := e^{\frac{1}{2}D_{\frac{3}{2}}(\rho||\sigma)} + e^{-\frac{1}{2}D_{\frac{1}{2}}(\rho||\sigma)} + 1. \tag{A.73}$$

We now bound the $\alpha$-entropies for $\alpha \approx 1$.

**Lemma A.28** (lemma 6.3 of [23]). *Let* $\mathrm{tr}(\rho) = 1$ *and let* $1 < \alpha < 1 + \frac{\log 3}{4\log v}$ *where* $v := \Upsilon(\rho||\sigma)$. *Then*

$$D_\alpha(\rho||\sigma) < S(\rho||\sigma) + 4(\alpha-1)(\log v)^2. \tag{A.74}$$

*Proof.* As in the proof of lemma A.26, we will assume without loss of generality that $\sigma$ is invertible. Let $\{|i\rangle\}$ be an orthonormal basis for $\mathcal{H}$, and let $|\text{MAX}\rangle = \sum_i |i\rangle \otimes |i\rangle$ be an unnormalized maximally entangled state on $\mathcal{H} \otimes \mathcal{H}$, and define $|\phi\rangle := \sqrt{C\rho}\,|\text{MAX}\rangle$, where $C$ is the central operator such that $\text{tr}[\cdot] = \text{tr}_{\text{can}}[C\cdot]$. Let $\beta := \alpha - 1$ and $X := \rho \otimes (\sigma^{-1})^T$. We first approximate $D_\alpha$ for small $\beta > 0$.

$$D_\alpha(\rho||\sigma) = \frac{1}{\beta}\log\langle\phi|X^\beta|\phi\rangle \leq \frac{1}{\beta}(\langle\phi|X^\beta|\phi\rangle - 1), \tag{A.75}$$

where we have used that $\log x \leq x - 1$ for $x > 0$. Now define $r_\beta(t) := t^\beta - \beta \log t - 1$. Then

$$D_\alpha(\rho||\sigma) \leq \frac{1}{\beta}(\langle\phi|r_\beta(X)|\phi\rangle - 1 + \text{tr}(\rho) + \beta\langle\phi|\log X|\phi\rangle)$$
$$\leq S(\rho||\sigma) + \frac{1}{\beta}\langle\phi|r_\beta(X)|\phi\rangle. \tag{A.76}$$

Now we continue to simplify by defining

$$s_\beta(t) := 2(\cosh(\beta\log t) - 1) \geq r_\beta(t). \tag{A.77}$$

One can confirm that $s_\beta(t)$ is monotonically increasing for $t \geq 1$ and concave in $t$ for $\beta \leq 1/2$ and $t \in [3, \infty)$. It also holds that $s_\beta(t) = s_\beta(1/t)$ and $s_\beta(t^2) = s_{2\beta}(t)$. Thus we can bound

$$s_\beta(t) \leq s_\beta\left(t + \frac{1}{t} + 2\right) = s_{2\beta}\left(\sqrt{t} + \frac{1}{\sqrt{t}}\right) \leq s_{2\beta}\left(\sqrt{t} + \frac{1}{\sqrt{t}} + 1\right). \tag{A.78}$$

Next use that $\sqrt{X} + 1/\sqrt{X} + \mathbb{1}$ has all eigenvalues in $[3, \infty)$ and that $2\beta < \frac{\log 3}{2\log v} \leq 1/2$ to get

$$\langle\phi|s_\beta(X)|\phi\rangle \leq \langle\phi|s_{2\beta}(\sqrt{X} + \frac{1}{\sqrt{X}} + \mathbb{1})|\phi\rangle \leq s_{2\beta}(v), \tag{A.79}$$

where in the last inequality we have used concavity and $v = \langle\phi|(\sqrt{X} + 1/\sqrt{X} + \mathbb{1})|\phi\rangle$. Finally, we use that $s_\beta(t) \leq \beta^2(\log t)^2\cosh(\beta t)$ to write

$$s_{2\beta}(v) \leq 4\beta^2(\log v)^2\cosh(2\beta\log v) < 4\beta(\log v)^2. \tag{A.80}$$

Combining this with (A.79) and (A.77) and plugging into (A.76) completes the proof. $\qquad\square$

**Theorem A.29** (theorem 6.4 of [23]). *Let* $\text{tr}(\rho) = 1$ *and* $v = \Upsilon(\rho||\sigma)$. *Then for any* $n > 10g(\varepsilon)/3$, *the operators* $\rho^{\otimes n}$ *and* $\sigma^{\otimes n}$ *satisfy*

$$\frac{1}{n}S_\infty^\varepsilon(\rho^{\otimes n}||\sigma^{\otimes n}) \leq S(\rho||\sigma) + \frac{\delta(\varepsilon, v)}{\sqrt{n}}, \quad \text{where} \quad \delta(\varepsilon, v) = 4\sqrt{g(\varepsilon)}\log v, \tag{A.81}$$

*and* $g(\varepsilon) = -\log(1 - \sqrt{1 - \varepsilon^2})$.

*Proof.* Let $\alpha := 1 + 1/2\mu\sqrt{n}$, for some $\mu$ we will optimize later. Using lemmas A.26 and A.28, we have

$$\frac{1}{n}S_\infty^\varepsilon(\rho^{\otimes n}||\sigma^{\otimes n}) \leq \frac{1}{n}D_\alpha(\rho^{\otimes n}||\sigma^{\otimes n}) + \frac{g(\varepsilon)}{n(\alpha - 1)}$$
$$= D_\alpha(\rho||\sigma) + \frac{2\mu}{\sqrt{n}}g(\varepsilon)$$
$$\leq S(\rho||\sigma) + \frac{2}{\sqrt{n}}\left(\frac{(\log v)^2}{\mu} + \mu g(\varepsilon)\right). \tag{A.82}$$

For the best bound, we would like to choose $\mu$ to minimize $(\log v)^2/\mu + \mu g(\varepsilon)$, but we must keep in mind that our use of lemma A.28 limits $1 < \alpha < 1 + \log(3)/4\log(v)$, restricting our choice of $\mu$ for any given $n$. Fortunately, the optimum can be achieved for large enough $n$, in particular:

$$\mu_* = \sqrt{\frac{(\log v)^2}{g(\varepsilon)}}, \quad \text{for} \quad n \geq \frac{10}{3}\frac{(\log v)^2}{\mu_*^2} = \frac{10}{3}g(\varepsilon), \tag{A.83}$$

where we have used that $\sqrt{6/5} < \log 3$. Substituting this optimum into the previous inequality completes the proof. $\qquad\square$

**Theorem A.30** (corollary 6.5 of [23]). *It holds that*

$$\frac{1}{n}H_{\min}^{\varepsilon}(A^n|B^n)_{\Psi^{\otimes n}} \geq S(A|B)_{\Psi} - \frac{\delta(\varepsilon, v)}{\sqrt{n}}, \tag{A.84}$$

$$\frac{1}{n}H_{\max}^{\varepsilon}(A^n|B^n)_{\Psi^{\otimes n}} \leq S(A|B)_{\Psi} + \frac{\delta(\varepsilon, v)}{\sqrt{n}}, \tag{A.85}$$

*where $\delta(\varepsilon, v)$ is as defined in Theorem A.29 and $v = \Upsilon(\Psi_A \| \Psi_B)$.*

*Proof.* From Proposition A.23 and Theorem A.29, it follows that

$$\begin{aligned}
\frac{1}{n}H_{\min}^{\varepsilon}(A^n|B^n)_{\Psi^{\otimes n}} &\geq -\frac{1}{n}S_{\infty}^{\varepsilon}(\Psi_A^{\otimes n} \| \Psi_B^{\otimes n}) \\
&\geq -S(\Psi_A \| \Psi_B) - \frac{\delta(\varepsilon, v)}{\sqrt{n}} \\
&= S(A|B)_{\Psi} - \frac{\delta(\varepsilon, v)}{\sqrt{n}}.
\end{aligned} \tag{A.86}$$

From duality it holds that $H_{\min}^{\varepsilon}(A^n|B^n)_{\Psi^{\otimes n}} = -H_{\max}^{\varepsilon}(B'^n|A'^n)_{\Psi^{\otimes n}}$ and also that $S(B'|A')_{\Psi} = -S(A|B)_{\Psi}$. Therefore

$$\frac{1}{n}H_{\max}^{\varepsilon}(B'^n|A'^n)_{\Psi^{\otimes n}} \leq S(B'|A')_{\Psi} + \frac{\delta(\varepsilon, v)}{\sqrt{n}}. \tag{A.87}$$

$\qquad\square$

**Corollary A.31** (QAEP, direct). *It holds that*

$$\lim_{n\to\infty} \frac{1}{n}H_{\min}^{\varepsilon}(A^n|B^n)_{\Psi^{\otimes n}} \geq S(A|B)_{\Psi} \geq \lim_{n\to\infty} \frac{1}{n}H_{\max}^{\varepsilon}(A^n|B^n)_{\Psi^{\otimes n}}. \tag{A.88}$$

Now we need to prove the converse direction. Essentially this will follow from $H_{\min}(A|B) \leq S(A|B) \leq H_{\max}(A|B)$. However, we would get too weak a bound if we naively smoothed to $\varepsilon > 0$ using the continuity of the conditional entropy. At the end we would also need to take the limit $\varepsilon \to 0$. We get a stronger bound as follows.

First, note that so far the smoothing optimizes over sub-normalized states. It will be convenient to isometrically extend the algebras such that the optimizing density matrix is normalized.

**Lemma A.32** (Adapted from Lemma 5.2 of [23]). *Given any density matrix $\rho_A \in \mathcal{M}_A \supset \mathcal{M}_B$, there exists an isometry $V: \mathcal{H} \to \widetilde{\mathcal{H}}$, satisfying the conditions of Lemma A.16, along with density matrices $\hat{\rho}_{\widetilde{A},\min}, \hat{\rho}_{\widetilde{A},\max} \in \mathcal{B}^{\varepsilon}(V\rho_A V^{\dagger})$, normalized on $\mathcal{M}_{\widetilde{A}}$, such that*

$$H_{\min}^{\varepsilon}(A|B)_{\rho_A} = H_{\min}(\widetilde{A}|\widetilde{B})_{\hat{\rho}_{\widetilde{A},\min}}, \quad H_{\max}^{\varepsilon}(A|B)_{\rho_A} = H_{\max}(\widetilde{A}|\widetilde{B})_{\hat{\rho}_{\widetilde{A},\max}}. \tag{A.89}$$

*Proof.* Let $V_0 : \mathcal{H} \to \mathcal{H} \otimes \mathcal{H}_{R_1}$ be defined as in Remark A.15 with $\mathcal{M}_{\widetilde{B}} = \mathcal{M}_B$ as in case 1 of Remark A.17.

Now given a sub-normalized density matrix $\rho_A^\varepsilon \in \mathcal{B}^\varepsilon(\rho_A)$ which optimizes the smoothed min-entropy, there exists a sub-normalized density matrix $\sigma_B$ such that

$$\rho_A^\varepsilon \leq e^{-\lambda} \sigma_B, \tag{A.90}$$

with $\lambda = H_{\min}^\varepsilon(A|B)_\rho$. We can then define the state $\hat{\rho}_{\widetilde{A}}$ as

$$\hat{\rho}_{\widetilde{A}} := \rho_A^\varepsilon \otimes |0\rangle\langle 0|_{R_1} + \frac{1 - \mathrm{tr}_A(\rho_A^\varepsilon)}{(\dim \mathcal{H}_{R_1} - 1) \, \mathrm{tr}_A[\sigma_B]} \sigma_B \otimes (\mathbb{1}_{R_1} - |0\rangle\langle 0|_{R_1}), \tag{A.91}$$

which is by construction normalized, $\mathrm{tr}_{\widetilde{A}} \hat{\rho}_{\widetilde{A}} = 1$. Furthermore, for large enough $\dim \mathcal{H}_{R_1}$, we have the inequality

$$\hat{\rho}_{\widetilde{A}} \leq e^{-\lambda} \sigma_B \otimes \mathbb{1}_{R_1}, \tag{A.92}$$

and so

$$H_{\min}(\widetilde{A}|B)_{\hat{\rho}_{\widetilde{A}}} \geq H_{\min}^\varepsilon(A|B)_{\rho_A}. \tag{A.93}$$

To prove equality, note that $\hat{\rho}_{\widetilde{A}} \in B^\varepsilon(V_0 \rho_A V_0^\dagger)$ because the distance between $\hat{\rho}_{\widetilde{A}}$ and $V_0 \rho_A V_0^\dagger = \rho_A \otimes |0\rangle\langle 0|$ is the same as that between $\rho_A^\varepsilon$ and $\rho_A$. Thus, we have

$$H_{\min}^\varepsilon(\widetilde{A}|B)_{V_0 \rho_A V_0^\dagger} \geq H_{\min}(\widetilde{A}|B)_{\hat{\rho}_{\widetilde{A}}}. \tag{A.94}$$

Then use Lemma A.16 with $\mathcal{T}$ the identity on $\mathcal{M}_B$ to obtain

$$H_{\min}^\varepsilon(\widetilde{A}|B)_{V_0 \rho V_0^\dagger} = H_{\min}^\varepsilon(A|B)_\rho. \tag{A.95}$$

Combining these implies what we wanted:

$$H_{\min}(\widetilde{A}|B)_{\hat{\rho}_{\widetilde{A}}} = H_{\min}^\varepsilon(A|B)_\rho. \tag{A.96}$$

This is what we wanted to show, but so far only for the min-entropy. We would like to use duality (Theorem A.5) to derive the analogous thing for max-entropy. Consider any $\rho_A \in \mathcal{M}_A$, and let the purification be $|\psi\rangle_{AA'}$. By duality,

$$H_{\max}^\varepsilon(A|B)_\psi = -H_{\min}^\varepsilon(B'|A')_\psi. \tag{A.97}$$

As above, we can find an isometry $W_0 : \mathcal{H} \to \mathcal{H} \otimes \mathcal{H}_{R_2}$ acting on $B'$, defined as in Remark A.15, and a normalized state $\hat{\rho}_{\widetilde{B}'} \in \mathcal{B}^\varepsilon(W_0 \rho_{B'} W_0^\dagger)$ such that

$$H_{\min}^\varepsilon(B'|A')_{\rho_{B'}} = H_{\min}(\widetilde{B}'|A')_{\hat{\rho}_{\widetilde{B}'}}, \tag{A.98}$$

with the algebras defined as in case 1 of Remark A.17. Here $\rho_{B'}$ is the reduced density matrix of $|\psi\rangle_{AA'}$ on $\mathcal{M}_B'$.

We would like to apply duality again to convert the right hand side of eq. (A.98) back to a max-entropy, but the issue is that $\hat{\rho}_{\widetilde{B}'}$ may not have a purification on $\mathcal{M}_{\widetilde{B}'}' = \mathcal{M}_B$. We solve this with a third isometry $X_0 : \mathcal{H} \to \mathcal{H} \otimes \mathcal{H}_{R_3}$ on $\mathcal{M}_B$, again defined as in Remark A.15. For sufficiently large $\mathcal{H}_{R_3}$, we can find $|\widetilde{\psi}\rangle \in \mathcal{H} \otimes \mathcal{H}_{R_2} \otimes \mathcal{H}_{R_3}$ that purifies $\hat{\rho}_{\widetilde{B}'}$. We have

$$H_{\max}(\widetilde{A}|\widetilde{B})_{\hat{\rho}_{\widetilde{A}}} = -H_{\min}(\widetilde{B}'|A')_{\hat{\rho}_{\widetilde{B}'}}, \tag{A.99}$$

where now $\hat{\rho}_{\widetilde{A}}$ is the reduced state of $|\widetilde{\psi}\rangle$ on $\mathcal{M}_{\widetilde{A}} = \mathcal{M}_A \otimes \mathcal{L}(\mathcal{H}_{R_3})$ with $\mathcal{M}_{\widetilde{B}} = \mathcal{M}_B \otimes \mathcal{L}(\mathcal{H}_{R_3})$. If $|\widetilde{\psi}\rangle$ is chosen to maximize the inner product with $X_0 W_0 |\psi\rangle$, we have $\hat{\rho}_{\widetilde{A}} \in \mathcal{B}^\varepsilon(X_0 \rho_A X_0^\dagger)$, which completes the proof.

Note that we used different isometries $V_0$ (on $\mathcal{M}_A$) and $X_0$ (on $\mathcal{M}_B$) in our constructions of $\hat{\rho}_{\widetilde{A},\min}, \hat{\rho}_{\widetilde{A},\max}$. However, we could have easily adapted both our proofs to define $V = X_0 V_0$ with $\mathcal{M}_{\widetilde{A}} = \mathcal{M}_A \otimes \mathcal{L}(\mathcal{H}_{R_1}) \otimes \mathcal{L}(\mathcal{H}_{R_3})$ and $\mathcal{M}_{\widetilde{B}} = \mathcal{M}_B \otimes \mathcal{L}(\mathcal{H}_{R_3})$ in both, which also satisfies all the conditions required for Lemma A.16. $\qquad\square$

**Lemma A.33** (proposition 5.5 of [23]). *Let* $\varepsilon' \geq 0$ *such that* $\varepsilon + \varepsilon' < 1$. *Then it holds that*

$$H_{\min}^\varepsilon(A|B)_\Psi \leq H_{\max}^{\varepsilon'}(A|B)_\Psi + \log \frac{1}{1 - (\varepsilon + \varepsilon')^2}. \tag{A.100}$$

*Proof.* According to lemma A.32, we can always extend the Hilbert space and algebras isometrically $\mathcal{M}_A \to \mathcal{M}_{\widetilde{A}}$ such that there exists a normalized state $\Psi_{\widetilde{A},\min}$ with $H_{\min}(\widetilde{A}|\widetilde{B})_{\Psi_{\widetilde{A},\min}} = H_{\min}^\varepsilon(A|B)_\Psi$. Similarly, there exists a normalized state $\Psi_{\widetilde{A},\max}$ with $H_{\max}^{\varepsilon'}(A|B)_\Psi = H_{\max}(\widetilde{A}|\widetilde{B})_{\Psi_{\widetilde{A},\max}}$. Both of these states can be found within $\varepsilon, \varepsilon'$ distance of the image of $\Psi_A$, respectively.

Hence there exists a normalized $\Psi_{\widetilde{B}}$ such that $\Psi_{\widetilde{A},\min} \leq e^{-\lambda} \Psi_{\widetilde{B}}$, with $\lambda = H_{\min}^\varepsilon(A|B)_\Psi$. Therefore

$$\begin{aligned}
H_{\max}^{\varepsilon'}(A|B)_\Psi &= H_{\max}(\widetilde{A}|\widetilde{B})_{\Psi_{\widetilde{A},\max}} \\
&\geq \log \| \sqrt{\Psi_{\widetilde{A},\max}} \sqrt{\Psi_{\widetilde{B}}} \|_1^2 \\
&\geq \lambda + \log \| \sqrt{\Psi_{\widetilde{A},\max}} \sqrt{\Psi_{\widetilde{A},\min}} \|_1^2 \\
&= \lambda + \log(1 - P^2(\Psi_{\widetilde{A},\min}, \Psi_{\widetilde{A},\max})) \\
&\geq H_{\min}^\varepsilon(A|B)_\Psi - \log \frac{1}{1 - (\varepsilon + \varepsilon')^2},
\end{aligned} \tag{A.101}$$

where the first inequality follows from the definition of the smooth max-entropy and that we picked a particular $\Psi_{\widetilde{B}}$ instead of supremizing, the fourth line from the definition of the purified distance $P$, and the final inequality from the triangle inequality for the purified distance, $P(\Psi_{\widetilde{A},\min}, \Psi_{\widetilde{A},\max}) \leq \varepsilon + \varepsilon'$. $\qquad\square$

**Theorem A.34** (QAEP, converse; corollary 6.7 of [23]). *Let* $0 < \varepsilon < 1$. *Then*

$$\lim_{n\to\infty} \frac{1}{n} H_{\min}^\varepsilon(A|B)_\Psi \leq S(A|B)_\Psi \leq \lim_{n\to\infty} \frac{1}{n} H_{\max}^\varepsilon(A|B)_\Psi. \tag{A.102}$$

*Proof.* From lemma A.33 and Theorem A.30, it follows that

$$\begin{aligned}
\frac{1}{n} H_{\min}^\varepsilon(A|B)_\Psi &\leq \frac{1}{n} H_{\max}^{\varepsilon'}(A|B)_\Psi + \frac{1}{n} \log \frac{1}{1 - (\varepsilon + \varepsilon')^2} \\
&\leq S(A|B)_\Psi + \frac{1}{n} \log \frac{1}{1 - (\varepsilon + \varepsilon')^2} + \frac{\delta(\varepsilon', \nu)}{\sqrt{n}},
\end{aligned} \tag{A.103}$$

where $\nu = \Upsilon(\Psi_A \| \Psi_B)$. Using duality to obtain the analogous inequality for the max-entropy then taking the limit $n \to \infty$ completes the proof. $\qquad\square$

## A.4 Bounds between the smooth min-, max-, and von Neumann entropies

In this subsection, we relate the smooth min- and max-entropies to the von Neumann entropy. It suffices to relate the smooth max-entropy to a smooth von Neumann entropy. The bound on the min-entropy follows by duality.

First, a technical lemma on the monotonicity in $\alpha$ of the sandwiched $\alpha$-Renyi divergences.

**Lemma A.35.** *The sandwiched quantum Renyi diverges obey the inequality*

$$S_\alpha(\rho_A||\sigma_A) \geq S_\beta(\rho_A||\sigma_A), \tag{A.104}$$

*for $\alpha \geq \beta$ and $\sigma_A \geq 0$ such that $\operatorname{supp}\rho_A \subseteq \operatorname{supp}\sigma_A$.*

*Proof.* Using the expression for the sandwiched Renyi divergences in (A.12), we see that it is enough to prove monotonicity in $\alpha$ for a fixed choice of $\tau_A$. In particular, we need to prove that

$$\frac{\alpha}{\alpha-1}\operatorname{tr}_A\left(\rho_A^{1/2}\sigma_B^{\frac{1-\alpha}{\alpha}}\rho_A^{1/2}\tau_A^{\frac{\alpha-1}{\alpha}}\right) \geq \frac{\beta}{\beta-1}\operatorname{tr}_A\left(\rho_A^{1/2}\sigma_B^{\frac{1-\beta}{\beta}}\rho_A^{1/2}\tau_A^{\frac{\beta-1}{\beta}}\right), \tag{A.105}$$

for $\alpha \geq \beta$. This is proved simply in Lemma 19 of [68] using Jensen's inequality. This fact is related to the monotonicity of the $\alpha$-norms defined in eq. (A.11). $\qquad\square$

From definition (A.2), it follows

$$H_\alpha(A|B) \geq H_\beta(A|B), \quad \text{for } \alpha \leq \beta. \tag{A.106}$$

**Theorem A.36.** *Given nested algebras $\mathcal{M}_A \supset \mathcal{M}_B$ and $\varepsilon, \varepsilon' \geq 0$ such that $\varepsilon + \varepsilon' < 1$, we have*

$$H_{\max}^\varepsilon(A|B) \geq S^{\varepsilon'}(A|B) - \log\frac{1}{(1-2(\varepsilon+\varepsilon'))^2}, \tag{A.107}$$

*with $S^\varepsilon(A|B) := \lim_{\alpha\to 1} H_\alpha^\varepsilon(A|B)$ the smooth conditional von Neumann entropy.*

*Proof.* By lemma A.32, we can embed $\mathcal{M}_A$ into $\mathcal{M}_{\widetilde{A}}$ with a trace-preserving isometry $V$ such that there exists a normalized state $\hat{\rho}_{\widetilde{A}} \in \mathcal{B}^\varepsilon(V\rho_A V^\dagger)$ with $H_{\max}^\varepsilon(A|B)_\rho = H_{\max}(\widetilde{A}|\widetilde{B})_{\hat{\rho}}$. Similarly, let $\rho_A^* \in \mathcal{B}^{\varepsilon'}(\rho_A)$ be such that $H^{\varepsilon'}(A|B)_\rho = S(A|B)_{\rho^*}$. Denote by $\rho_B^*$ the density matrix of this state reduced to $B$. Then, by the definition of $H_{\max}$, we have the chain of inequalities

$$\begin{aligned}
H_{\max}^\varepsilon(A|B)_\rho = H_{\max}(\widetilde{A}|\widetilde{B})_{\hat{\rho}} &\geq \log F^2(\hat{\rho}_{\widetilde{A}}, V\rho_B^* V^\dagger) = \log\left(1 - P^2(\hat{\rho}_{\widetilde{A}}, V\rho_B^* V^\dagger)\right) \\
&\geq \log\left(1 - P^2(V\rho_A^* V^\dagger, V\rho_B^* V^\dagger) - 2(\varepsilon+\varepsilon') - (\varepsilon+\varepsilon')^2\right) \\
&\geq \log F^2(\rho_A^*, \rho_B^*) - \log\frac{1}{(1-2(\varepsilon+\varepsilon'))^2} \\
&= -S_{1/2}(\rho_A^*||\rho_B^*) - \log\frac{1}{(1-2(\varepsilon+\varepsilon'))^2} \\
&\geq S^{\varepsilon'}(A|B) - \log\frac{1}{(1-2(\varepsilon+\varepsilon'))^2}. \tag{A.108}
\end{aligned}$$

In the first inequality, we used the definition of $H_{\max}$. In the second inequality, we used the triangle inequality of the purified distance and the fact that the isometry $V$ preserves the purified distance. In the third inequality, we used that the purified distance obeys $0 < P < 1$, as well as the fact that $1 - P^2 \geq F^2$. Then in the final inequality we used lemma A.35, together with the definition of $\rho_A^*$ and $\rho_B^*$. $\qquad\square$

Note that from Theorem A.19, we can bound the smoothed von Neumann entropy by

$$H_{\min}^{\varepsilon'}(A|B) - \log\frac{1}{(1-2(\varepsilon+\varepsilon'))^2} \leq S^\varepsilon(A|B) \leq H_{\max}^{\varepsilon''}(A|B) + \log\frac{1}{(1-2(\varepsilon+\varepsilon''))^2}. \tag{A.109}$$

Taking the various smoothing parameters independently to zero gives us bounds between smoothed and non-smoothed conditional entropies.

## A.5 The chain rule

In this subsection, we will prove the chain rule that we need in the main text. Given a chain of inclusions of algebras $\mathcal{M}_A \supset \mathcal{M}_B \supset \mathcal{M}_C$ with a state $\rho_A \in \mathcal{M}_A$, the chain rule states that for $\varepsilon, \varepsilon', \varepsilon'' > 0$, then

$$H_{\min}^{\varepsilon+2\varepsilon'+\varepsilon''}(A|C) \geq H_{\min}^{\varepsilon'}(A|B) + H_{\min}^{\varepsilon''}(B|C) - \log\frac{2}{\varepsilon^2}. \tag{A.110}$$

**Remark A.37.** Let $|\Psi\rangle \in \mathcal{H}$ be a pure state, and let $\mathcal{L}(\mathcal{H}) \supset \mathcal{M}_A \supset \mathcal{M}_B$ be algebras. Then for any projector $\Pi_B \in \mathcal{M}_B$, there exists a projector $\Pi_{B'} \in \mathcal{M}_B'$ such that $\Pi_B \Psi_B^{-1/2} |\Psi\rangle = \Psi_B^{-1/2} \Pi_{B'} |\Psi\rangle$. The proof follows from using the Schmidt decomposition on $|\Psi\rangle$.

**Lemma A.38** (Lemma 21 of [72]). *Given a (possibly subnormalized) pure state $|\Psi\rangle \in \mathcal{H}$ along with nested algebras $\mathcal{L}(\mathcal{H}) \supset \mathcal{M}_A \supset \mathcal{M}_B$, then there exists a projection $\Pi_{B'} \in \mathcal{M}_B'$ such that $P(|\psi\rangle, \Pi_{B'}|\psi\rangle) \leq \varepsilon$ and*

$$-S_\infty(\Psi_A' || \Psi_B) \geq H_{\min}(A|B)_\Psi - \log\frac{2}{\varepsilon^2}, \tag{A.111}$$

*where $|\Psi'\rangle = \Pi_{B'}|\Psi\rangle$.*

*Proof.* Consider an arbitrary projector $\Pi_{B'} \in \mathcal{M}_B'$. By remark A.37, there exists a dual projector $\Pi_B \in \mathcal{M}_B$ such that $\Pi_B \Psi_B^{-1/2}|\Psi\rangle = \Psi_B^{-1/2}\Pi_{B'}|\Psi\rangle$. Therefore by (A.2) and lemma A.4, we have the expression

$$S_\infty(\Psi_A' || \Psi_B) = \log \sup_{\tau_A : \mathrm{tr}_A \tau_A = 1} \mathrm{tr}_A\left[\tau_A \Pi_B \Psi_B^{-1/2} \Psi_A \Psi_B^{-1/2} \Pi_B\right]$$
$$\leq -H_{\min}(A|B)_\Psi + \log \sup_{\tau_A : \mathrm{tr}_A \tau_A = 1} \mathrm{tr}_A\left[\tau_A \Pi_B \Psi_B^{-1/2} \sigma_B \Psi_B^{-1/2} \Pi_B\right], \tag{A.112}$$

where $\sigma_B$ is the state which optimizes the Renyi-divergence in the definition of $H_{\min}(A|B)_\Psi$, and we have plugged in $\Psi_A \leq e^{-H_{\min}(A|B)_\Psi}\sigma_B$. Since the optimization over $\tau_A$ computes the maximum eigenvalue of the operator $O_B := \Pi_B \Psi_B^{-1/2}\sigma_B \Psi_B^{-1/2}\Pi_B$ and since the spectral projectors of $O_B$ are in both $\mathcal{M}_A$ and $\mathcal{M}_B$, we have the relation

$$S_\infty(\Psi_A' || \Psi_B) \leq -H_{\min}(A|B)_\Psi + \log \sup_{\tau_B : \mathrm{tr}_B \tau_B = 1} \mathrm{tr}_B\left[\tau_B \Pi_B \Psi_B^{-1/2} \sigma_B \Psi_B^{-1/2} \Pi_B\right]. \tag{A.113}$$

This has been for a general projector. Now consider a $\Pi_B^*$ which projects onto the smallest eigenvalues of $\Gamma_B := \Psi_B^{-1/2}\sigma_B \Psi_B^{-1/2}$ such that $\langle\Psi|\Pi_B^*|\Psi\rangle \geq \langle\Psi|\Psi\rangle - \varepsilon^2/2$. Note that by the definition of the purified fidelity distance this inequality guarantees that

$$P(|\Psi\rangle, \Pi_{B'}^*|\Psi\rangle) \leq \varepsilon, \tag{A.114}$$

with $\Pi_{B'}^*$ the conjugate of $\Pi_B^*$ on $B$ under the Schmidt decomposition of $|\Psi\rangle_{BB'}$.

Let $\Pi_B^+$ be a projector onto the maximal eigenvalue of $O_B^* = \Pi_B^* \Gamma_B \Pi_B^*$. Using that all the projectors commute with $\Gamma_B$, we can then write

$$\sup_{\tau_B : \mathrm{tr}_B \tau_B = 1} \mathrm{tr}_B\left[\tau_B \Pi_B^* \Gamma_B \Pi_B^*\right] = \inf_{\tau_B : \mathrm{tr}_B \tau_B = 1} \mathrm{tr}_B\left[\tau_B(\mathbb{1} - \Pi_B^* + \Pi_B^+)\Gamma_B\right]. \tag{A.115}$$

This follows because the left-hand side equals the largest eigenvalue of $\Pi_B^* \Gamma_B \Pi_B^*$, while the right-hand side equals the same thing: its smallest eigenvalue in the union of the orthogonal subspace and the maximal eigenvector of $\Pi_B^* \Gamma_B \Pi_B^*$. Then, picking the case

$$\tau_B = \frac{(\mathbb{1} - \Pi_B^* + \Pi_B^+)\Psi_B(\mathbb{1} - \Pi_B^* + \Pi_B^+)}{\mathrm{tr}_B\left[(\mathbb{1} - \Pi_B^* + \Pi_B^+)\Psi_B(\mathbb{1} - \Pi_B^* + \Pi_B^+)\right]}, \tag{A.116}$$

we have that

$$\sup_{\tau_B : \mathrm{tr}_B \tau_B = 1} \mathrm{tr}_B \left( \tau_B \Pi_B^* \Gamma_B \Pi_B^* \right) \leq \frac{\mathrm{tr}_B \left( \Gamma_B^{1/2} \Psi_B \Gamma_B^{1/2} (\mathbb{1} - \Pi_B^* + \Pi_B^+) \right)}{\langle \Psi | (\mathbb{1} - \Pi_B^* + \Pi_B^+) | \Psi \rangle} \leq \frac{2}{\varepsilon^2}, \tag{A.117}$$

where in the last line we used the bound on the overlap, $\langle \Psi | (\Pi_B^* - \Pi_B^+) | \Psi \rangle \leq \langle \Psi | \Psi \rangle - \varepsilon^2/2$, together with the fact that $\mathrm{tr}_B \left[ \Gamma_B^{1/2} \Psi_B \Gamma_B^{1/2} (1 - \Pi_B^* + \Pi_B^+) \right] \leq \mathrm{tr}_B \left[ \Gamma_B^{1/2} \Psi_B \Gamma_B^{1/2} \right] = \mathrm{tr}_B \sigma_B \leq 1$. This proves the bound. $\qquad \square$

Using the above lemma, we now prove the chain rule inequality.

**Theorem A.39** (Chain rule; lemma A.8 of [29]). *Given a (possibly subnormalized) density matrix $\rho_A \in \mathcal{M}_A \subset \mathcal{B}(\mathcal{H})$ and inclusions $\mathcal{M}_A \supset \mathcal{M}_B \supset \mathcal{M}_C$, then the conditional min entropies obey*

$$H_{\min}^{\varepsilon + 2\varepsilon' + \varepsilon''}(A|C)_\rho \geq H_{\min}^{\varepsilon'}(A|B)_\rho + H_{\min}^{\varepsilon''}(B|C)_\rho - \log \frac{2}{\varepsilon^2}. \tag{A.118}$$

*Proof.* In this proof, we will use asterisks to denote states that optimize the relevant quantity. For example, let $\rho_A^* \in \mathcal{B}^{\varepsilon'}(\rho_A)$ such that

$$H_{\min}(A|B)_{\rho^*} = H_{\min}^{\varepsilon'}(A|B)_\rho. \tag{A.119}$$

Furthermore, let $\tilde{\rho}_B^*$ and $\sigma_C$ be states such that $\tilde{\rho}_B^* \in \mathcal{B}^{\varepsilon''}(\rho_B)$ with

$$\tilde{\rho}_B^* \leq e^{-H_{\min}^{\varepsilon''}(B|C)_\rho} \sigma_C, \tag{A.120}$$

and $H_{\min}^{\varepsilon''}(B|C)_\rho = H_{\min}(B|C)_{\tilde{\rho}_B^*}$.

Given a purification $|\Psi^*\rangle$ of $\rho_A^*$ on $AA'$, by Lemma A.38, we can find a projector $\Pi_{A'}$ such that $\langle \Psi^* | \Pi_{A'} | \Psi^* \rangle \geq 1 - \varepsilon^2/2$ so that $\left( \rho_P^* \right)_{AA'} := \Pi_{A'} |\Psi^*\rangle \langle \Psi^* | \Pi_{A'} \in \mathcal{B}^{\varepsilon}(\rho_{AA'}^*)$ as well as

$$\left( \rho_P^* \right)_A \leq \rho_B^* e^{-H_{\min}(A|B)_{\rho^*} + \log\left( \frac{2}{\varepsilon^2} \right)} = \rho_B^* e^{-H_{\min}^{\varepsilon'}(A|B)_\rho + \log\left( \frac{2}{\varepsilon^2} \right)}. \tag{A.121}$$

Note that by construction the purified distance between $|\Psi^*\rangle$ and $\Pi_{A'} |\Psi^*\rangle$ is

$$P(|\Psi^*\rangle, \Pi_{A'} |\Psi^*\rangle)_A \leq \varepsilon. \tag{A.122}$$

Now, by Lemma B.3 in [29], there is an operator $T_B$ such that $T_B |\Psi^*\rangle_{AA'} = |\tilde{\Psi}^*\rangle_{AA'}$ where $|\tilde{\Psi}^*\rangle_{AA'}$ is a purification of $\tilde{\rho}_B^*$ onto $AA'$ and

$$P(|\Psi^*\rangle, |\tilde{\Psi}^*\rangle)_{AA'} = P(|\Psi^*\rangle, |\tilde{\Psi}^*\rangle)_B. \tag{A.123}$$

Applying $T_B$ to the states on either side of (A.121), we get

$$T_B \left( \rho_P^* \right)_A T_B^\dagger \leq \tilde{\rho}_B^* e^{-H_{\min}^{\varepsilon'}(A|B)_\rho + \log\left( \frac{2}{\varepsilon^2} \right)} \leq \sigma_C e^{-H_{\min}^{\varepsilon'}(A|B)_\rho - H_{\min}^{\varepsilon''}(B|C) + \log\left( \frac{2}{\varepsilon^2} \right)}. \tag{A.124}$$

To finish the proof, we thus just need to show that $T_B \left( \rho_P^* \right)_A T_B^\dagger \in \mathcal{B}^{\varepsilon + 2\varepsilon' + \varepsilon''}(\rho_A)$, after which the result follows by definition of the smoothed conditional min-entropy. Let $|\Psi\rangle$ be a purification of $\rho_A$. Then

$$P(T_B \Pi_{A'} |\Psi^*\rangle, |\Psi\rangle)_A \leq P(T_B \Pi_{A'} |\Psi^*\rangle, \Pi_{A'} |\Psi^*\rangle)_A + P(\Pi_{A'} |\Psi^*\rangle, |\Psi^*\rangle)_A + P(|\Psi^*\rangle, |\Psi\rangle)_A, \tag{A.125}$$

using the triangle inequality. Moreover,

$$P(T_B \Pi_{A'} |\Psi^*\rangle, \Pi_{A'} |\Psi^*\rangle)_A \leq P(T_B \Pi_{A'} |\Psi^*\rangle, \Pi_{A'} |\Psi^*\rangle)_{AA'} \leq P(T_B |\Psi^*\rangle, |\Psi^*\rangle)_{AA'}$$
$$= P(|\Psi^*\rangle, |\tilde{\Psi}^*\rangle)_B, \tag{A.126}$$

by monotonicity and the fact that projections decrease the purified distance. Putting this all together we get

$$
\begin{aligned}
P(T_B \Pi_{A'} |\Psi^*\rangle, |\Psi\rangle)_A &\leq P(|\Psi^*\rangle, |\tilde{\Psi}^*\rangle)_B + P(\Pi_{A'} |\Psi^*\rangle, |\Psi^*\rangle)_A + P(|\Psi^*\rangle, |\Psi\rangle)_A \\
&\leq P(|\Psi^*\rangle, |\Psi\rangle)_B + P(|\Psi\rangle, |\tilde{\Psi}^*\rangle)_B + P(\Pi_{A'} |\Psi^*\rangle, |\Psi^*\rangle)_A + P(|\Psi^*\rangle, |\Psi\rangle)_A \\
&\leq \varepsilon + 2\varepsilon' + \varepsilon'',
\end{aligned} \tag{A.127}
$$

where again we used the triangle inequality and the fact that the relevant states are in their respective $\varepsilon$-balls. This is what we needed to show. □

## A.6 Strong sub-additivity

In this subsection, we prove Theorem 2.40 which is the statement of strong sub-additivity of the conditional min-, max- and vN entropies. First we recall lemmas about completely positive and trace preserving maps.

**Lemma A.40** (Theorem 2 of [73]). *Let $\rho, \sigma$ be positive operators in some algebra $\mathcal{M}$. Let $\varepsilon : \mathcal{M} \to \mathcal{N}$ be a completely-positive and trace-preserving map (CPTP). Then the sandwiched quantum Renyi divergences from definition A.1 are monotonically decreasing under action by the map*

$$
S_\alpha(\rho \| \sigma) \geq S_\alpha(\varepsilon(\rho) \| \varepsilon(\sigma)). \tag{A.128}
$$

**Corollary A.41.** *The purified distance between two density matrices $\rho, \sigma \in \mathcal{M}$ is monotonically decreasing under action by a CPTP map $\varepsilon : \mathcal{M} \to \mathcal{N}$,*

$$
P(\varepsilon(\rho), \varepsilon(\sigma)) \leq P(\rho, \sigma). \tag{A.129}
$$

*Proof.* The fidelity between $\rho, \sigma$ is related to the sandwiched Renyi entropy for $\alpha = 1/2$ as $F(\rho, \sigma) = -S_{1/2}(\rho \| \sigma)$. Given the definition of the purified distance in (36) in terms of the fidelity, we see that Lemma A.40 implies the claim. □

**Theorem A.42** (Strong subadditivity). *Let $\mathcal{M}_{A_0}$, $\mathcal{M}_{A_1}$, $\mathcal{M}_{B_0}$ and $\mathcal{M}_{B_1}$ be von Neumann algebras with corresponding traces acting on $\mathcal{H}$ with the following inclusion structure: $\mathcal{M}_{A_0} \supset \mathcal{M}_{B_0} \supset \mathcal{M}_{B_1}$ and $\mathcal{M}_{A_0} \supset \mathcal{M}_{A_1} \supset \mathcal{M}_{B_1}$. Let $\mathrm{tr}_{A_0 \to A_1} : \mathcal{M}_{A_0} \to \mathcal{M}_{A_1}$ and $\mathrm{tr}_{B_0 \to B_1} : \mathcal{M}_{B_0} \to \mathcal{M}_{B_1}$ be partial traces such that the restriction $\mathrm{tr}_{A_0 \to A_1}|_{B_0}$ is a map $\mathrm{tr}_{A_0 \to A_1}|_{B_0} : \mathcal{M}_{B_0} \to \mathcal{M}_{B_1}$ and $\mathrm{tr}_{A_0 \to A_1}|_{B_0} \leq \mathrm{tr}_{B_0 \to B_1}$. Then*

$$
H_{\min}^\varepsilon(A_0|B_0) \leq H_{\min}^\varepsilon(A_1|B_1), \tag{A.130}
$$

$$
S(A_0|B_0) \leq S(A_1|C_1), \tag{A.131}
$$

$$
H_{\max}^\varepsilon(A_0|B_0) \leq H_{\max}^\varepsilon(A_1|B_1). \tag{A.132}
$$

*Proof.* We begin by proving the statements for $\varepsilon = 0$. Using Definition 2.28, we have that the equation

$$
H_{\min}(A_0|B_0) = - \min_{\sigma_{B_0} : \mathrm{tr}_{B_0}[\sigma_{B_0}] \leq 1} \inf \left\{ \lambda : \rho_{A_0} \leq e^\lambda \sigma_{B_0} \right\}. \tag{A.133}
$$

Let $\sigma_{B_0}$ and $\lambda_0$ be such that

$$
\rho_{A_0} \leq e^{\lambda_0} \sigma_{B_0}, \tag{A.134}
$$

where $\lambda_0 = -H_{\min}(A_0|B_0)$. Then by the fact that $\mathrm{tr}_{A_0 \to A_1}$ is a partial trace, it holds that

$$
\rho_{A_1} = \mathrm{tr}_{A_0 \to A_1}[\rho_{A_0}] \leq e^{\lambda_0} \mathrm{tr}_{A_0 \to A_1}[\sigma_{B_0}] \leq e^{\lambda_0} \mathrm{tr}_{B_0 \to B_1}[\sigma_{B_0}] = e^{\lambda_0} \sigma_{B_1}. \tag{A.135}
$$

Therefore $H_{\min}(A_1|B_1) \geq -\lambda_0$, proving A.130. To prove inequality (A.132), we can just use the inequality for $H_{\min}$ together with statement of duality, Theorem 2.33.

Finally, to prove strong sub-additivity for the von Neumann conditional entropy, we just use that we can write the conditional entropy in terms of the relative entropy as

$$S(A_0|B_0) = -S_{\text{rel}}(\rho_{A_0}||\rho_{B_0}) := -\text{tr}_{A_0}[\rho_{A_0} \log \rho_{A_0}] + \text{tr}_{A_0}[\rho_{A_0} \log \rho_{B_0}], \tag{A.136}$$

where $\rho_{B_0} \in \mathcal{M}_{B_0}$ is a viewed as an operator on $\mathcal{M}_{A_0}$ via the standard inclusion of $\mathcal{M}_{B_0} \subset \mathcal{M}_{A_0}$. Since $\text{tr}_{A_0 \to A_1}$ is a completely-positive map by assumption of being a partial trace, it follows that Lemma A.40 applied in the limit of $\alpha \to 1$ gives

$$S_{\text{rel}}(\rho_{A_0}||\rho_{B_0}) \geq S_{\text{rel}}(\text{tr}_{A_0 \to A_1}[\rho_{A_0}]||\text{tr}_{A_0 \to A_1}[\rho_{B_0}]) = S_{\text{rel}}(\rho_{A_1}||\text{tr}_{A_0 \to A_1}[\rho_{B_0}]). \tag{A.137}$$

Now by assumption

$$\text{tr}_{A_0 \to A_1}(\rho_{B_0}) \leq \text{tr}_{B_0 \to B_1}(\rho_{B_0}) = \rho_{B_1}, \tag{A.138}$$

and so because the log function is an operator monotone,

$$\log \text{tr}_{A_0 \to A_1}(\rho_{B_0}) \leq \log \rho_{B_1}. \tag{A.139}$$

Plugging this in above, we get the desired inequality

$$S(A_0|B_0) = -S_{\text{rel}}(\rho_{A_0}||\rho_{B_0}) \leq -S_{\text{rel}}(\rho_{A_1}||\rho_{B_1}) = S(A_1|B_1). \tag{A.140}$$

We now prove the statements for $\varepsilon > 0$. Let $\rho_{A_0}^* \in \mathcal{B}^\varepsilon(\rho_{A_0})$ be a density matrix which optimizes the min-entropy. Because the purified distance is monotonically decreasing under completely positive maps [23], it holds that $\text{tr}_{A_0 \to A_1}(\rho_{A_0}^*) \in \mathcal{B}^\varepsilon(\rho_{A_1})$ and so we have the inequality

$$H_{\min}^\varepsilon(A_0|B_0)_\rho = H_{\min}(A_0|B_0)_{\rho^*} \leq H_{\min}^\varepsilon(A_1|B_1)_\rho. \tag{A.141}$$

The use of duality for the smoothed conditional entropies then proves the corresponding inequality for the max-entropy. $\qquad \square$

# B  State-specific reconstruction for algebras

In this appendix we give a formal definition of state-specific reconstruction for finite-dimensional von Neumann algebras and prove some basic results about it. The definitions given here are based heavily on those of [14] which gave an in-depth discussion of state-specific reconstruction for tensor product Hilbert spaces. We will focus here on the formal details of the generalization to algebras with centers, and refer readers to [14] for detailed motivation and discussion.

**Definition B.1** (Haar unitaries on algebras). Let $\mathcal{M}_A$ be a finite-dimensional von Neumann algebra. We say that a unitary $U_A \in \mathcal{M}_A$ is Haar random if

$$U_A = \oplus_\alpha U_{A_\alpha}, \tag{B.1}$$

with $U_{A_\alpha}$ independently sampled Haar random unitaries on $\mathcal{H}_{A_\alpha}$ (with $\mathcal{H}_{A_\alpha}$ defined as in Theorem 2.12).

**Definition B.2.** Let $\mathcal{M}_A$ be a finite-dimensional von Neumann algebra acting on a Hilbert space $\mathcal{H}$ and let $\mathcal{H}_{U_A}$ be the space of square-integrable functions on the group of unitaries $U_A \in \mathcal{M}_A$. We define the isometry $W_A : \mathcal{H} \to \mathcal{H} \otimes \mathcal{H}_{U_A}$ by

$$W_A := \int dU_A |U_A\rangle_{U_A} \otimes U_A, \tag{B.2}$$

and $dU_A$ is the Haar measure normalized to $\int dU_A = 1$.

**Remark B.3.** The isometry $W_A$ commutes with $\mathcal{M}'_A$ and hence maps $\mathcal{M}_A$ into $\mathcal{M}_A \otimes \mathcal{L}(\mathcal{H}_{U_A})$ in the sense of Definition A.11.

**Lemma B.4.** *We have $\mathcal{H}_{U_A} \cong \otimes_\alpha \mathcal{H}_{U_{A_\alpha}}$ where $\mathcal{H}_{U_{A_\alpha}}$ is the Hilbert space of square-integrable functions on the unitary group on $\mathcal{H}_{A_\alpha}$. $\mathcal{H}_{U_{A_\alpha}}$ can be decomposed using Peter-Weyl duality as*

$$\mathcal{H}_{U_{A_\alpha}} \cong \bigoplus_\mu \left( \mathcal{H}_\mu \otimes \mathcal{H}_\mu^* \right), \tag{B.3}$$

*where $\{\mu\}$ is the set of irreducible representation of the unitary group on $\mathcal{H}_{A_\alpha}$ and $\mathcal{H}_\mu$ is the Hilbert space on which the representation $\mu$ acts. Given $|\psi_\alpha\rangle \in \mathcal{H}_{A_\alpha} \otimes \mathcal{H}_{A'_\alpha}$, we have*

$$W_A |\psi_\alpha\rangle = \left( \otimes_{\widetilde{\alpha} \neq \alpha} |0\rangle_{U_{A_{\widetilde{\alpha}}}} \right) O_{\text{SWAP}} |\psi_\alpha\rangle |\text{MAX}\rangle_{U_{A_\alpha}}, \tag{B.4}$$

*where $|0\rangle_{U_{A_{\widetilde{\alpha}}}}$ is the trivial representation state in $\mathcal{H}_{U_{A_{\widetilde{\alpha}}}}$, $|\text{MAX}\rangle_{U_{A_\alpha}}$ is the canonical maximally entangled state in $\mathcal{H}_{\mu_0} \otimes \mathcal{H}_{\mu_0}^* \subset \mathcal{H}_{U_{A_\alpha}}$ with $\mu_0$ the fundamental representation, and $O_{\text{SWAP}}$ swaps $\mathcal{H}_{A_\alpha}$ with $\mathcal{H}_{\mu_0}$.*

*Proof.* See the proof of Lemma 4.4 in [14]. The only novel ingredient here is the additional tensor product factor of $\otimes_{\widetilde{\alpha} \neq \alpha} |0\rangle_{U_{A_{\widetilde{\alpha}}}}$ which follows from the fact that $|\psi_\alpha\rangle$ is invariant under unitaries $U_{A_{\widetilde{\alpha}}}$ acting on $\mathcal{H}_{A_{\widetilde{\alpha}}}$ with $\widetilde{\alpha} \neq \alpha$. □

Heuristically, we can think of $W_A$ as extracting all information from $\mathcal{M}_A$ into $\mathcal{H}_{U_A}$.

**Definition B.5** (State-specific reconstruction). Let $V : \mathcal{H}_{\text{code}} \to \mathcal{H}_{\text{phys}}$ be an isometry and let $\mathcal{M}_b \subseteq \mathcal{L}(\mathcal{H}_{\text{code}})$ and $\mathcal{M}_B \subseteq \mathcal{L}(\mathcal{H}_{\text{phys}})$ be finite-dimensional von Neumann algebras with commutants $\mathcal{M}_{b'} := \mathcal{M}'_b$ and $\mathcal{M}_{B'} := \mathcal{M}'_B$. We say that $\mathcal{M}_B$ state-specifically reconstructs $\mathcal{M}_b$ for the state $|\psi\rangle$ with error $\varepsilon$ if there exists an isometry $W_B : \mathcal{H}_{\text{phys}} \to \mathcal{H}_{\text{phys}} \otimes \mathcal{H}_{U_b}$ mapping $\mathcal{M}_B$ to $\mathcal{M}_B \otimes \mathcal{L}(\mathcal{H}_{U_b})$ such that for all isometries $T_{b'} : \mathcal{H}_{\text{code}} \to \mathcal{H}_{\text{code}} \otimes \mathcal{H}_R$ mapping $\mathcal{M}_{b'}$ to $\mathcal{M}_{b'} \otimes \mathcal{L}(\mathcal{H}_R)$,

$$\|W_B V T_{b'} |\psi\rangle - V W_b T_{b'} |\psi\rangle\| \leq \varepsilon, \tag{B.5}$$

with the isometry $W_b$ defined as in Definition B.2.

**Remark B.6.** We demand $W_B$ works for all $T_{b'}$ so that the reconstruction depends only on the state within $b$, and not on the state in $b'$. Indeed, if the reconstruction of $b$ is allowed to depend on the bulk state outside $b$, there exist known examples where a region $b$ that is larger than the max-EW can be completely reconstructed. See Section 7.3 of [1].

**Remark B.7.** Definition B.5 (and Theorems B.8 and B.12 below) also extends to linear maps $V$ – such as those studied in [74] – that are not isometric but that nonetheless approximately preserve the normalization of all relevant states.[23]

Definition B.5 may seem unfamiliar to readers used to definitions of bulk reconstruction involving reconstructing *any* bulk operator with a boundary operator (as in e.g. (110)). The following two theorems connect these ideas, showing that being able to reconstruct the single isometry $W_b$ is (morally) equivalent to being able to reconstruct a large class of unitary operators $U_b$ with state-specific boundary unitaries $U_B$.

It is worth emphasizing that, as a general rule, not all unitaries $U_b$ will be reconstructible even when state-specific reconstruction is possible. Intuitively this is because some $U_b$ make the max-EW smaller and thus exclude themselves from the reconstructible region. This is true even though all unitaries, $U_b$, are integrated over in the definition of $W_b$, and $W_b$ is, by

---

[23]For nonisometric codes, it is natural to restrict the isometry $T_{b'}$ to have subexponential complexity. Such a restriction does not materially affect either of the proofs below.

definition, reconstructible! The consistency of these two statements depends crucially on the fact that the reconstruction of $W_b$ is only approximate; see [14] for detailed discussion of this point.

**Theorem B.8** (State-specific reconstruction of operators). *Let $\mathcal{M}_b \subseteq \mathcal{L}(\mathcal{H}_{\text{code}})$ be state-specifically reconstructible from $\mathcal{M}_B \subseteq \mathcal{L}(\mathcal{H}_{\text{phys}})$ with error $\varepsilon$ for both the state $|\psi\rangle$ and the state $U_b|\psi\rangle$ with $U_b \in \mathcal{M}_b$ unitary. Then there exists $U_B \in \mathcal{M}_B$ such that for all isometries $T_{b'}$,*

$$\|U_B V T_{b'}|\psi\rangle - V U_b T_{b'}|\psi\rangle\| \le 2\varepsilon + 2\varepsilon^{1/2}. \tag{B.6}$$

*Proof.* Let $\mathcal{H}_{\text{code}} \cong \oplus_\alpha(\mathcal{H}_{b,\alpha} \otimes \mathcal{H}_{b',\alpha})$ with $\mathcal{M}_b = \oplus_\alpha \mathcal{L}(\mathcal{H}_{b,\alpha})$. We have $U_b = \oplus_\alpha U_{b,\alpha}$. We define the unitary $F_{b,\alpha} \in \mathcal{L}(\mathcal{H}_{U_{b_\alpha}})$ to act as $U_{b,\alpha}^T$ on $\mathcal{H}_{\mu_0}^*$ within the fundamental representation sector and act trivially within all other sectors and define $F_b \in \mathcal{L}(\mathcal{H}_{U_b})$ by

$$F_b = \otimes_\alpha F_{b,\alpha}. \tag{B.7}$$

It follows from Lemma B.4 that

$$F_b W_b = W_b U_b. \tag{B.8}$$

By assumption,

$$\|W_B V T_{b'}|\psi\rangle - V W_b T_{b'}|\psi\rangle\| \le \varepsilon, \tag{B.9}$$

$$\|\widetilde{W}_B V T_{b'}|\psi\rangle - V W_b U_b T_{b'}|\psi\rangle\| \le \varepsilon, \tag{B.10}$$

for isometries $W_B$ and $\widetilde{W}_B$. Now define $O_B := \widetilde{W}_B^\dagger F_b W_B$. By the triangle inequality, we have

$$\begin{aligned}
\|O_B V T_{b'}|\psi\rangle - V U_b T_{b'}|\psi\rangle\| \le &\|O_B V T_{b'}|\psi\rangle - \widetilde{W}_B^\dagger F_b V W_b T_{b'}|\psi\rangle\| \\
&+ \|\widetilde{W}_B^\dagger F_b V W_b T_{b'}|\psi\rangle - \widetilde{W}_B^\dagger V W_b U_b T_{b'}|\psi\rangle\| \\
&+ \|\widetilde{W}_B^\dagger V W_b U_b T_{b'}|\psi\rangle - V U_b T_{b'}|\psi\rangle\|,
\end{aligned} \tag{B.11}$$

for any $T_{b'}$. The first term on the righthand side is upperbounded by $\varepsilon$ because of (B.9) and the fact that $\|\widetilde{W}_B^\dagger F_b\|_\infty \le 1$. The second term vanishes because of (B.8). Finally, the third term is upperbounded by $\varepsilon$ because of (B.10). We conclude that

$$\|O_B V T_{b'}|\psi\rangle - V U_b T_{b'}|\psi\rangle\| \le 2\varepsilon. \tag{B.12}$$

This is almost what we want, except that $O_B$ is not necessarily unitary. However, we do have

$$\|O_B\|_\infty \le \|\widetilde{W}_B^\dagger\|_\infty \cdot \|F_b\|_\infty \cdot \|W_B\|_\infty \le 1. \tag{B.13}$$

We can define a unitary $U_B \in \mathcal{M}_B$ by

$$U_B := O_B(O_B^\dagger O_B)^{-1/2}, \tag{B.14}$$

as in the polar decomposition.[24] We then have

$$U_B^\dagger O_B = O_B^\dagger U_B = (O_B^\dagger O_B)^{1/2} \ge O_B^\dagger O_B, \tag{B.15}$$

where the final inequality follows from (B.13). Hence

$$\|(U_B - O_B)V T_{b'}|\psi\rangle\|^2 \le \|U_B V T_{b'}|\psi\rangle\|^2 - \|O_B V T_{b'}|\psi\rangle\|^2 \le 4\varepsilon, \tag{B.16}$$

where in the second inequality we used the fact that $\|O_B V T_{b'}|\psi\rangle\| \ge 1 - 2\varepsilon$ by (B.12). The result then follows by applying the triangle inequality and (B.16) to (B.12).

$$\square$$

---

[24]If $O_B$ is not invertible, we define $U_B$ to act as given in (B.14) on the support of $O_B^\dagger O_B$ and as the identity on the kernel of $O_B^\dagger O_B$ to ensure that it is unitary.

**Definition B.9** (One-design). A finite set $\mathcal{S} \subseteq \mathcal{M}_A$ of unitary matrices is said to form a one-design for $\mathcal{M}_A$ if

$$\frac{1}{|\mathcal{S}|} \sum_{\widetilde{U}_A \in \mathcal{S}} P_{(1,1)}(\widetilde{U}_A) = \int dU_A P_{(1,1)}(U_A) \,, \tag{B.17}$$

where $P_{(1,1)}(U_A)$ is any polynomial of degree at most one in the matrix elements of $U_A$ and at most one in the matrix elements of $U_A^*$, $dU_A$ is the Haar measure on unitaries in $\mathcal{M}_A$ normalized to $\int dU_A = 1$, and $|\mathcal{S}|$ is the size of the set $\mathcal{S}$.

**Remark B.10.** Let the set $\mathcal{S}_\alpha$ form a one-design for $\mathcal{L}(\mathcal{H}_{A_\alpha})$ (e.g. the generalized Pauli group on $\mathcal{H}_{A_\alpha}$) for each $\alpha$-sector in $\mathcal{M}_A$. Then the set

$$\mathcal{S} = \left\{ \oplus_\alpha \widetilde{U}_{A_\alpha} : \, \widetilde{U}_{A_\alpha} \in \mathcal{S}_\alpha \right\} ,$$

forms a one-design for $\mathcal{M}_A$.

**Remark B.11.** If $\mathcal{M}_A \cong \mathcal{M}_{A_1} \otimes \ldots \mathcal{M}_{A_n}$, the set of product unitaries $U_{A_1} \otimes \ldots U_{A_n}$ forms a one-design for $\mathcal{M}_A$.[25] This set (with the algebras $\mathcal{M}_{A_i}$ each describing operators at a local bulk site) played a central role in [14] because such operators cannot change the entanglement structure – and hence the max-EW – of the state and therefore should always be reconstructible.

**Theorem B.12** (One-design reconstruction). *Let $\mathcal{S} \subseteq \mathcal{M}_b \subseteq \mathcal{L}(\mathcal{H}_{\text{code}})$ form a unitary one-design for $\mathcal{M}_b$. If for a state $|\psi\rangle$ and every $U_b \in \mathcal{S}$, there exists $U_B \in \mathcal{M}_B$ such that for all $T_{b'}$*

$$\|U_B V T_{b'} |\psi\rangle - V U_b T_{b'} |\psi\rangle\| \le \varepsilon \,, \tag{B.18}$$

*then $\mathcal{M}_b$ can be state-specifically reconstructed from $\mathcal{M}_B$ with error $\varepsilon$ for the state $|\psi\rangle$.*

*Proof.* Define $\mathcal{H}_\mathcal{S}$ to be the Hilbert space spanned by the orthonormal basis $\{|\widetilde{U}_b\rangle : \widetilde{U}_b \in \mathcal{S}\}$. We define the isometry $\widetilde{W}_b : \mathcal{H}_{\text{code}} \to \mathcal{H}_{\text{code}} \otimes \mathcal{H}_\mathcal{S}$ by

$$\widetilde{W}_b = \frac{1}{\sqrt{|\mathcal{S}|}} \sum_{\widetilde{U}_b \in \mathcal{S}} |\widetilde{U}_b\rangle_\mathcal{S} \otimes \widetilde{U}_b \,. \tag{B.19}$$

Let $|\text{MAX}\rangle \in \mathcal{H}_{\text{code}} \otimes \mathcal{H}_R$ be maximally entangled. We have, by the definition of a unitary one-design,

$$\text{Tr}_\mathcal{S}[\widetilde{W}_b |\text{MAX}\rangle\langle\text{MAX}| \widetilde{W}_b^\dagger] = \frac{1}{|\mathcal{S}|} \sum_{\widetilde{U}_b \in \mathcal{S}} \widetilde{U}_b |\text{MAX}\rangle\langle\text{MAX}| \widetilde{U}_b^\dagger \tag{B.20}$$

$$= \int dU_b U_b |\text{MAX}\rangle\langle\text{MAX}| U_b^\dagger \tag{B.21}$$

$$= \text{Tr}_{U_b}\left[ W_b |\text{MAX}\rangle\langle\text{MAX}| W_b^\dagger \right] . \tag{B.22}$$

Because all purifications are related by an isometry, it follows that there exists an isometry $W_\mathcal{S} : \mathcal{H}_\mathcal{S} \to \mathcal{H}_{U_b}$ such that

$$W_\mathcal{S} \widetilde{W}_b |\text{MAX}\rangle = W_b |\text{MAX}\rangle \,. \tag{B.23}$$

---

[25]Since the set of product unitaries is infinite, it really only satisfies a slight generalization of Definition B.9 where the uniform measure on a finite set is replaced by the Haar measure on product unitaries. A true example of a finite one-design satisfying Definition B.9 as written is given by the set of products of elements of a one-design for each algebra $\mathcal{M}_{A_i}$.

This in turn implies $W_{\mathcal{S}}\widetilde{W}_b = W_b$. With this result in hand, we can define

$$W_B := \frac{1}{\sqrt{|\mathcal{S}|}} \sum_{\widetilde{U}_b \in \mathcal{S}} W_{\mathcal{S}} |\widetilde{U}_b\rangle_{\mathcal{S}} \otimes \widetilde{U}_B \,, \tag{B.24}$$

where $\widetilde{U}_B$ satisfies (B.18) for $\widetilde{U}_b$. We then have

$$\|W_B V T_{b'} |\psi\rangle - V W_b T_{b'} |\psi\rangle\| = \|W_B V T_{b'} |\psi\rangle - W_{\mathcal{S}} V \widetilde{W}_b |\psi\rangle\| \tag{B.25}$$

$$= \frac{1}{\sqrt{|\mathcal{S}|}} \sum_{\widetilde{U}_b \in \mathcal{S}} \|\widetilde{U}_B V T_{b'} |\psi\rangle - V \widetilde{U}_b T_{b'} |\psi\rangle\| \tag{B.26}$$

$$\leq \varepsilon \,. \tag{B.27}$$

$\square$

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
