# Peer review of "One-shot holography"

_SciPost Physics, doi:SciPost Phys. 16, 144 (2024)_

## Round 1 · Referee Report · Anonymous (Referee 1) · 2024-3-21

Report

Conditions like the generalized second law and the more recent (restricted) quantum focusing are believed to be fundamental consistency conditions in the semiclassical limit of quantum gravity. In AdS/CFT, these conditions imply that entanglement wedges of nested boundary CFT regions nest in the bulk, upholding CFT causality. In arXiv:2008.03319, subregion duality on time-symmetric bulk slices in AdS/CFT was refined: in general, the codimension-2 boundary between parts of the bulk which are fully encoded in one boundary subsystem and parts which are not was argued to have a certain vanishing "one-shot quantum expansion". If this is correct, it is then very important to find general (beyond time-symmetric slices) definitions of these codimension-2 boundaries. It would then be very plausible that conditions analogous to (restricted) quantum focusing exists to uphold the consistency of this more refined bulk-to-boundary dictionary.

This paper accomplishes the task of providing fully covariant definitions of these refined bulk subregions. And appropriately, conjectures conditions analogous to QFC which upholds various consistency conditions of the dictionary. The paper is well-organized and very clear. I therefore recommend it for publication as is

---

## Round 1 · Referee Report · Anonymous (Referee 2) · 2024-3-22

Report

This paper explores the reconstruction of bulk subregions from the boundary, in the context of AdS/CFT correspondence. The authors attempt to refine the entanglement wedge reconstruction by defining “min-entanglement wedge” and “max- entanglement wedge”. Given a boundary subregion B, the authors conjecture that max-EW(B) is the largest bulk region (b) such that all information within b can be reconstructed from the boundary region B. Whereas, the min-EW(B) is the smallest bulk region (b) such that no information outside b can be reconstructed from the boundary region B. Usually, we expect these wedges to be identical. However, the authors point that there are states in semiclassical gravity where entanglement wedge doesn’t exists and in such situations, the min- and max- entanglement wedges do not overlap. It would be helpful if authors include some examples or details of semiclassical states where the entanglement wedge doesn't exist.

The covariant definition of min- and max- entanglement wedges provided by the authors are based on the one-shot min- and max- quantum entropies. The authors define the one-shot entropies and study their properties in details. They also extend some known results to finite dimensional von-Neumann algebras.

In order to support the conjectured interpretation of max- and min- EW, the authors prove various consistency conditions. They show that in the cases where min- and max- EW coincides, the two wedges equal the entanglement wedge. According to the conjecture, all information within max-EW can be reconstructed from the boundary subregion, whereas, no information outside min-EW can be reconstructed. This is consistent, only if max-EW is contained inside min-EW, a result that the authors prove. The authors also show that max-EW contains the causal wedge and that the max-entanglement wedges show a nested structure that is, if A is contained in B, then max-EW[A] is contained in max-EW[B]. The proofs of these properties required the authors to assume a version of quantum focusing conjecture, that they explain in the paper.

Overall, the main results of the paper are conjectural, but interesting.

Requested changes

  1. Explain the notation q \le p for operators p and q in definition 2.14. How is the ordering defined?
  2. Authors may consider including a diagram to illustrate the definitions of wedge, wedge union and edge of the wedge in section 3.1.
  3. Though they have provided references, the authors may consider including some examples of states in semiclassical gravity where the min- and max- entanglement wedges are not identical and the entanglement wedge doesn't exist.

---

## Round 2 · Referee Report · Anonymous (Referee 2) · 2024-5-13

Report

I am satisfied with the modifications made by the authors in response to my previous report. I happily recommend the paper for publishing.

Recommendation

Publish (easily meets expectations and criteria for this Journal; among top 50%)

---

## Round 2 · Author Response

We thank the reviewers for their helpful comments and have implemented the suggested changes.

---

## Round 2 · List of Changes

1. We added a sentence explaining the meaning of q \le p above definition 2.14
  2. Immediately prior to the definitions in section 3.1, we indicate a section of a reference that contains figures for each definition.
  3. We included a footnote in the introduction stating an example of a state in semiclassical gravity where the min- and max-entanglement wedges are not identical and the entanglement wedge is not well-defined.

---

## Editorial Decision

published